# A methodological framework for the evaluation of short-range flash-flood hydrometeorological forecasts at the event scale

Maryse Charpentier–Noyer[1], Daniela Peredo[2,3], Axelle Fleury[4], Hugo Marchal[4], François Bouttier[4], Eric Gaume[1], Pierre Nicolle[1], Olivier Payrastre[1], and Maria-Helena Ramos[3]

[1]GERS/LEE, Univ Gustave Eiffel, Nantes, 44344, France
[2]UMR Metis, Sorbonne Université, Paris, France
[3]Université Paris-Saclay, INRAE, UR HYCAR Antony, 92160, France
[4]CNRM, Université de Toulouse, Météo-France, CNRS, Toulouse, France

**Correspondence:** Maryse Charpentier–Noyer (maryse.charpentier-noyer@developpement-durable.gouv.fr)

**Abstract.**

This paper presents a methodological framework designed for the event-based evaluation of short-range hydro-meteorological ensemble forecasts, in the specific context of an intense flash-flood event characterized by high spatio-temporal variability. The proposed evaluation adopts the point of view of end-users in charge of the organization of evacuations and rescue operations at a regional scale. Therefore, the local exceedance of discharge thresholds should be anticipated in time and accurately localized. A step-by-step approach is proposed, including first an evaluation of the rainfall forecasts. This first step helps to define appropriate spatial and temporal scales for the evaluation of flood forecasts. The anticipation of the flood rising limb (discharge thresholds) is then analyzed at a large number of ungauged sub-catchments, using simulated flows and zero-future rainfall forecasts as references. Based on this second step, several gauged sub-catchments are selected, at which a detailed evaluation of the forecast hydrographs is finally achieved.

This methodology is tested and illustrated on the October 2018 flash-flood, which affected part of the Aude River basin (south-eastern France). Three ensemble rainfall nowcasting research products recently proposed by Météo-France are evaluated and compared. The results show that, provided that the larger ensemble percentiles are considered (75% percentile for instance), these products correctly retrieve the area where the larger rainfall accumulations were observed, but have a tendency to overestimate its spatial extent. The hydrological evaluation indicates that the discharge threshold exceedances are better localized and anticipated if compared to a naive zero-future rainfall scenario, but at the price of a significant increase of false alarms. Some differences in the performances between the three ensemble rainfall forecast products are also identified.

Finally, even if the evaluation of ensemble hydro-meteorological forecasts based on a low number of documented flood events remains challenging due to the limited statistical representation of the available data, the evaluation framework proposed herein should contribute to draw first conclusions about the usefulness of newly developed rainfall forecast ensembles for flash-flood forecasting purpose, and about their limits and possible improvements.

# 1 Introduction

Flash floods contribute in a significant proportion to flood-related damages and fatalities in Europe, particularly in the Mediterranean countries (Barredo, 2006; Llasat et al., 2010, 2013; Petrucci et al., 2019). As an illustration, over the period 1989-2018, four of the eight most damaging floods in France were flash floods, each having caused insurance losses that exceeded 500 million euros according to the French Central Reinsurance Fund (CCR, 2020). These floods are characterized by fast dynamics and high specific discharges (Gaume et al., 2009; Marchi et al., 2010), which largely explain their destructive power. They generally result from heavy precipitation events, typically exceeding hundreds of millimeters of rainfall totals in less than 6 hours, falling over river basins of less than 1000 km$^2$ of drainage area. They are also characterized by a high spatio-temporal variability and limited predictability (Georgakakos, 1986; Borga et al., 2008). Improving the capacity of flood monitoring and forecasting systems to anticipate such events is a key factor to limit their impacts and improve flood risk management. Although several operational flash-flood monitoring services based on weather radar rainfall are already implemented worldwide (Price et al., 2012; Clark et al., 2014; de Saint-Aubin et al., 2016; Javelle et al., 2016; Gourley et al., 2017; Park et al., 2019), these systems can only offer limited anticipation due to the short response times of the affected catchments.The integration of short-range, high resolution rainfall forecasts in flash-flood monitoring services is required to increase anticipation times beyond current levels (Collier, 2007; Hapuarachchi et al., 2011; Zanchetta and Coulibaly, 2020).

Weather forecasting systems that are well suited to capture heavy precipitation events have been developed in the last decade with the emergence of high resolution and convection-permitting numerical weather prediction (NWP) models (Clark et al., 2016). The spatial and temporal resolution of such models (typically 1 km and 1 min) are more relevant to the scales of the (semi-)distributed hydrological models that are commonly used in flash-flood monitoring and forecasting systems. Convection-permitting NWP models may also be combined with radar measurements through assimilation and/or blending techniques to obtain improved and seamless short-range rainfall forecasts (Davolio et al., 2017; Poletti et al., 2019; Lagasio et al., 2019). Despite these advances, the use of high resolution rainfall forecasts to issue flash-flood warnings still faces numerous challenges, mainly due to the uncertainties in the temporal distribution and the spatial location of the high rainfall accumulations cells over small areas (Silvestro et al., 2011; Addor et al., 2011; Vincendon et al., 2011; Hally et al., 2015; Clark et al., 2016; Armon et al., 2020; Furnari et al., 2020). Accurate forecasts and a reliable representation of forecast uncertainties are thus necessary to provide useful flash-flood warnings based on outputs of NWP models. The question of quantifying uncertainties in hydrometeorological forecasting systems has been increasingly addressed through ensemble forecasting approaches (e.g. Valdez et al., 2022; Bellier et al., 2021; Thiboult et al., 2017). Several ensemble flash-flood forecasting chains have been proposed in the literature, involving either convection-permitting NWP models for early warnings (Silvestro et al., 2011; Addor et al., 2011; Vié et al., 2012; Alfieri and Thielen, 2012; Davolio et al., 2013, 2015; Hally et al., 2015; Nuissier et al., 2016; Amengual et al., 2017; Sayama et al., 2020; Amengual et al., 2021), or radar advection approaches and/or radar data assimilation in NWP models for very short-range forecasting (Berenguer et al., 2011; Vincendon et al., 2011; Silvestro and Rebora, 2012; Davolio et al., 2017; Poletti et al., 2019; Lagasio et al., 2019).

New approaches to flash-flood forecasting need to be appropriately evaluated, and cannot rely only on the evaluation of the high resolution rainfall forecasts used as input. In one sense, flood forecasting verification can be seen as a form of fuzzy verification of rainfall forecasts (Ebert, 2008; Roberts and Lean, 2008), accounting for the averaging effect and the non-linearity of the rainfall-runoff process, and also for the positions of the watershed limits. Flood forecast evaluation is generally based on long time series of observed and forecast data. However, in the case of flash-flood forecast evaluation, working on long

time series is often not possible. At first, high resolution ensemble rainfall forecasts from convection-permitting NWP models are rarely available for long periods of re-forecasts. This is due to the fast evolution of input data (type, availability) and the frequent updates brought to the NWP models or the data assimilation approaches (Anderson et al., 2019). Secondly, because of the limited frequency of occurrence of heavy precipitation events triggering flash floods, the data-sets available for evaluation often include only a few significant flood events, which highly limits the possibility of satisfying the typical requirements for

a statistically robust evaluation of hydrological forecasts (Addor et al., 2011; Davolio et al., 2013). Therefore, the evaluation of experimental flash-flood forecasting systems based on newly developped rainfall forecast products, has often to begin with event-based evaluations. Even if such preliminary evaluations cannot provide a complete picture of the performance of the forecasting systems, they may bring useful information about their value for some rare, high-impact flood events, and they can help deciding if the tested systems are worth running in real time for preoperational evaluations at larger space and time scales.

They can also be useful communication tools to exchange and get feedback from the end users of forecasts in the systems design phase (Dasgupta et al., 2023). Finally, event-based and statistical evaluations of flood forecast systems should probably not be opposed but rather considered as complementary (ECMWF, 2022).

Event-based evaluations generally rely on the visual inspection of forecasts against observed hydrographs or on the assessment of the anticipation of exceedances over pre-defined discharge thresholds, and such often at a few gauged outlets, where the main

hydrological responses to the high rainfall accumulations were observed during the event (Vincendon et al., 2011; Vié et al., 2012; Davolio et al., 2013; Hally et al., 2015; Nuissier et al., 2016; Amengual et al., 2017; Lagasio et al., 2019; Sayama et al., 2020). When different forecast runs are available for the same event and along its duration (e.g., short-range forecasts generated by NWP models from different forecast initialization cycles) and/or when different basins are affected by the same event, statistical scores and frequency analyses, such as the RMSE, the CRPS, contingency tables or ROC curves, may also be used

to provide a synthetic evaluation of the performance of the forecasts for the event being evaluated (Davolio et al., 2017; Poletti et al., 2019; Sayama et al., 2020). Although widely used in the scientific literature and in post-event reports, these evaluation frameworks raise several methodological questions: i) the focus on one event or a few typical severe events may generate an event-specific evaluation, which might not be reproducible across different events or might not be statistically representative of forecast performance for other future events; ii) scores that offer a synthesis of performance over spatial and temporal

scales might conceal the internal (in space and in time) variability of forecast performance (over different forecast initialization times and along lead times); iii) forecast evaluations that focus on gauged outlets that display the main hydrological responses to rainfall only offers a partial view of the forecasting system's performance, notably when impacts are also observed at ungauged sites and/or when significant spatial shifts exist between observed and forecast rainfalls. Therefore, the evaluation of short-range flash-flood forecasts at the event scale requires considering specific forecast quality attributes evaluated at gauged

sites (where observations are available) but also a more regional-scale evaluation at ungauged sites, in order to achieve a more robust evaluation of forecast performance. Several authors already pointed out the interest of providing such regional scale hydrological evaluations (Silvestro and Rebora, 2012; Davolio et al., 2015; Anderson et al., 2019; Sayama et al., 2020).

In this paper, an evaluation of three new ensemble rainfall forecast products is presented in the perspective of their use for flash flood forecasting. The evaluated products have been specifically developed by the French meteorological service (Météo-France) to generate short-range rainfall forecasts (1 to 6-hours of lead time) that can potentially better capture Mediterranean heavy precipitation events. The products comprise the French AROME-EPS reference ensemble forecast (Bouttier et al., 2012; Raynaud and Bouttier, 2016), and two experimental products merging AROME-EPS and another convection permitting NWP model-AROME-NWC (Auger et al., 2015), with optional incorporation of spatial perturbations as post-processing (Vincendon et al., 2011). Since the two experimental products have been released only for the autumn 2018 period in France, the evaluation can only be based on one significant flash-flood event, i.e. the heavy flood that occurred in the Aude River basin on October 15[th], 2018. Therefore, a new framework is proposed for the evaluation of flash-flood hydro-meteorological ensemble forecasts at the event scale. This approach is based on the combination and adaptation of well known evaluation metrics, with the objective to provide a detailed and as meaningful as possible analysis of the considered event. The evaluation is mainly focused on the capacity of the hydrometeorological forecasts to anticipate the exceedance of predefined discharge thresholds and to accurately localize the affected streams within the region of interest. These are two essential qualities of hydrometeorological forecasts that are needed to plan rescue operations in real time. The forecast-based financing approach developed for humanitarian actions adopts a similar pragmatic approach, but with the aim to release funding and trigger short-term actions in disaster-prone areas worldwide (Coughlan de Perez et al., 2015). Others methods with a specific interest for operational considerations have also been recently proposed for the case of deterministic forecasts (Lovat et al., 2022).

In the following, Section 2 presents the step-by-step evaluation framework proposed for the event-scale evaluation of ensemble forecasts. Section 3 presents the case study, the data and models used to produce discharge forecasts. In Section 4, the obtained results are presented and evaluated. Section 5 discusses the main outcomes, while Section 6 summarizes the conclusions and draws the perspectives of this study.

## 2   Methodology for an event-scale evaluation of hydro-meteorological ensemble forecasts

The proposed evaluation framework aims at determining if the magnitude of the floods generated by heavy precipitation events can be correctly anticipated based on ensemble rainfall forecasting products. It is considered that such products might not perfectly capture the complex spatial and temporal patterns of the observed rainfall, although they might still be useful to inform flood risk decision-making. More precisely, the question of anticipating high discharges that might exceed predefined discharge thresholds is addressed. The evaluation should not focus only on selected river sections, but offer a comprehensive view of anticipation capacities for the whole river network, including ungauged rivers.

Another challenge of the event-scale evaluation is to select a limited number of aggregated criteria that help drawing sound conclusions, owing to the possible high spatial and temporal variability of rainfall and runoff values and of model performance,

as well as to the many possible combinations of time steps, forecast lead times and locations along the river network, that need to be considered in the evaluation. For this reason, a step-by-step approach is proposed (Figure 1). It is first based on an initial assessment of the rainfall forecasts, with a focus on the time and space windows of observed or forecast high rainfall accumulations. Then, a geographical analysis of the anticipation capacities of the ensemble flood forecasts is performed at a large number of ungauged outlets, focusing on the most critical phase of the floods (hydrograph rising limbs). Based on this second step, a detailed evaluation of the performance of the flood forecasts is conducted at some selected representative catchment outlets. These different steps are described below.

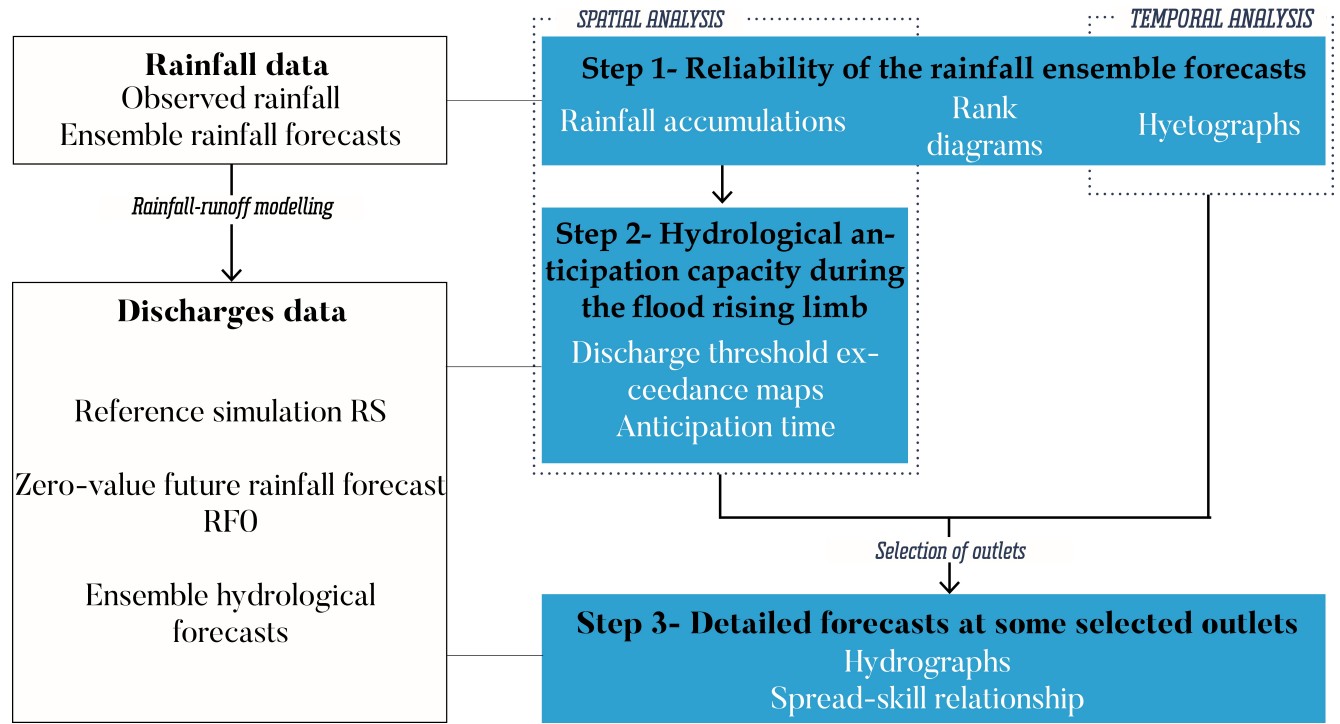

**Figure 1.** Overall principle of the proposed evaluation framework with its three steps of evaluation along the hydrometeorological forecasting chain

## 2.1 Step 1: Reliability of the rainfall ensemble forecasts

The twofold objective of this initial phase is to analyze the quality of the rainfall forecasts and to define the relevant spatial and temporal scales to be used in the subsequent analyses. Three different aspects are considered for a comparison of observed and forecast rainfall values:

– The hyetographs of the average rainfall intensities over the studied area are first plotted for each rainfall ensemble product and the different forecast lead times. They are used to assessing if, on average, the time sequence and the magnitude of the

rainfall intensities are well captured by the products. A reduced time window, where significant intensities are forecast or measured, is selected for the next steps of the analysis. This time window is hereafter called "hydrological focus time" (HFT).

- Maps of the sum of forecast and observed hourly rainfall amounts during the HFT are then generated to assess if the areas where high rainfall accumulations were predicted and actually occurred coincide. One map is produced per forecast product, forecast lead time and ensemble percentile. These maps compare the spatial distribution of accumulated rainfalls from the aggregation of all forecast runs that were delivered during the event. It is thus the rainfall totals that are first evaluated. These maps also help to delineate the areas affected by high totals of measured or forecast rainfalls. The next evaluation steps will focus on these areas, which are hereafter designated as "hydrological focus area" (HFA).

- Classical rank diagrams (Talagrand and Vautard, 1997; Hamill, 2001) are computed for the entire HFA domain and HFT period, for each forecast product and specific lead times. These diagrams can obviously not be considered as statistically representative of the performance of rainfall forecasts for other events, but they just aim here to quickly detect or confirm possible systematic biases or lack of variability in the forecast ensemble rainfall products, for the considered event. The diagrams are calculated considering all rainfall pixels over the entire HFA, and also for particular high rainfall intensities of interest (in mm/h). Rank diagrams show the frequencies at which the observation falls in each of the ranks of the ensemble members for each forecast product, when members are sorted from lowest to highest. They are used to determine the reliability of ensemble forecasts and to diagnose errors in its mean and spread (Hamill, 2001). Typically, sloped diagrams will indicate consistent biases in the ensemble forecasts (under- or over-estimation of the observations); U-shaped or concave diagrams are a signal of a lack of variability in the distribution given by the ensemble forecasts; an excess of variability will result in a rank diagram where the middle ranks are overpopulated.

## 2.2 Step 2: Hydrological anticipation capacity during the flood rising limb

The objective of this second step is to characterize the anticipation capacity of pre-selected discharge threshold exceedances for the whole HFA (including a large number of ungauged outlets) and during the most critical phase of the event (i.e. the flood rising limb), based on the ensemble discharge forecasts. The evaluation is essentially based on a classical contingency table approach (Wilks, 2011), with some important adaptations aiming to focus the analysis on the most critical time window from a user perspective (runs of forecasts preceding the threshold exceedance), and to aggregate the forecasts issued during this time window, independently of the lead-times (i.e., a hit is considered if at least one of the forecasts has exceeded the threshold at any lead time). Based on this framework, forecasting anticipation times are also computed (see Appendix B for a detailed description of the implemented method).

To ensure a certain homogeneity over the focus area (HFA), the same return period is used to define the discharge thresholds at the different outlets. A 10-year return period was considered appropriate in this study given the magnitude of the flood event investigated (see Section 3), but it can be adapted to the intensity of any evaluated flood event. In addition, maps can be drawn for each forecasting system, each threshold level and ensemble percentile, to show the spatial distribution of the outlets that

display hits, misses, false alarms or correct rejections. The corresponding histograms of misses, false alarms and hits, sorted by categories of anticipation time, can also be drawn to provide an overall performance visualization for the comparison of the different systems when considering all the HFA focus area. Finally, ROC curves based on the above definitions of hits, misses and false alarms are drawn to help to rank the methods independently of a specific percentile (see Appendix B). Again here, these ROC curves should not be considered as statistically representative of the performances of forecasts for future events, but rather as a synthetic comparison of the available forecats for the considered event.

To enable the integration of ungauged outlets in the analysis, the hydrographs simulated with observed radar rainfall are considered as reference values for the computation of the evaluation criteria (reference simulation, RS hereafter). An additional runof forecast is generated to help to interpret the results: it corresponds to a zero-value future rainfall scenario (RF0 hereafter), i.e. forecasts are based on the propagation along the river network of past generated runoff only. This scenario helps to distinguish the part of the anticipation that can be explained by the propagation delays of the generated runoff flows along the river network, and the part that is actually attributable to the rainfall forecasts.

## 2.3 Step 3: Detailed forecast analysis at some selected outlets

The main objective of this last step is to make the connection between the discharge thresholds anticipation results (step 2), and the detailed features of the ensemble hydrological forecasts. This is achieved by an in depth analysis of forecast hydrographs at different outlets, covering the whole HFT. The outlets are selected according to the results appearing on the maps elaborated in step 2, with the objective to cover the various situations (hit, miss or false alarm). The evaluation is based on the visual analysis of the forecast hydrographs for fixed forecast lead times (Berenguer et al., 2005) and on the spread-skill relationship, which evaluates the consistency between ensemble spread and ensemble mean error for the different lead times (Fortin et al., 2014; Anctil and Ramos, 2017). The spread-skill score for each lead time is obtained by comparing the RMSE of the ensemble mean (the skill) and the average of the standard deviations of the ensemble forecasts (the spread), as suggested by Fortin et al., 2014. The advantage of this score is that it can be easily calculated from the forecast outputs and provides also a measure of reliability of the ensembles (Christensen, 2015; Hopson, 2014). Finally, gauged outlets can also be examined in this phase, allowing the evaluation framework to also incorporate the hydrological modeling errors in the analysis.

## 3 Case study, data and models

### 3.1 The October 2018 flash-flood event in the Aude River basin

The Aude River basin is located in southern France (Figure 2). It extends from the Pyrenees mountains, in its South upstream edge, to the Mediterranean Sea. Its drainage area is 6,074 km$^2$. The climate is Mediterranean, with hot and dry summers and cool and wet winters. The mean annual precipitation over the basin is about 850 mm. High discharges are observed in winter and spring, but the major floods generally occur in autumn, and result from particularly intense convective rainfall events.

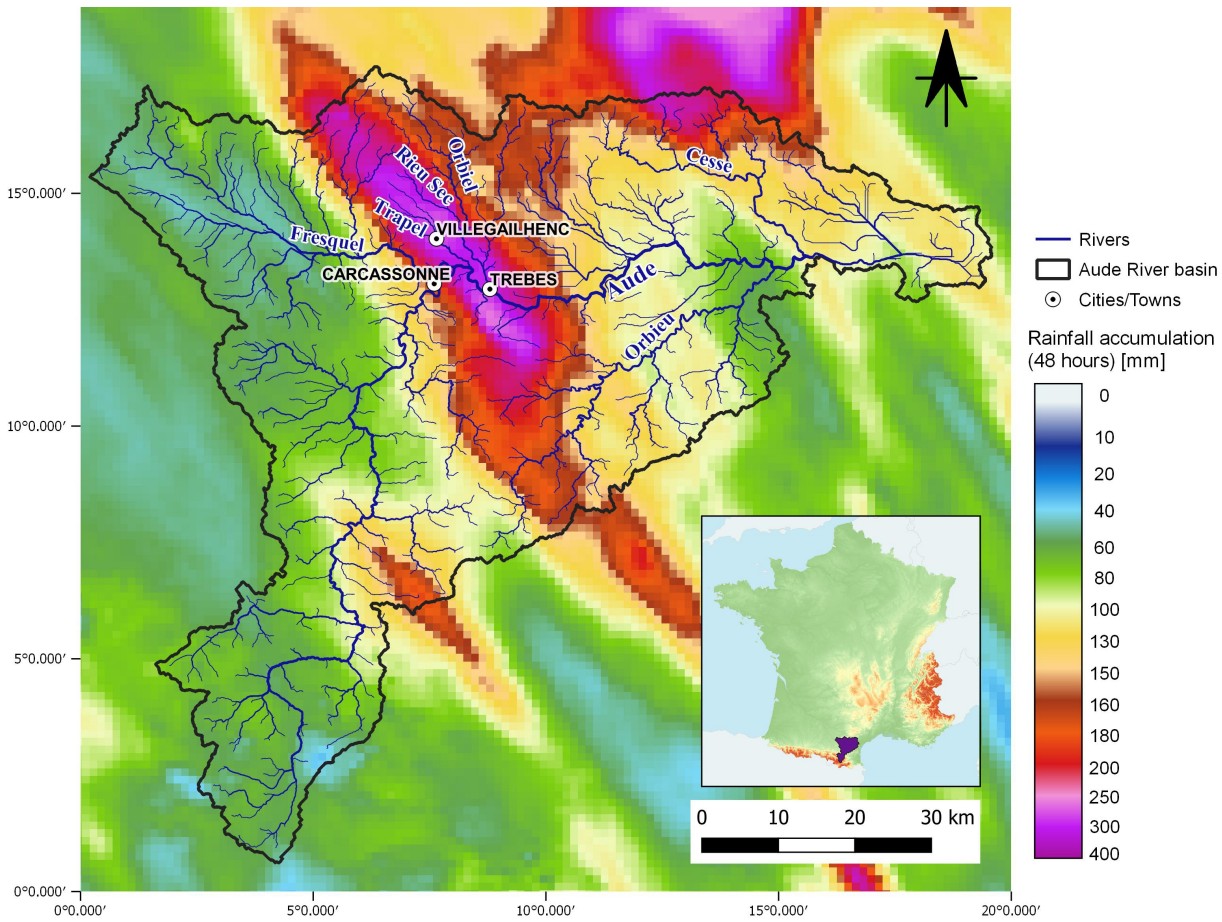

**Figure 2.** The Aude River basin, its river network, and the rainfall accumulations observed from 14 October 2018 00:00 to 15 October 2018 23:00, according to the ANTILOPE J+1 quantitative precipitation estimates (see Section 3.2

).

A major precipitation event occurred in this area from the 14<sup>th</sup> to the 16<sup>th</sup> October 2018, with particularly high rainfall accumulations during the night of 14<sup>th</sup> to 15<sup>th</sup> October. The maximum rainfall accumulations hit the intermediate part of the Aude River basin, just downstream the city of Carcassonne (Figure 2). Up to 300 mm of point rainfall accumulation was observed (Caumont et al., 2021). The maximum accumulated rainfall amounts over short durations were also extreme: up to 60 mm in one hour and 213 mm in six hours recorded at Villegailhenc (Figure 2), while the local 100-year rainfall accumulation is 200 mm in six hours (Ayphassorho et al., 2019). By its intensity and spatial extent, the 2018 event nears the record storm and flood event that hit the Aude River basin in November 1999 (Gaume et al., 2004).

The October 2018 floods of the Aude River and its tributaries led to the activation of the highest (red) level of the French flood warning system "Vigicrues" (Vigicrues.gouv.fr). They caused 14 deaths, about 100 injuries and between 130 and 180

million euros of insured damages (CCR, 2018). The floods were particularly severe on the tributaries affected by the most intense rains in the intermediate part of the Aude basin (Trapel, Rieu Sec, Orbiel, Lauquet), and also on the upstream parts of the Cesse and Orbieu tributaries (see Figure 2). The peak discharge on the main Aude River at Trèbes reached the 100-year value (Ayphassorho et al., 2019) only 9 hours after the onset of the rainfall event. In contrast, the peak discharges were high but not exceptional and caused limited inundation in the upstream and downstream parts of the Aude River basin. The high observed spatial and temporal rainfall and runoff variability makes this flash-flood event a challenging and interesting case study, for the event-scale evaluation framework of short range flood forecasts proposed in this study.

### 3.2 Observed hydrometeorological data

High resolution quantitative precipitation estimates(QPEs) for the event were obtained from the Météo-France's ANTILOPE algorithm (Laurantin, 2008), which merges operational weather radar (30 radars operating in October 2018) and rain gauge observations, including not only real time observations, but all observations available one day after the event. These QPEs are called ANTILOPE J+1 hereafter. A comprehensive reanalysis of this product is available for the period from 1 January 2008 to 18 October 2018, at the hourly time step and 1 km by 1 km spatial resolution. This product was used in this study to calibrate and run the hydrological models.

Discharge series were retrieved from the French Hydro database (Leleu et al., 2014; Delaigue et al., 2020) for the 31 stream gauges located in the Aude River basin, over the period 2008-2018. Additionally, peak discharges of the October 2018 flood event at ungauged locations were estimated during a post-flood field campaign (Lebouc et al., 2019) organized within the Hydrological cycle in the Mediterranean eXperiment (HyMeX; Drobinski et al., 2014) research program.

Evapotranspiration values, necessary to run the continuous rainfall-runoff model, were estimated using the Oudin formula (Oudin et al., 2005), based on temperature data extracted from the SAFRAN meteorological reanalysis produced by Météo-France on an 8 km x 8 km square grid (Vidal et al., 2010).

### 3.3 AROME-based short-range rainfall ensemble forecast products

Rainfall forecasts are based on Météo-France's AROME-France NWP model (Seity et al., 2011; Brousseau et al., 2016). AROME-France is an operational limited area model that provides deterministic weather forecasts up to two days ahead. Its high horizontal resolution allows to explicitly resolve deep convection, which is well suited to forecast heavy precipitations. Three different AROME-based short-range rainfall ensemble forecast products were evaluated in this work:

- **AROME-EPS** is the operational ensemble version of AROME-France (Bouttier et al., 2012; Raynaud and Bouttier, 2016). AROME-EPS results from perturbations of model equations, initial conditions and large-scale coupling of the NWP model. In 2018, AROME-EPS was a 12-member ensemble forecast, updated every 6 hours (four times per day: at 3h, 9h, 15h and 21h UTC), and providing forecasts up to two days ahead.

- **pepi** is an experimental product aiming to address the specific requirements for short-range flash-flood forecating, i.e. high temporal and spatial resolutions, high refresh rate, and seamless forecasts. **pepi** is a combination of AROME-EPS

and AROME-NWC (Auger et al., 2015). AROME-NWC is a configuration of AROME-France designed for now-casting purposes; it is updated every hour and provides forecasts up to 6 hours ahead. In order to take into account sudden weather changes, AROME-NWC is used with time lagging (Osinski and Bouttier, 2018; Lu et al., 2007), which consisted here in using the last 6 successive runs of AROME-NWC instead of using only the most recent run. The resulting pepi product provides forecasts for a maximum lead time of 6-hours. It combines 12 members from the last available AROME-EPS run, and 1 to 6 members from AROME-NWC, depending on the considered lead time. The resulting number of members varies between 13 (for a 6-hour lead time) and 18 (for a 1-hour lead time).

– **pertDpepi** is a second experimental product, obtained by shifting the original pepi members 20 km in the four cardinal directions on top of the unperturbed pepi members, to account for uncertainties in the forecast rainfall location. The number of members is five times the number of the "pepi" ensemble, i.e. varies from 65 to 90 members depending on the lead time. This product is based on the concepts proposed by Vincendon et al. (2011), but uses a simpler framework to derive the test-product specifically designed for this study. The shift scale of 20 km represents a typical forecast location error scale: according to Vincendon et al. (2011), 80% of location errors are less than 50 km. The value of 20 km has been empirically tuned to produce the largest possible ensemble spread on a set of similarly intense precipitation cases, without noticeably degrading the ensemble predictive value as measured by user-oriented scores such as the area under the ROC curves.

These three rainfall forecast products have the same spatial (0.025° by 0.025°) and temporal (1 hour) resolutions, and cover the same spatial window as AROME (metropolitan territory of France). For the comparison with rainfall observations, they were disaggregated on the corresponding 1 km by 1 km grid. In order to issue hydrological forecasts every hour, the last available runs of AROME-NWC and AROME-EPS (and the resulting pepi and pertDpepi ensembles) were systematically used, according to the products updates. The AROME-NWC and AROME-EPS forecasts were supposed to be immediately available for each update (i.e. the computation delays were not considered).

## 3.4 Rainfall-runoff models

The ensemble rainfall forecasts were used as input of two rainfall-runoff models, which differ in their resolution and structure. The GRSDi model is a semi-distributed continuous hydrological model adapted from the GRSD model (Le Moine et al., 2008; Lobligeois, 2014; De Lavenne et al., 2016) to better simulate autumn Mediterranean floods that typically occur after long dry summer periods (Peredo et al., 2022). The Cinecar model is an event-based distributed model, specifically developed to simulate flash-floods of small ungauged headwater catchments, with limited calibration needs (Versini et al., 2010; Naulin et al., 2013; Le Bihan, 2016). Both models were calibrated against observations and presented good performance for the 2018 flood event, as presented below, where we provide more information about the models and their implementations.

The objective of this study is not to compare the rainfall-runoff models. Since the RS hydrographs (hydrographs simulated with ANTILOPE J+1 rainfall observations) are systematically used as reference for the evaluation of the flood forecasts, the evaluation results should not be directly dependent on the rainfall-runoff model but rather on the nature of the rainfall forecasts

used as input. The interest of using two models here is mainly to strengthen the evaluation, by involving two complementary models in terms of resolution and calibration approach: a) because of its high spatial resolution, the Cinecar models helps to extend the evaluation of discharge threshold anticipation to small ungauged catchments, b) because it was not specifically calibrated on the 2018 event (calibration on the whole 2008-2018 period), the GRSDi model offers an evaluation of the total forecast errors at gauged outlets, including both the rainfall forecasts errors and the rainfall-runoff modeling errors. This is achieved by the comparison of flood forecasts with both RS hydrographs and observed hydrographs. However, the proposed evaluation framework could also be applied by using one unique rainfall runoff model.

**GRSDi model**

GRSDi is a soil moisture accounting, reservoir-based hydrological model that runs at an hourly time step with rainfall and evapotranspiration as input data. The model has 7 parameters to be calibrated against observed discharges. For its implementation, the Aude River basin was divided into 123 modelling units (gray contours in Figure 3a) of approximately 50 km$^2$. The model calibration and validation were performed for the same period (2008-2018), including the October 2018 flood event: 16 gauged outlets were selected for the calibration and 15 for the validation. The averaged KGE (Kling-Gupta Efficiency - Gupta et al., 2009) values were of 0.80 (0.71) for the 16 calibration outlets (15 validation outlets), which indicates good model performance, except for one validation outlet, where a low KGE value of 0.1 was obtained (Figure 3a). After visual inspection of the simulated discharges, this low performance was explained by an overestimation of the base flow, with however limited impact on the peak discharges during the high flows and flood events. The model and its performance evaluation is presented in details in Peredo et al. (2022).

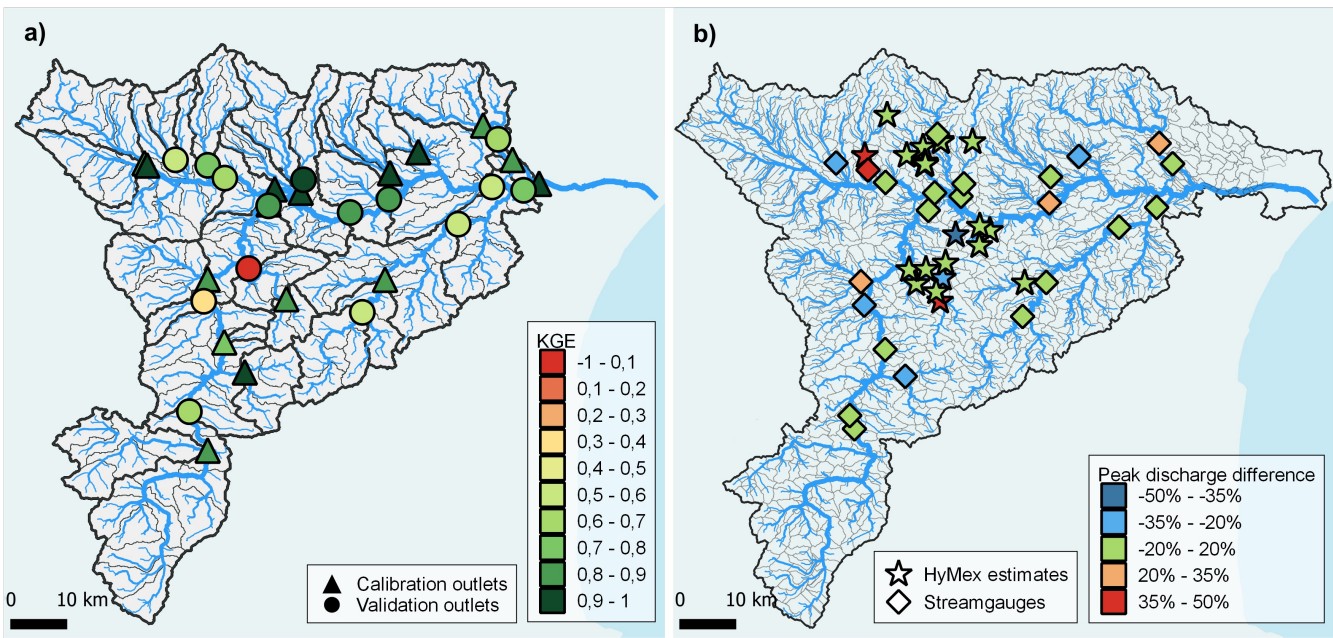

**Figure 3.** Calibration-validation results for the two rainfall-runoff models: a) KGE values at calibration and validation outlets (stream gauges) for the GRSDi model and the period 01/10/2008 at 0h to 15/10/2018 at 23h, b) Peak discharge difference (in %) between CINECAR simulated discharges and HyMex (HYdrological cycle in the Mediterranean EXperiment) estimates (stars - Lebouc et al., 2019) or stream gauges (diamonds) when the model is calibrated over the Aude 2018 flood event.

**CINECAR model**

The Cinecar model combines a SCS-CN (Soil Conservation Service - Curve Number, $CN$) model for the generation of effective runoff on the hill slopes and a kinematic wave propagation model on the hill slopes and in the stream network. The model runs at a 15 min time-step and was specifically developed to simulate fast runoff during flash floods, with a lesser focus on reproducing delayed recession limbs. The main parameter of the model requiring calibration is the Curve Number (Naulin et al., 2013). For the implementation of the model, the Aude River basin was divided into 1174 sub-basins with an average area of 5 km$^2$ (gray contours in Figure 3b). The shapes of the river reaches and hill slopes are simplified in the model but their main geometric features (slopes, areas, length, width) are directly extracted from the Digital Terrain Model (DTM). The $CN$ values, first fixed on the basis of soil types and antecedent conditions, were further tuned to reach a better agreement between simulated and observed peak discharges for the Aude 2018 flood (Hocini et al., 2021). The resulting model was overall consistent with field observations, with errors on peak discharges generally comprised in the $\pm 20\%$ range (Figure 3b).

# 4 Results

## 4.1 Overall performance of the rainfall ensemble forecast products

Figure 4 shows the temporal evolution of the hourly observed rainfall (ANTILOPE J+1) and the hourly forecast rainfall for the three ensemble products (AROME-EPS, pepi and pertDpepi) at 3-hours of lead time from the 14 [th] October 07:00 to the 15 [th] October 19:00. Figures corresponding to 1-hour and 6-hours lead times are presented in Appendix A (Figures A1 and A2). Rainfall intensities (mm/h) are averaged over the entire Aude River basin area. Overall, the observed rainfalls are well captured by the forecast products. The ensemble forecast distributions are similar among the three products, except for some time steps corresponding to the most intense rainfall period, where the added value of the AROME-NWC model used in the pepi and pertDpepi products can be noticed. As expected, the spread of the ensemble forecasts increases from AROME-EPS to pepi and pertDpepi, with some rare exceptions. This is in agreement with the increase in the number of ensemble members.

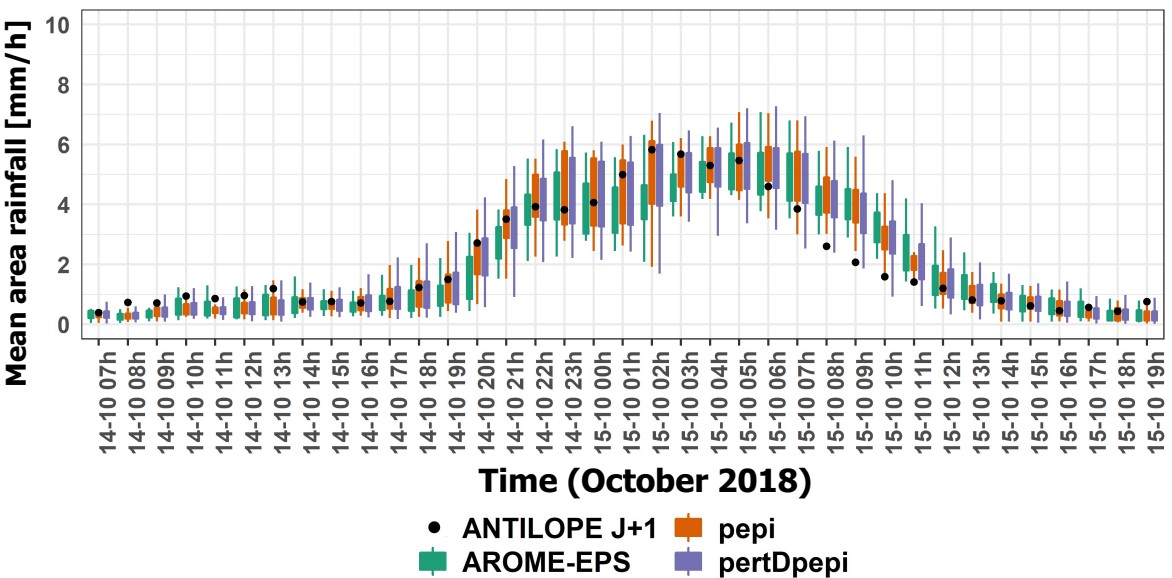

**Figure 4.** Temporal evolution of observed hourly rainfall rates (black dots), and 3- hours lead time ensemble forecasts (boxplots), during the October 2018 flood event. Rainfall rates are averaged over the Aude River basin (6074 km$^2$). The boxplots correspond to AROME-EPS (green), pepi (orange) and pertDpepi (purple). Whiskers reflect the min-max range and boxes the inter-percentile (25%-75%) for the forecasts.

In general, the observed average rainfall rates are contained in the ensemble forecast ranges, except at the end of the rainfall event, on the 15$^{th}$ October between 7:00 and 11:00 UTC, where the ensemble forecasts overestimate the rainfall rates for the 3-hours and 6-hours lead times. This is expected to have limited impact on the anticipation of floods, since it affects only the

end of the rainfall event. The shape and magnitude of the average observed rainfall hyetograph is well anticipated by the 1-hour and 3-hours lead time forecasts, while for the 6-hours lead time forecasts a time-shift of 1 to 2 hours is observed during the whole event. Even if not totally satisfactory, the time-shift remains significantly lower than the lead time of 6-hours. This means that the rainfall forecasts are still helpful to anticipate the actually observed intense rainfall period, even if it is not perfectly positioned in time. This should result in an added value of the hydro-meteorological ensemble forecasts to anticipate the flood rising limbs, for this specific area and rainfall event.

As mentioned in section 2, the objective of this first step is not only to evaluate the overall quality of the rainfall forecast products, but also to define relevant space and time frames (HFA and HFT, respectively), which will help in illustrating the quality and usefulness of these products for flood forecasting. The selected space and time frames must include observed as well as forecast high intensities, but not many areas or time steps with low intensities in both observation and forecasts. These areas are of little interest to flood forecasting, and they may mask the main features of the forecasts when carrying out an event-based forecast quality analysis. Based on the hyetographs presented on Figure 4 and in Appendix A, the HFT was set hereafter from the 14th of October at 20:00 UTC to the 15th at 11:00 UTC. This corresponds to a 15-hour period over which hourly observed and forecast average rainfall intensities on the Aude River basin were larger than 2 mm/h. The choice of this threshold is relatively subjective and just corresponds to a significant average rainfall intensity. Other threshold values could also have been selected. The choice of the HFA will be commented hereafter.

Figure 5 shows the spatial distribution of accumulated observed and forecast rainfall over the 15-hour HFT time window. For the forecast products, the ensemble mean, the 75% percentile and the 95% percentile at 3-hours lead time are plotted. 1-hour and 6-hours lead times are presented in Appendix A (Figures A3 and A4). Note that the forecast pannels do not correspond to rainfall accumulations for one unique run of forecast, but rather to a cumulated representation of the areas affected with high forecast rainfall intensities, for the successive forecasts issued during the event. The Figure 5 shows that the area affected by the high rainfall accumulations (over 280 mm within 15 hours) is, in general, captured by the rainfall forecast products. The added value of the AROME-NWC model, generating locally more intense rainfall rates in the forecasts, can be seen by comparing the AROME-EPS and the pepi fields. The influence of the spatial shifts introduced in the pertDpepi product is also visible, when comparing it with the pepi product. It is also interesting to note that the ensembles means are not able to capture the magnitude of the highest observed rainfall accumulations. Only the tail of the ensemble distributions at each pixel (75% and 95% percentiles) can approach the observed intensities. To produce hydrological forecasts based on a good estimate of the rainfall rates over the high rainfall accumulation period and area, it may be necessary to work based on a high ensemble percentile value (a least the 75% percentile in the present case study; Figure 5.e-g), rather than on the ensemble mean value. However, for these percentiles, the area of high intensities spreads and becomes larger than the area seen in the observed field of rainfall accumulations. This may be attributed to the location errors of some members in the successive runs of forecasts. This behavior is particularly marked in the pertDpepi product, which is probably a combined effect with the spatial shift of the perturbations introduced by this product. Finally, Figure 5 also provides the required information to set the HFA. Even if not entirely hit by the observed heavy precipitation event, the Aude River basin is almost entirely covered with repeated high forecast rainfall intensities during the event, at least for the larger quantiles. This led to the choice of keeping the entire

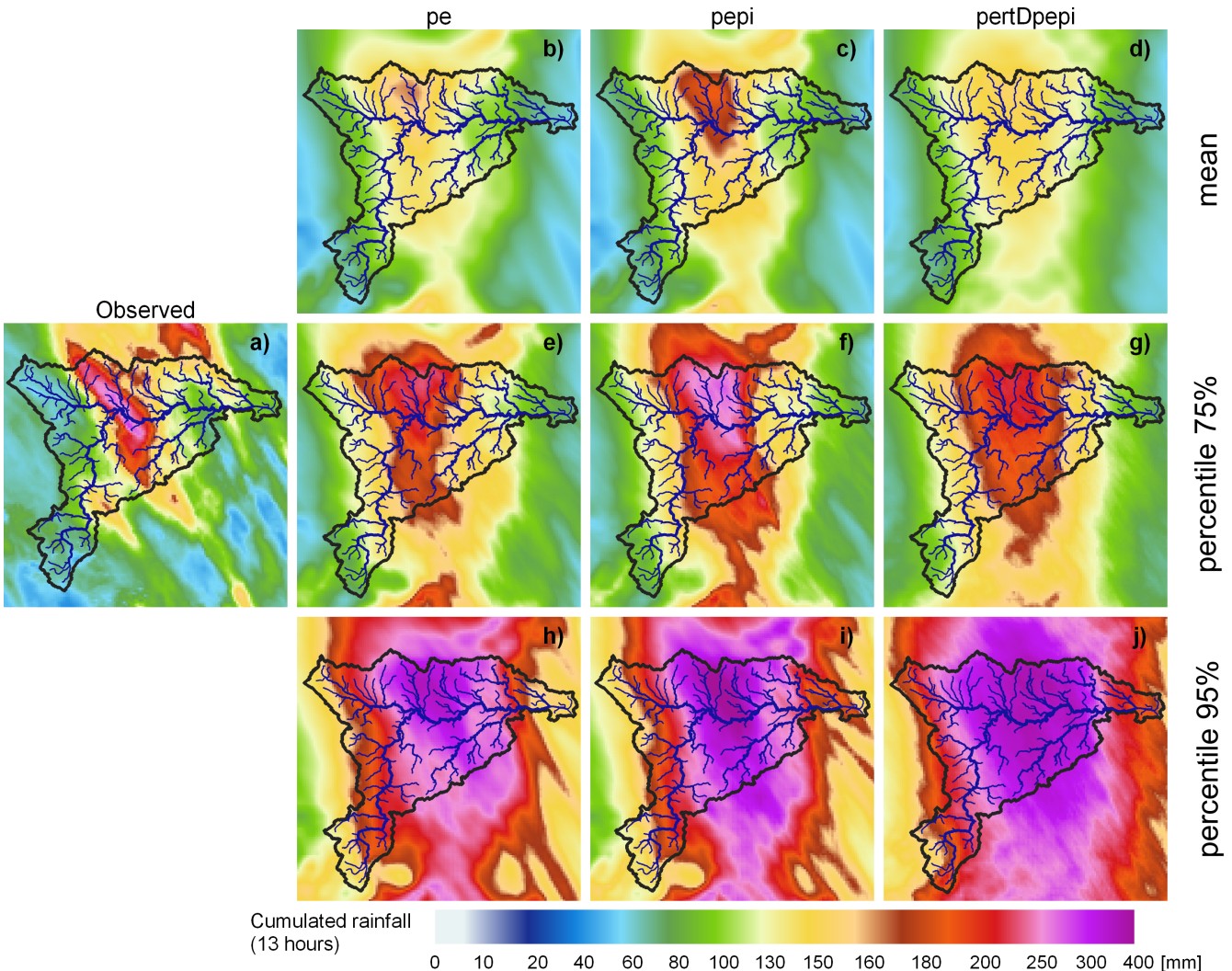

**Figure 5.** Comparison of observed and forecast (3-hour lead time) rainfall amounts over 15 hours (from 14$^{th}$ of October 20:00 UTC to 15$^{th}$ of October 11:00 UTC : a) observed rainfall, b) e) h) AROME-EPS ensemble mean, 75% and 95% percentiles, c) f) i) pepi ensemble mean, 75% and 95% percentiles, d) g) j) pertDpepi ensemble mean, 75% and 95% percentiles

Aude River basin as HFA. Considering this whole area will help in evaluating the risks of false alarms attributed to rainfall forecast location errors when forecasting floods. But as for the HFT, the choice of the exact limits of the HFA remains relatively subjective, and other extents of the HFA could also have been selected.

The overall spread of the rainfall ensemble products can be further analyzed based on the rank diagrams presented on Figure 6. The diagrams are plotted for a lead time of three hours and pull together all the time steps within the HFT and the

pixels of the HFA. Note also that they have been divided (i.e., stratified) into three sub-samples according to the observed hourly rainfall intensities. The rank diagrams for 1-hour and 6-hours lead times are available in the Appendix A (Figures A5 and A6). Overall, the U-shape of the rank histograms indicates a lack of spread of the three ensemble products (under-dispersive ensembles), although still moderate, when it comes to capture the observations within the ensemble spread at the right time and location. The histograms obtained for each sub-sample reveal a shift towards negative (resp. positive) bias, when the highest

(resp. lowest) observed rainfall intensities are considered. This bias appearing in the rank diagrams when data sets are stratified based on observed values, even for perfectly calibrated ensembles, is a well-documented phenomenon (Bellier et al., 2017). Observation-based stratification should therefore be considered with caution in forecast quality evaluation. It is nevertheless interesting to note the influence of stratification in the bias revealed by the rank diagrams. It can be a consequence of the limited spread of the ensemble rainfall forecasts. But it may also reflect the shifts and mismatches in time and space of the ensemble

forecasts, already illustrated on Figures 4 and 5.

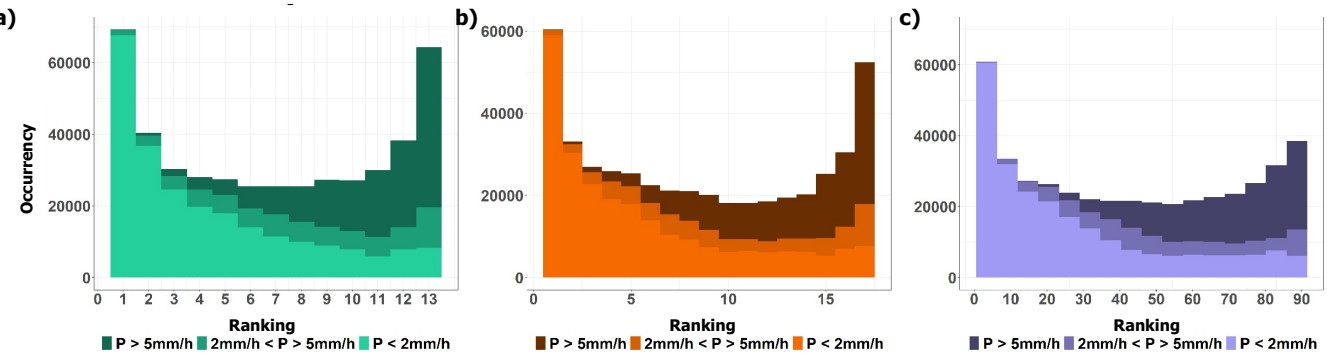

**Figure 6.** Rank diagrams of the three ensemble rainfall forecast products for rainfall rates under 2 mm/h, between 2 and 5 mm/h and above 5 mm/h and for a lead time of three hours : a) AROME-EPS, b) pepi, c) pertDpepi.

## 4.2 Hydrological anticipation capacity

As mentioned in Section 2, the hydrological forecasts are first evaluated on their ability to detect, with anticipation, the exceedance of pre-defined discharge thresholds. The objective is to provide a detailed and comprehensive view of the antici-pation capacity for ungauged streams and main gauged rivers. The criteria and maps presented were computed based on the

CINECAR hydrological model simulations and forecasts, and include the entire sample of 1174 sub-basins outlets. The hy-drological simulations based on observed ANTILOPE J+1 rainfall are considered as the reference simulation (RS). The model is run from the 14$^{th}$ October 07:00 and hourly rainfall accumulations are uniformly disaggregated to fit the 15-min time reso-lution of the model. The forecasts are issued every hour, by using the ANTILOPE J+1 rainfall up to the time of forecast, and one of the 3 rainfall forecast ensembles, or a zero future rainfall scenario (RF0), for the 6 next hours.The hydrological model

is first run for each member of the rainfall forecast to generate a hydrological forecast ensemble. From this ensemble, at each hydrological outlet in the HFA time series of forecast discharges are obtained for several probability thresholds.

The discharge thresholds correspond to the 10-year discharge return period, estimated by the SHYREG method (Aubert et al., 2014). This method provides flood quantiles estimates for all ungauged outlets with a drainage area exceeding 5 $km^2$. The choice of a 10-year return period was motivated by two main reasons: i) this is a discharge level for which significant river

overflows and damages are likely to be observed on many streams ; ii) according to the RS scenario, during the October 2018 flood event about half of the 1174 sub-basins in the HFA were hit by floods with peak discharge exceeding this threshold. It is also important to remind that six runs of forecasts and all lead times are combined at each outlet for the computation of the scores presented in this section (see Appendix B): i.e. one exceedance detected in advance, for at least one of the 6 forecast runs issued just before the exceedance, is considered as a detection (hit). This means that one unique result (either a hit, a miss,

a false alarm or a correct rejection) is obtained for each of the 1174 sub-basins and for each ensemble percentile.

Figure 7 illustrates the resulting ROC curves obtained for the hydrological ensemble forecasts based on the three rainfall products.

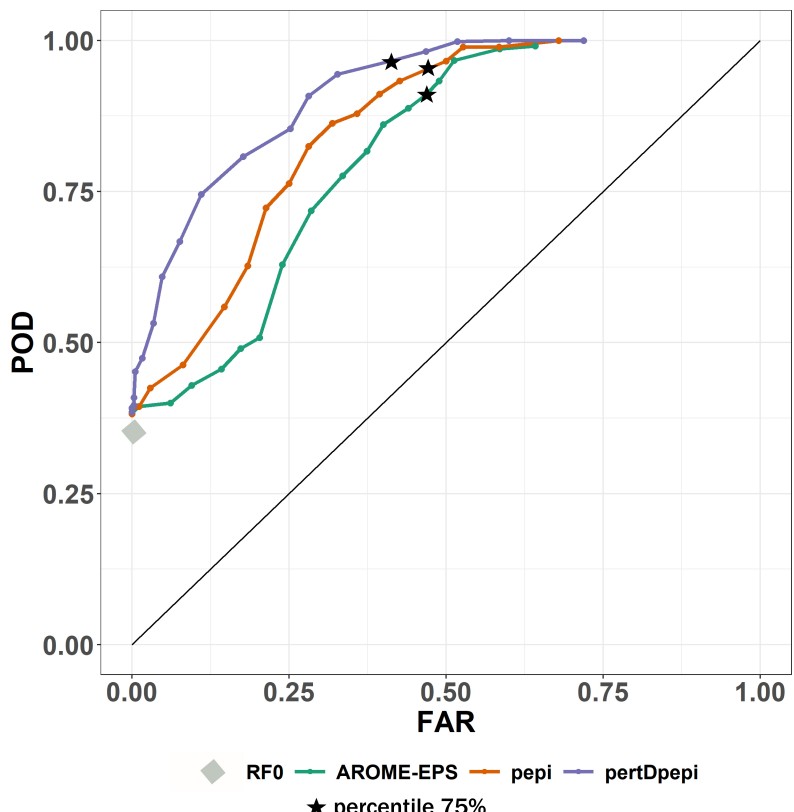

**Figure 7.** ROC curves (Probability Of Detection as a function of False Alarm Rate) summarizing the anticipation of exceedances of the 10-year discharge threshold, for the October 2018 flood event in the Aude River basin. The hydrological forecasts presented are based on AROME-EPS, pepi and pertDpepi rainfall ensemble products, and the CINECAR hydrological model. The black stars indicate the scores obtained for the 75% percentile of the hydrological ensemble forecasts. The points represent the scores obtained for the other percentiles, from 5% to 95%.The gray diamond shows the POD for the RF0 forecasts (zero-future rainfall forecast).

In the figure, the points related to the 75% percentiles of the ensembles are highlighted on each curve. The point obtained for forecasts using a zero rainfall scenario (RFO) is also presented. The added value of the pepi ensemble, compared to the
AROME-EPS ensemble, was already observed in the forecast rainfall analysis (Section 4.1). It is also clearly visible here : for a selected ensemble percentile, both ensembles provide similar false alarm ratios, but the ensemble based on the pepi rainfall product has a higher probability of detection. For a spatial overview of the results, the maps of hits, misses, false alarms, and correct rejections based on the 75% percentiles are presented on Figure 8. In the central area of the Aude River basin, the missed detections for pepi ensemble are lower (26, Figures 8.c), when compared to the AROME-EPS ensemble (50,
Figures 8 b). This is in line with the better capacity of the pepi ensemble forecast product to capture the observed high rainfall accumulations (Figure 5). Overall, the use of the pertDpepi ensemble product leads to a reduction of the number of false alarms

(255, Figure 8.d), when compared to the two other ensemble products (290 for AROME-EPS and 292 for pepi). In practice, the effects of the spatial perturbation introduced by the pertDpepi ensemble differ depending on the area and ensemble percentile considered: it leads to an increase of the area covered by high accumulated rainfalls when considering the highest percentile values of the rainfall ensembles, as clearly visible on Figure 5 for the 95% percentile, but it may also have a smoothing effect of the high rainfall values when considering intermediate ensemble percentiles. This may be an explanation for the reduction of false alarms in some areas for the 75% percentile of the flood forecasts based on the pertDpepi product (Figure 8).

The comparison of the ROC results of the ensemble hydrological forecasts with those of the RF0 scenario - i.e. future rainfall set equal to zero - helps to further evaluate the added value of the rainfall forecast products. All ensemble forecasts lead to an increase of the number of hits (Figure 8) and of the probability of detection (Figure 7). However, this gain is obtained at the cost of an increase in the number of false alarms. When we look at the 75% percentile of the ensembles (black stars), close to 50% of the sub-basins of the Aude River basin where the 10-year discharge return period is not exceeded are associated with a false alarm.

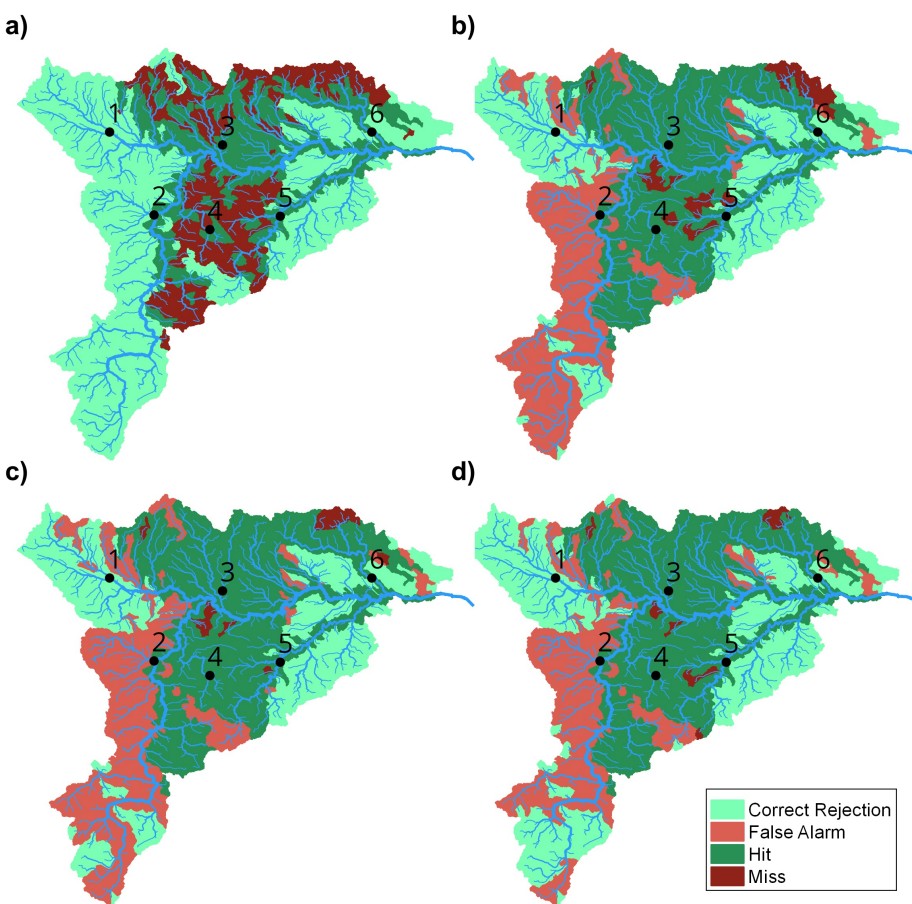

**Figure 8.** Maps illustrating the detailed anticipation results (hits - misses - false alarms - correct rejections) of the 10-year return period discharge threshold, for the hydrological forecasts based on: a) RF0 scenario, b) AROME-EPS ensemble (75% percentile), c) pepi ensemble (75% percentile), d) pertDpepi ensemble (75% percentile). Hits are displayed for anticipation times exceeding 15 min. Black dots represent the outlets where a detailed analysis of the forecasts is proposed in Section 4.3.

Figure 9 shows the distribution of anticipation times, within categories ranging from 15 min to 6 hours, for sub-basins
with hits in Figure 8. Note that the anticipation time is defined here as the difference between the time of exceedance of the discharge threshold by the RS hydrograph, and the time of the first run of forecast that detects the threshold exceedance (see Appendix B). The Figure also shows the number of sub-basins where either no anticipation (i.e., those with misses) or false alarms are observed, and confirms the observations already made on Figure 8: lower number of false alarms for the ensemble forecasts based on the pertDpepi rainfall product, when compared to AROME-EPS and pepi ; and higher number of sub-basins
associated with misses for forecasts of the RF0 reference scenario, as expected. But the results presented on Figure 9 also confirm the added value of using the rainfall ensemble products to increase anticipation times of the exceedance of the 10-year

discharge threshold: the anticipation times remain mostly between 15 min and 1 hour for RF0 forecasts, whereas they exceed 2 hours for more than a half of the sub-basins in hit with the three ensemble rainfall products. The use of the rainfall ensemble forecasts is key in the present case study to extend anticipation times beyond 2 hours, even if this result remains specific to the considered event and cannot be extrapolated to future events.

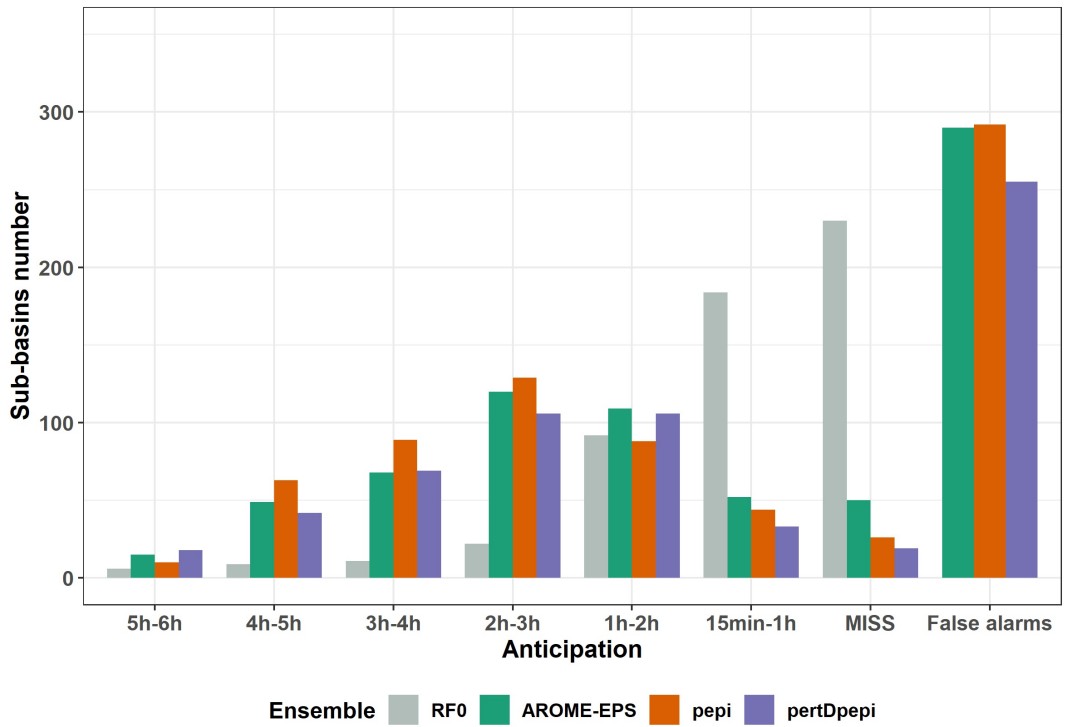

**Figure 9.** Comparison of 10-year threshold anticipation times for the hydrological forecasts based on the RF0 scenario and the three rainfall ensemble products. For the hydrological forecasts ensembles, the 75% percentile is considered.

Figure 9 illustrates the necessary trade-off between gaining hits and larger anticipation times, and increasing false alarms as a consequence, when using ensemble hydrological forecasts to anticipate discharge threshold exceedances. Is it worth when compared to the RF0 forecasts (no future rainfall), where false alarms are avoided, but the number of misses is increased, and the anticipation times are lower, as in the case of the 2018 event? The answer to this question depends on the end-users of the forecasts and their capabilities to respond to flood warnings. Further considerations on the cost and benefits associated with the hits and increased forecasting anticipation times, but also with the misses and false alarms, would be needed to fully address the question. For instance, in our case study, if we summarize the contingency tables used to draw figures 8 and 9 by using the Percent Correct (PC) score (Wilks, 2011) - i.e. the fraction of the N forecasts for which a forecast correctly anticipated the event (hit) or the no-event (correct rejection) - then the conclusion would be that the RF0 forecast outperforms the three

ensemble hydrometeorological forecasts, at least when the decision is based on their 75% percentile. This is because the PC scores are of 80% for RF0, 71% for AROME-EPS, 73% for pepi and 77% for pertDpepi. However, this score gives the same cost or benefit to misses, false alarms, hits and correct rejections. Additionally, it does not consider the added value of the increased anticipation of the threshold exceedances. This shows that an in-depth evaluation of hydrometeorological forecasts could strongly benefit from scores and metrics taking into account the end-user's constraints and needs, as much as possible. This would help to measure the actual cost/benefit balance of using hydrometeorological forecasts in real-time operations. Nevertheless, from the evaluation framework proposed here, we were able to identify that the hydrometeorological ensemble forecasts can offer better anticipation times than the RF0 forecasts, although this anticipation has a cost, i.e. an increase in the number of false alarms, which the user would need to be able to accept.

### 4.3 Detailed hydrological forecasts evaluation at selected outlets

To complement the assessment results, a detailed analysis of the hydrometeorological forecasts is carried out at six selected gauged outlets, covering a variety of situations (see Figure 8 for their location).

The results presented hereafter are based on the GRSDi model simulations and forecasts. The choice of this model is driven by the possibility to compare the forecast hydrographs to both the simulated (RS scenario) and the observed hydrographs at gauging stations, with a hydrological model which was not specifically calibrated to the 2018 flood. This allows us to compare errors related to the rainfall forecasts to those related to the modelling errors. Figures 10 to 15 present, for each of the six selected outlets, three panels of results. The first one on the left shows the spread/skill score of discharge forecasts for all lead times (one to six hours). The spread is the average of the standard deviation of ensemble forecasts. It is divided by the skill, which is the RMSE of the ensemble mean, to obtain the spread/skill score. A value of 1 means that spread and skill are equivalent. The other panels allow us to visually inspect the shape and spread of the forecast hydrographs (mean and quantiles of the ensemble forecasts) for an intermediate lead time of three hours, and to compare the forecasts to the simulated (RS) and observed discharges. Only pepi (middle panel) and pertDpepi (right panel) forecast hydrographs are presented (the AROME-EPS hydrographs are provided in the Appendix C). It should be noted that the "forecast hydrographs" are represented for a fixed three hours lead time, and are not continuous discharges series in time as in the case of the observed and simulated hydrographs. They rather represent what could be predicted by the forecasts issued three hours before each time step $t$ of the flood event.

Outlets 1 and 2 (Figures 10 and 11) correspond to weak hydrological reactions - i.e. the peak discharge of the reference simulation (RS) remains largely below the 10-year discharge threshold in both cases. At outlet 1, the previous step, i.e. the evaluation of discharge threshold anticipation capacity during the flood rising limb, showed a correct rejection, whereas at outlet 2 a false alarm was obtained with the three ensemble forecast products. Outlets 3 and 4 (Figures 12 and 13) correspond to small size watersheds, located in the most intense observed rainfall area. For both outlets, the 10-year threshold exceedance could not be detected with significant anticipation with the RF0 forecast scenario (15 min of anticipation time for outlet 3, and 0 min -i.e. miss - for outlet 4), whereas anticipation times of 3h15min were obtained with the 3 ensemble forecast products. Outlets 5 and 6 (Figures 14 and 15) represent larger watersheds. At these outlets, the 10-year threshold exceedance could

be anticipated with the RF0 forecast scenario (anticipation times of 45 and 60 min respectively). However, these outlets also show an increase of the anticipation capacity with the ensemble rainfall forecasts: anticipation times ranging from 2h45min to 4h45min at outlet 5 depending on the rainfall forecast product, and of 6 hours at outlet 6 for the three products.

For outlet 1, the simulated (RS) hydrograph is overall correctly retrieved by the forecasts (Figure 10). For outlet 2, the forecast hydrographs show a larger dispersion and a significant overestimation, in particular for the upper percentiles (Figure 11). This overestimation is present in the forecasts based on the three ensemble rainfall products. It explains the false alarm observed on Figure 8, when the percentile 75% is considered. It is directly related to the overestimation of rainfall in areas surrounding the actually observed high rainfall accumulations area, as identified on Figure 5. The spread/skill relationship scores (Figures 10 a) and 11 a)) show that the hydrological ensemble forecasts tend to have a higher spread/skill score for the first lead times in all rainfall products. It becomes close to one after 2 to 4 hours of lead time for the hydrological forecasts based on AROME-EPS and pepi rainfall products (i.e., the spread correlates well with the errors). For the forecasts based on pertDpepi, the score tends to stabilize around 1.5 (i.e. there is 50% more spread than ensemble mean skill in these forecasts), highlighting a shortcoming of potentially over - or unnecessary - wide spread in this ensemble product. It might be caused by an excessive spatial shift with respect to the geographical size of the investigated catchment. The larger dispersion of the pertDpepi forecast ensemble is confirmed by the hydrographs, and is in particular visible for the 5%-95% percentiles inter-quantile range, while the range defined by the 25% and 75% percentiles show limited difference with the respective inter-quantile range in the pepi ensemble product. This confirms the tendency of pertDpepi rainfall product to modify differently the extreme percentiles and the intermediate ones when compared to pepi. In practice, the over-dispersion observed for extreme percentiles could have limited the performance of the pertDpepi ensemble if these extreme percentiles had been used by a user. This was also visible on Figure 7: a large increase of false alarm rates with the pertDpepi ensemble product in the upper part of the ROC curve.

Lastly, the comparison of observed and simulated hydrographs shows a different performance of the hydrological model between outlets 1 and 2. The modelling errors are limited in the case of outlet 1, whereas the observed discharges are overestimated by the model in the case of outlet 2. In this second case, the comparison between observed, simulated and forecast hydrographs shows that a large part of the total forecast error can be attributed to modelling errors.

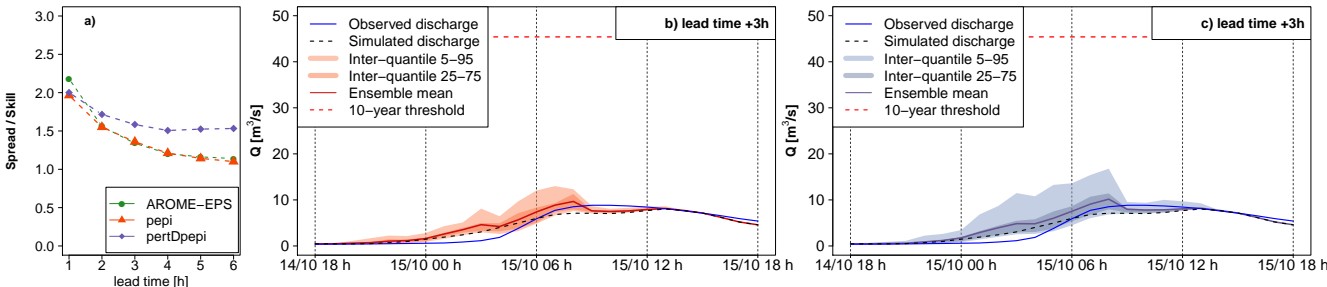

**Figure 10.** Detailed hydrological forecasts evaluation at outlet 1 (216 km$^2$): a) spread/skill relationship, b) hydrographs of 3-hours lead time forecasts with pepi, c) hydrographs of 3-hours lead time forecasts with pertDpepi. In blue, the observed hydrograph at the gauging station.

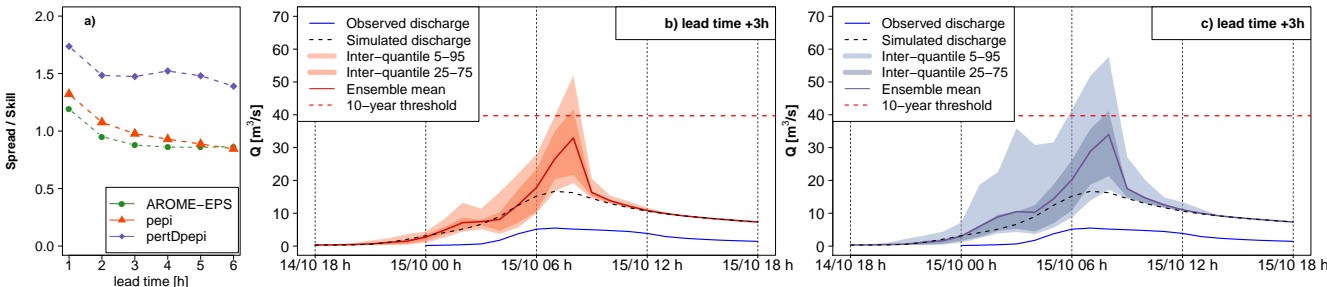

**Figure 11.** Detailed hydrological forecasts evaluation at outlet 2 (197 km$^2$): a) spread/skill relationship, b) hydrographs of 3-hours lead time forecasts with pepi, c) hydrographs of 3-hours lead time forecasts with pertDpepi. In blue, the observed hydrograph at the gauging stations.

In the cases of outlets 3 and 4 (small size watersheds in the most intense observed rainfall area - Figures 12 and 13), the RS hydrograph largely exceeds the 10-year discharge threshold at the very beginning of the flood rising limb. The 3-hour forecasts
also exceed the thresholds and anticipate well the initial increase of the river discharges. This results in the hits presented on Figure 8, and in the good anticipation times, exceeding 3-hours for both outlets. A forecast user could be satisfied with such information (anticipation of threshold exceedance) to start a flood response.

For both outlets, the forecast hydrographs show a delay in the flood rising limb and flood peak, comparatively to the simulated hydrographs. This means that shortcomings are present in the rainfall forecast products in terms of the dynamics of the time
evolution of the rainfall event. These shortcomings could not be directly captured in the previous evaluation scores, but might be an important feature for forecast users. The flood rising limb is overall better represented in the case of outlet 3. At outlet 4, the 3-hour simulated hydrograph presents a large delay comparatively to the simulated hydrographs (3-hours between observed and 75% percentile 3-hour forecast peak discharges). This delay comes with a significant underestimation of the flood peak magnitude by the ensemble mean and 75% percentile, although the 95% percentile is closer to the simulated discharge peaks.
This goes along with the general tendency of the rainfall ensemble means to underestimate the higher rainfall accumulations (Figures 5 and 6).

For both outlets, the spread/skill relationship scores reflect the influence of the temporal shifts between the simulated and forecast hydrographs. At outlet 3, where forecasts resemble better the simulated hydrographs, the skill-spread relationship is close to one for the ensembles based on AROME-EPS and pertDpepi, which can be interpreted as a sufficient variance of the
forecast spread to cover the errors. The ensemble based on pepi, on the contrary, has 1.5 times more spread than skill at 3-hours of lead time. At outlet 4, the disparity between forecast and simulated hydrographs is too high and, the large forecast spread, as seen in the hydrographs, is not enough to cover the timing errors. At both outlets, and comparatively to outlets 1 and 2, we also note that the spread-skill relationship score almost does not change with forecast lead time, except for pepi-based forecast at the longer lead times.
The comparison of observed hydrographs with simulated and forecast hydrographs also provides interesting information about the relative importance of modelling and forecasting errors. At outlet 3, the observed and simulated hydrographs have a different shape, whereas the simulated and forecast hydrographs appear similar despite a time delay. At outlet 4, observed and

simulated discharges resemble (despite a time delay), but both are very different from the forecast hydrograph. This indicates that modelling errors are a major source of uncertainty at outlet 3 (if we consider that observations represent well the true flow values), comparatively to rainfall forecast errors, while the opposite is seen at outlet 4.

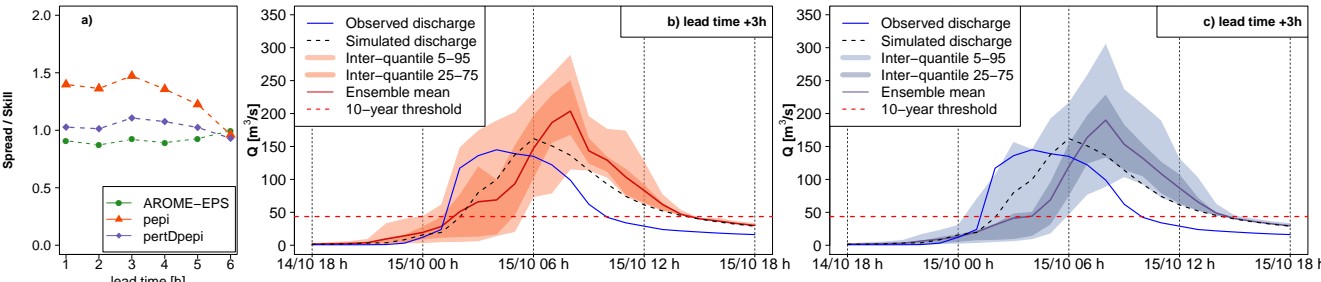

**Figure 12.** Detailed hydrological forecasts evaluation at outlet 3 (85 km$^2$): a) spread/skill relationship, b) hydrographs of 3-hours lead time forecasts with pepi, c) hydrographs of 3-hours lead time forecasts with pertDpepi. In blue, the observed hydrograph at the gauging station.

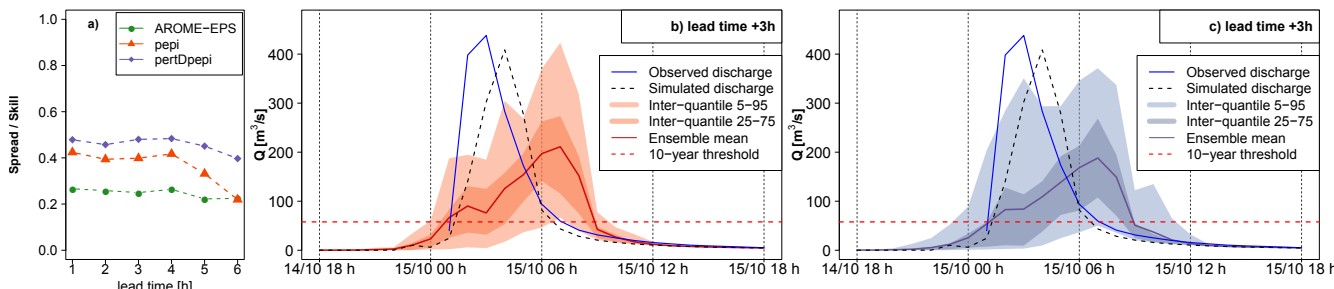

**Figure 13.** Detailed hydrological forecasts evaluation at outlet 4 (173 km$^2$): a) spread/skill relationship, b) hydrographs of 3-hours lead time forecasts with pepi, c) hydrographs of 3-hours lead time forecasts with pertDpepi. In blue, the observed hydrograph at the gauging station.

Finally, Figures 14 and 15 present the forecast results obtained for outlets 5 and 6, corresponding to outlets at which significant anticipation is possible even without rainfall forecasts (RF0 scenario, see Figure 8.a). It can be seen that the hydrological reactions at these outlets are important, as the 10-year discharge threshold is exceeded by the RS hydrograph in the intermediate phase of the flood rising limb, although not as much as for the outlets 3 and 4, where the discharges largely exceeded this threshold. This is explained by the location of these catchments, at the limits of the most intense observed rainfall area. Overall, the forecast hydrographs are less dispersed than in the preceding cases, and the differences between the three rainfall forecast products is less evident, with only a slightly larger spread for the forecasts based on the pertDpepi ensembles. This lower difference between products can be a direct result of the larger influence of flood propagation in the hydrological forecast, limiting the influence of rainfall forecast uncertainties in comparison with the former cases. However, some shortcomings of the rainfall forecasts are still visible in both cases. At outlet 5, the flood rising limb of the simulated hydrograph is well captured by the 25%-75% inter-quantile range of the 3-hour ahead forecasts, but the forecasts display a delay of about 2 hours

in terms of timing of the peak discharge. This time shift of forecasts may explain the misses that appear in the upstream parts of the catchment (Figure 8). At outlet 6, the forecasts tend to slightly overestimate the peak of the simulated hydrograph, although the 5% quantiles remain close to the simulated peak.

The influence of modelling errors also clearly appears here. At outlet 5, simulated and observed hydrographs are close to each other during the main peak, but the first peak of the observed hydrograph is not represented in the simulation. At outlet 6, the simulated discharges follow the same pattern as the observed discharges, but the magnitude of the peak discharge is underestimated by the model. In both cases, the modelling errors are partially compensated in the forecasts, under the condition that a user looks at the higher quantiles of the ensemble forecasts, as the 5%-95% inter-quantile range of the 3-hour

ahead forecasts gets closer to the observed peak discharges. Overall, these two outlets mainly illustrate how the evaluation of hydrological forecasts can be influenced also by the ability of the rainfall-runoff model to correctly represent the flood dynamics, when the evaluation is achieved based on actual flow observations.

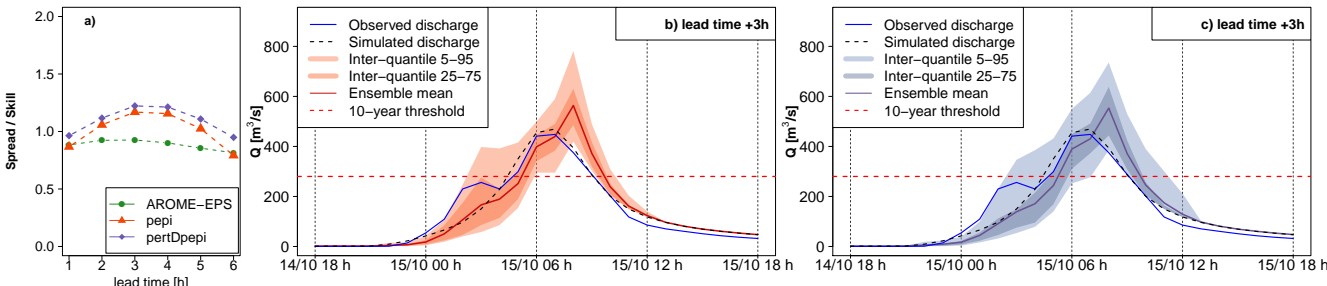

**Figure 14.** Detailed hydrological forecasts evaluation at outlet 5 (263 km$^2$): a) spread/skill relationship, b) hydrographs of 3-hours lead time forecasts with pepi, c) hydrographs of 3-hours lead time forecasts with pertDpepi. In blue, the observed hydrograph at the gauging station.

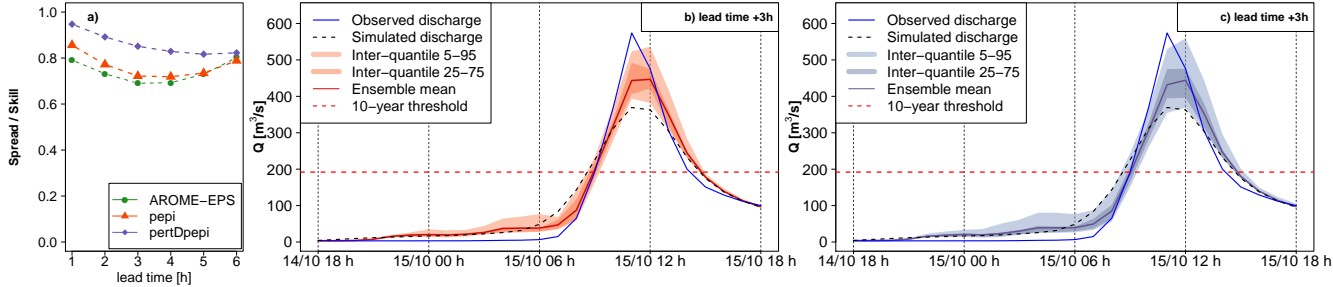

**Figure 15.** Detailed hydrological forecasts evaluation at outlet 6 (257 km$^2$): a) spread/skill relationship, b) hydrographs of 3-hours lead time forecasts with pepi, c) hydrographs of 3-hours lead time forecasts with pertDpepi. In blue, the observed hydrograph at the gauging station.

## 5   Discussion

### 5.1   Added value and limitations of the implemented evaluation framework

After an initial analysis of the performance of the three rainfall forecast products, the evaluation procedure proposed in this paper focused on the capacity of the hydrological forecasts to anticipate the exceedance of selected discharge thresholds. Several authors have suggested using thresholds and contingency tables to perform a robust regional evaluation of hydrological ensemble forecasts (Silvestro and Rebora, 2012; Anderson et al., 2019; Sayama et al., 2020). In this study, a step further is proposed towards (i) focusing only on the rising limb phase of the flood hydrograph - i.e. the most critical phase in terms of

anticipation needs for preparedness and emergency response, as already suggested by Anderson et al. (2019), and (ii) splitting the analysis into a geographical point of view (discharge threshold anticipation maps) and a temporal point of view (histogram of anticipation times). The illustrative example of the October 2018 flood in the Aude River basin in France showed that this approach provides a synthetic and meaningful information about the possible gains in anticipation compared with the RF0 reference forecast where future rainfalls are considered as null. It also informs about the relative performance of the evaluated

rainfall forecast ensemble products, even if the products appeared relatively similar at first sight in the example showed here. This analysis can be easily adapted to other flash flood events with different characteristics, or even to other types of floods. For instance, different discharge thresholds can be set to better distinguish the areas where the main flood reactions were observed, a larger threshold can be applied to the anticipation time to decide for a hit (a minimum threshold of 15 minutes of anticipation time was used in this study), and different ensemble percentiles can be selected to better illustrate the role that the probability

threshold plays when evaluating different forecast products.

The threshold anticipation maps are also helpful for the selection of outlets where hydrographs should be analyzed in more details in the last phase of the evaluation. They avoid focusing only on outlets where large anticipation is possible without rainfall forecasts (RF0 scenario). The differences between the hydrological forecasts based on rainfall ensemble products may be more difficult to identify at these outlets, because of the large influence of flood propagation in the hydrological

forecasts. The threshold anticipation maps also allow selecting outlets corresponding to a variety of situations (i.e. hit/miss/false alarm/correct rejection), which may help in bringing up details about the complex features of the hydrological ensemble forecasts.

If the proposed approach has several advantages, it also relies on several methodological choices which could be adapted or improved. It can be noted that only the runs of forecasts covering the most critical phase of the event (i.e. the time of the

threshold exceedance, or the maximum of the RS hydrograph) were considered to build the contingency tables. The results obtained could differ if other runs of forecasts and/or other phases of the event were considered. Particularly, in case of events with multiple flood peaks, the fact that the method focuses only on the first threshold exceedance can be seen as a limitation. In such a situation, each rising phase of the flood event could be examined separately, even if in some cases the different phases of the floods may be difficult to separate. An alternative could be to analyze the anticipation of threshold exceedances for each

run of forecast during the event, independently of the times the thresholds are exceeded for the RS hydrographs. Providing an evaluation for multiple flood events, or for multiple thresholds for the same flood, may also be interesting complements to the

results presented here. This may nevertheless cause some difficulties in setting the HFT and HFA, which are event specific. The HFA may also require to be adapted to the considered threshold. To avoid changing the limits of the HFA, an option could be to use a score that does not account for Correct Rejections, and therefore would be less sensitive to the extent of the

HFA, such as the Critical Success Index. In addition, the procedure used for the computation of the contingency tables and the anticipation times can be considered as optimistic, at least in some situations. At first, as explained by Richardson et al., 2020, having a sequence of several consistent forecasts is often a desired quality in forecasting systems. Specific tests of the consistency of forecasts have been proposed (Ehret and Zehe, 2011; Pappenberger et al., 2011). In this study, only the first time a forecast exceeds the discharge threshold value was considered to conclude to a hit (or a false alarm) and to compute the

anticipation time, without considering the consistency of several successive forecasts. This can be seen pertinent when dealing with flash floods, when it is often not possible to wait for forecast consistency in real-time to activate flood response actions. However, it could also be interesting to use the same method and to consider the number of runs correctly anticipating the threshold exceedances (or non-exceedances) to fill the contingency table. Moreover, since two of the three rainfall products used in this study are still experimental (pepi and pertDpepi), and are not yet included in the real time production workflow of

Météo-France, it was not possible to integrate the actual delivery times of the forecasts in this study. To obtain a more realistic view of the actual anticipation capacity, the delivery times of the different rainfall forecast products need to be integrated in the computation of the contingency tables and anticipation times, as proposed by Lovat et al. (2022). This means that the hydrological ensemble forecasts should be considered only from the delivery time of the rainfall forecast products used as input, instead of from the beginning of the sequence of time covered by the rainfall forecasts. This integration would probably

lead to a reduction of the number of hits and of the anticipation times obtained with the hydrological ensemble forecasts in this study, whereas it would not affect the results obtained for the RF0 forecast, where no future rainfall scenario is used.

Lastly, it should also be noted that the last step of the evaluation analysis proposed in this study, which is based on the whole hydrographs and spread and skill scores, remains essential to bring a detailed view of the strengths and limits of the different ensemble products evaluated. Particularly, the shortcomings of an ensemble with a too large dispersion (which is the case of the

pertDpepi product) and the impact of temporal shifts on the quality of the forecasts would have been more difficult to identify without this last step of the analysis. The analysis of hydrographs also provides detailed explanations for the hits/misses/false alarms/correct rejections observed on the discharge threshold anticipation maps.

### 5.2 Performance of the three ensemble forecast products

The contribution of the rainfall forecast ensemble products to the anticipation of discharge thresholds, with respect to the

scenario where no future rainfall forecast is available (RF0 scenario) was clearly illustrated here. This contribution lies mainly in the high reduction of the number of misses on the threshold anticipation maps (Figure 8), and in a significant gain in the anticipation times for almost all sub-basins (Figure 9). It also depends on the ensemble probability (or percentile) that is considered to define a forecast event (in our case, the 75% percentile). The use of ensemble forecasts was essential to obtain anticipation times greater than one hour when a hit was observed, even if this positive effect was counterbalanced by the

occurrence of a high number of false alarms in other areas of the catchment. This implies that, in order to conclude about

the added value of ensemble forecasts, it is necessary to consider the balance between the actual gains and costs for a given user of the forecasts (Verkade and Werner, 2011). Hydrometeorological forecasting is no exception to the well-known decision principle: the larger the uncertainties (i.e. the lower the sharpness) affecting the predictions of the variable(s) on which the decision is based, the larger the safety margin will be to guard against the risk of unwanted consequences. Particularly in the case of flash floods, the user is often interested in reducing misses and increasing hits and anticipation times. This means that users need to consider how much of a false alarm rate they can handle, and how much risk they can take (i.e., what percentile of the ensemble distribution they should consider). These are, however, difficult questions to answer. More flood event evaluations are needed to enhance our understanding of these trade-offs.

In the present case study, the pepi rainfall ensemble product was built from the AROME-EPS product by adding one to six additional members (depending on the lead time). This strategy had globally a positive influence on the quality of the ensemble forecasts, with a significant decrease in the number of misses for an equivalent number of false alarms (Figure 9). This can be directly related to the increased reliability of the pepi forecasts compared to the AROME-EPS forecasts, and therefore the better capture of the high rainfall accumulations periods (Figure 4) and extension (Figure 5) when considering the 75% percentile. Finally, in the case of this event, the pepi product provides an added value for the characterization of the main intense rain cell, without a significant degradation of the ensemble performance in the surrounding areas, where the rainfall accumulations were less intense. However, this overall improvement does not result in a significant increase of the anticipation times obtained with the pepi product compared to the AROME-EPS product (Figure 9).

Adding spatial perturbations in the four cardinal directions to the members of the pepi ensemble led to an increase of the number of members in the pertDpepi ensemble product, which varies from 65 to 90 members depending on the lead time. Overall, this resulted in a larger dispersion of the ensemble, particularly visible on the extreme percentiles (5%-95%), and led to positive as well as negative effects, both visible in the evaluation results. The high rainfall accumulation period is still well captured (Figure 4), but the maximum rainfall area is not as well localized, comparatively to the observations, as for the pepi ensemble (Figure 5). Regarding the discharge threshold anticipation maps (Figures 8 c) and d)), the reduction of the number of false alarms with pertDpepi is mainly due to the choice of the 75% percentile, which is not modified in the same way as higher percentiles when compared to the pepi ensemble. This is confirmed by the hydrographs analyses (Figures 10 to 15), which overall confirm the over-dispersion of pertDpepi for higher percentiles of the ensemble distribution.

Moreover, the anticipation times are slightly reduced with pertDpepi when compared to the two other ensemble products (Figure 9): 221 (over 1174) sub-basins show an anticipation time of more than two hours, against 245 for AROME-EPS and 259 for pepi. Again, the detailed hydrographs analyses (Figures 10 to 15) confirm that pertDpepi does not efficiently compensate the time shifts in the representation of hydrographs. The addition of spatial perturbations leads to a significant increase in the number of members, which significantly increases the computing times with the pertDpepi rainfall ensemble product (up to 72 additional members compared to pepi), but there is no clear benefit in adding these members in the case evaluated here.

## 6 Conclusions

In this paper, we proposed and tested a methodological framework for the evaluation of short-range flash-flood hydrome-
teorological ensemble forecasts at the event scale. Focusing on the evaluation of a single flood event is necessary because of
i) the relatively low frequency of flash floods occurrences, and ii) the limited length of available re-forecast periods for newly
developed ensemble rainfall forecast products based on convection permitting NWP models. In order to enhance the value of a
single-event evaluation, the point of view of an end-user of the forecasts is adopted in the proposed methodological approach.
This approach combines different steps. Firstly, it comprises a preliminary analysis of rainfall observations and forecasts, which
provides a first overview of the quality of the ensemble rainfall products and information needed to select the most relevant
ensemble percentile and spatial and temporal scales for the analysis. Secondly, a hydrological forecast analysis evaluates the
detection of discharge threshold exceedances and the corresponding anticipation times. It considers the flood rising limb of the
hydrographs, and illustrates the gains in anticipation comparatively to a reference scenario where zero rainfall is forecast for
the future. Thirdly, a detailed analysis of forecast hydrographs and spread/skill scores at outlets selected based on the former
step is carried out. It aims to better illustrate the skills and the limits of the evaluated hydrological ensembles. This methodol-
ogy was tested and illustrated on the major flash-flood event that hit the Aude River basin in South-Eastern France in October
2018. Three ensemble rainfall forecast products recently developed by Météo-France were compared, i.e. the AROME-EPS
ensemble (12 members), the pepi ensemble combining AROME-EPS and AROME-NWC members (13 to 18 members), and
the pertDpepi ensemble adding spatial perturbations to the pepi ensemble (65 to 90 members).

The study illustrated the multifaceted and complex issue of evaluating single flood events. It confirmed the interest of
the proposed evaluation procedure to help forecast users to efficiently define beforehand the space and time windows to be
used for the forecast performance evaluation. This was achieved by comparing the observed and forecast rainfall hyetographs
and accumulations over the entire catchment area (step 1). The analysis of discharge threshold exceedance detection (step
2) contributed to illustrate the geographical features of the flood event and the anticipation capacities offered by each of the
rainfall ensemble products. The comparison with a zero rainfall future scenario confirmed the gains in anticipation offered
by the ensembles. From the threshold anticipation maps, it was also possible to select outlets that correspond to a variety of
situations of interest to a forecast user (hits, misses, false alarms, correct rejections). These outlets could then be investigated in
more details, based on observed, simulated (with observed rainfall as input to the hydrological model) and forecast hydrographs
(step 3). This last phase illustrated further strengths and limitations of the different rainfall forecast products, including the shifts
in timing that affect forecast skill and the main sources of forecast errors (rainfall forecasts or hydrological model).

The evaluation finally led to balanced conclusions about the three products of rainfall ensemble forecasts. These products
clearly enhance the anticipation times when compared to a zero-rainfall future scenario, but tend to overestimate the spatial ex-
tent of the area of the highest rainfall accumulations. This results in a higher risk of overestimating flood threshold exceedances
in the surroundings of this area (i.e., an increase of false alarms). The actual added value of the evaluated forecasts for an end-
user therefore depends on the relative benefits of the increased anticipation and costs of the false alarms. However, the results
also showed a hierarchy between the three evaluated products, and a clear added value of the pepi ensemble product could be

identified. According to the significant time shifts of forecasts hydrographs in several sub-basins, which seem to significantly limit the anticipation times, the introduction of temporal perturbations could probably be an interesting alternative to be tested

Finally, even if evaluating ensemble hydrological forecasts based on a single flood event remains a very challenging issue
due to the limited statistical representation of the available data, single event evaluations are needed and important to advance operational flood forecasting systems. The evaluation framework proposed in this study could be helpful in drawing rapid and meaningful analyses about the interest of newly developed rainfall ensemble forecast approaches.

*Data availability*

*Author contributions* The initial idea was proposed by MCN, OP, DP and MHR. The initial version of the paper was written
by MCN and DP with a general contribution of EG, MHR, and OP, and specific contributions of HM and FB for the description of the AROME-based short-range rainfall ensemble forecast products, and of PN for the rainfall-runoff modelling. AF generated the pepi and pertDpepi short-range rainfall ensemble forecast products. MCN performed the Cinecar rainfall-runoff modelling, and used the calibration results obtained by PN and OP for the Aude 2018 event. DP performed the GRSDi rainfall-runoff modelling.

*Competing interests* The authors declare that they have no conflict of interest.

*Acknowledgements* Peak discharge data for the Aude 2018 flood event were obtained as part of the HyMeX research program (http://hymex.org), with financial support from the MISTRALS program of the CNRS and the Ministry of Ecological and Solidarity Transition (DGPR/SCHAPI). Rainfall observation and forecasts data were provided by Météo-France.

*Financial support* This research is part of the PICS research project (https://pics.ifsttar.fr) and has been supported by the
690 Agence Nationale de la Recherche (grant no. ANR-17-CE03-0011).

**Appendix A: Comparison of observed rainfall and ensemble forecasts for 1-hour and 6-hour lead times (hourly rainfall rates, maps of rainfall amounts, and rank diagrams**

**A1 Temporal evolution of observed hourly rainfall rates and ensemble forecasts for 1-hour and 6-hours lead times**

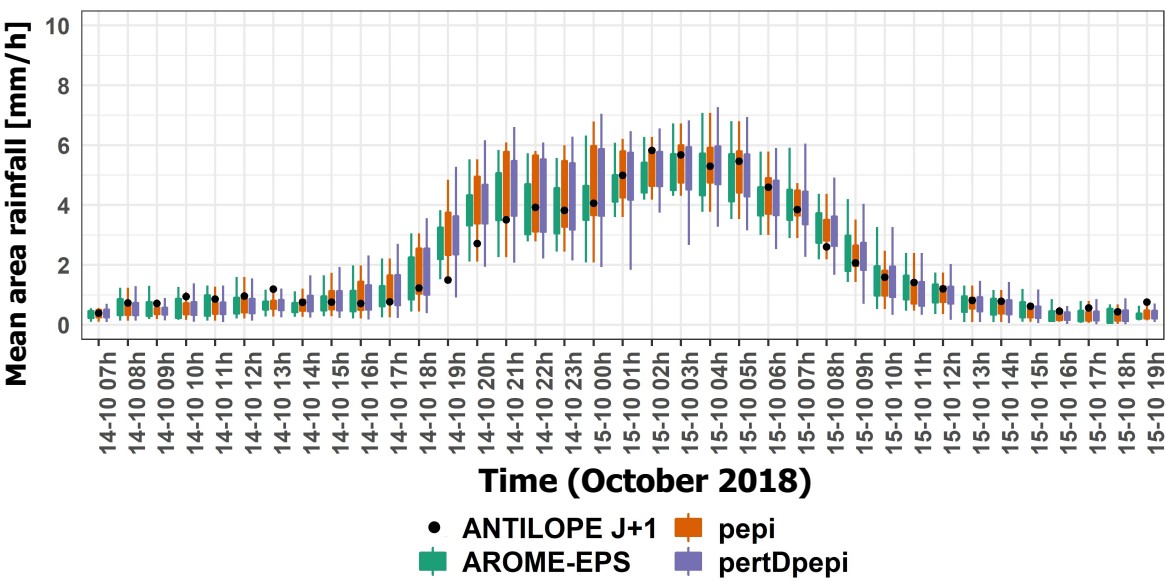

**Figure A1.** Temporal evolution of observed hourly rainfall rates (black dots), and 1-hour lead time ensemble forecasts (boxplots), during the October 2018 flood event. Rainfall rates are averaged over the Aude River basin (6074 km$^2$). The boxplots correspond to AROME-EPS (green), pepi (orange) and pertDpepi (purple). Whiskers reflect the min-max range and boxes the inter-percentile (25%-75%) for the forecasts.

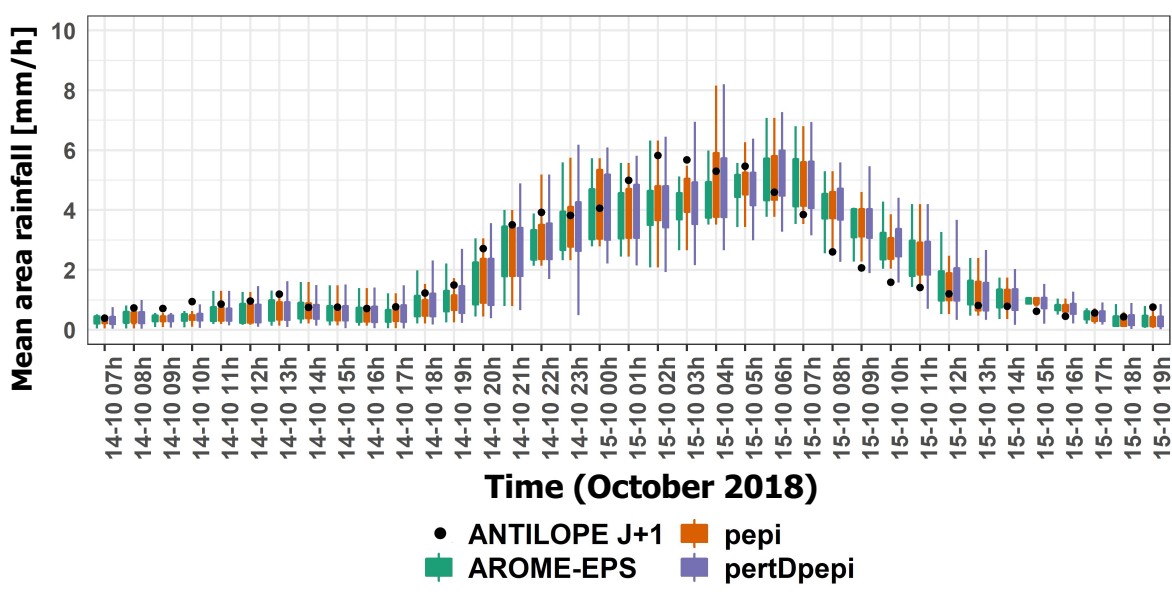

**Figure A2.** Temporal evolution of observed hourly rainfall rates (black dots), and 6-hours lead time ensemble forecasts (boxplots), during the October 2018 flood event. Rainfall rates are averaged over the Aude River basin (6074 km$^2$). The boxplots correspond to AROME-EPS (green), pepi (orange) and pertDpepi (purple). Whiskers reflect the min-max range and boxes the inter-percentile (25%-75%) for the forecasts.

**A2    Comparison of observed and forecast rainfall amounts over 15 hours for the 1-hour and 6-hours lead times**

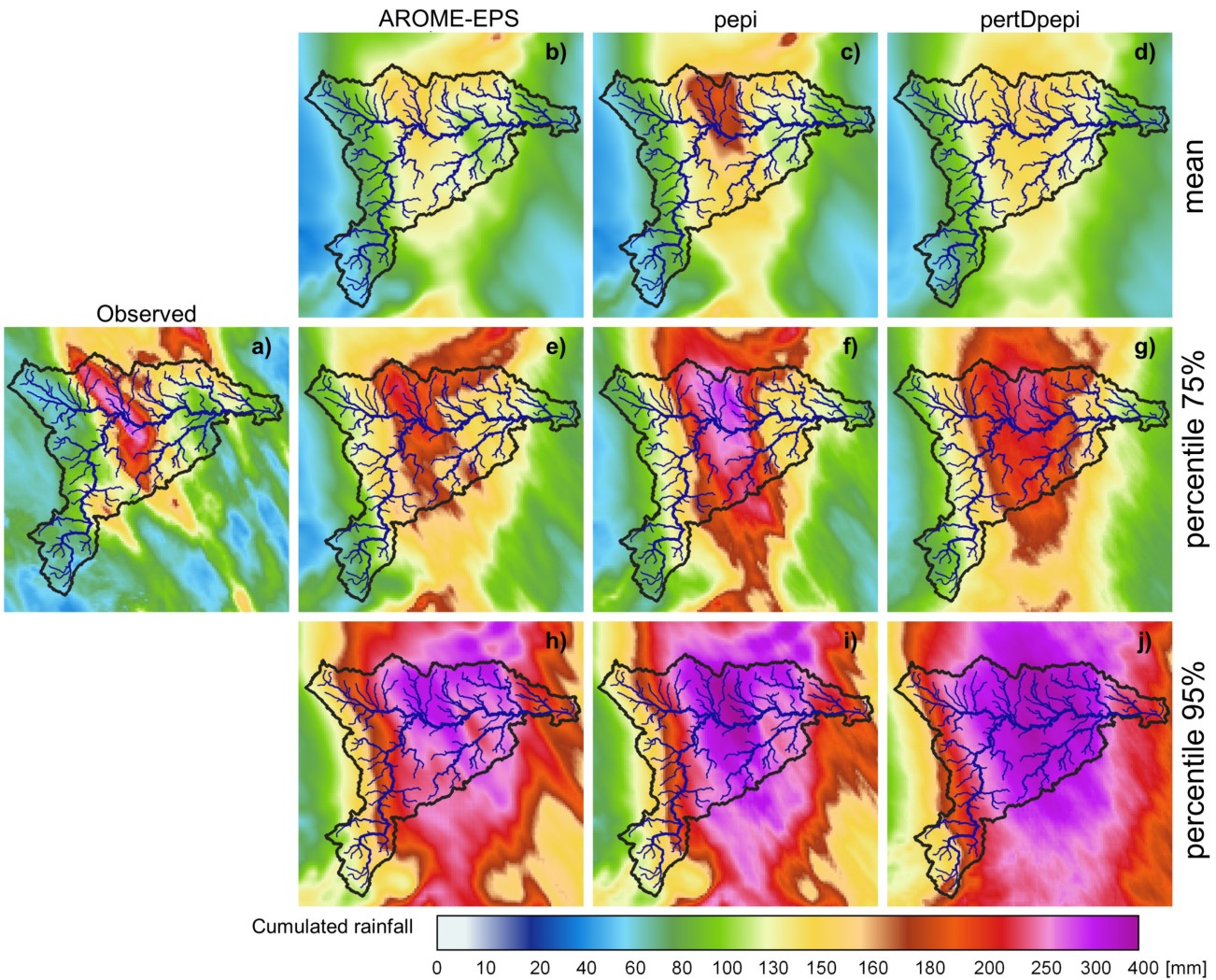

**Figure A3.** Comparison of observed and forecast (1-hour lead time) rainfall amounts over 15 hours (from 14$^{th}$ of October 20:00 UTC to 15$^{th}$ of October 11:00 UTC) : a) observed rainfall, b) e) h) AROME-EPS ensemble mean, 75% and 95% percentiles, c) f) i) pepi ensemble mean, 75% and 95% percentiles, d) g) j) pertDpepi ensemble mean, 75% and 95% percentiles

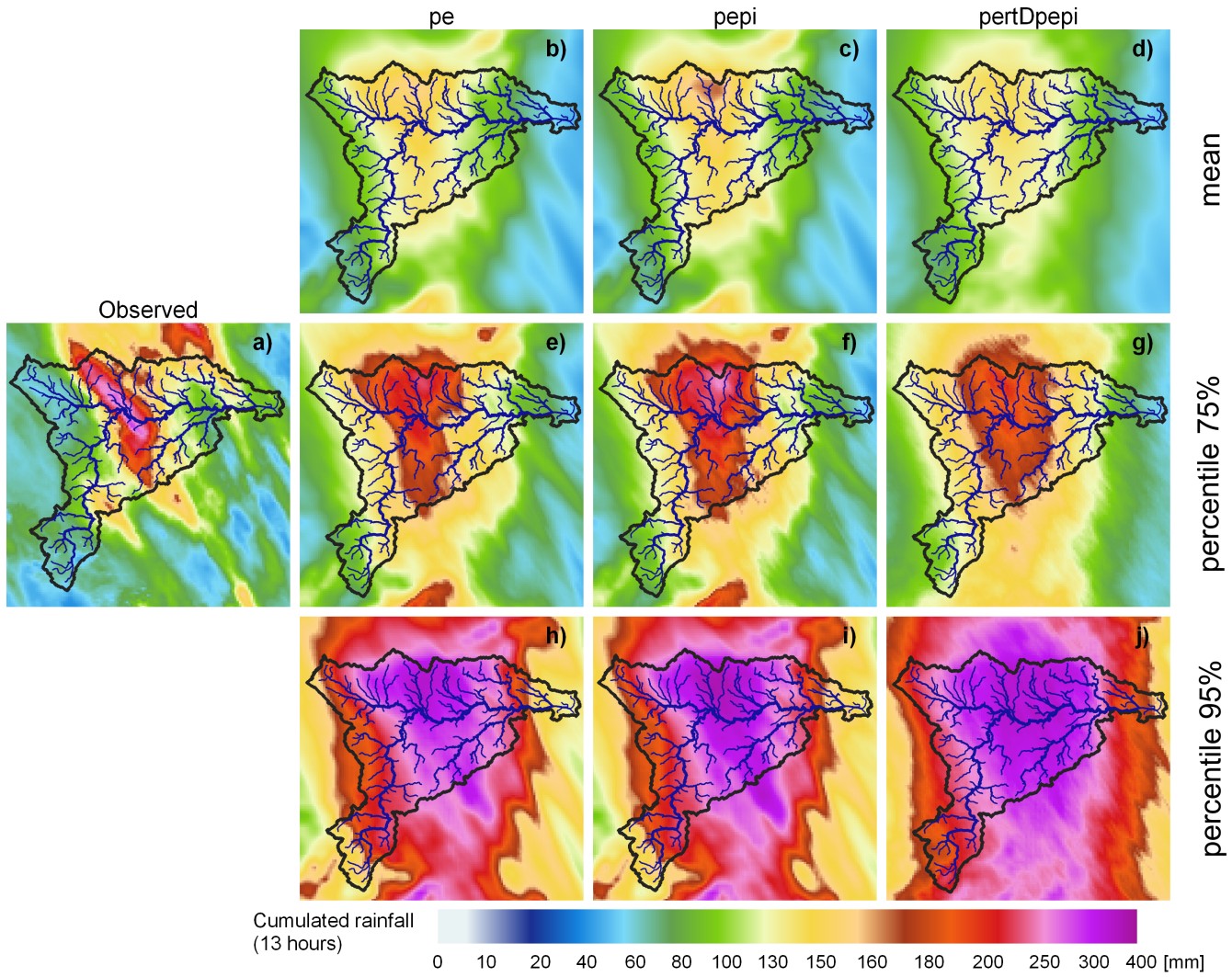

**Figure A4.** Comparison of observed and forecast (six hours lead time) rainfall amounts over 15 hours (from 14[th] of October 20:00 UTC to 15[th] of October 11:00 UTC) : a) observed rainfall, b) e) h) AROME-EPS ensemble mean, 75% and 95% percentiles, c) f) i) pepi ensemble mean, 75% and 95% percentiles, d) g) j) pertDpepi ensemble mean, 75% and 95% percentiles

 **A3 Rank diagrams for the 1-hour and 6-hours lead times rainfall forecasts**

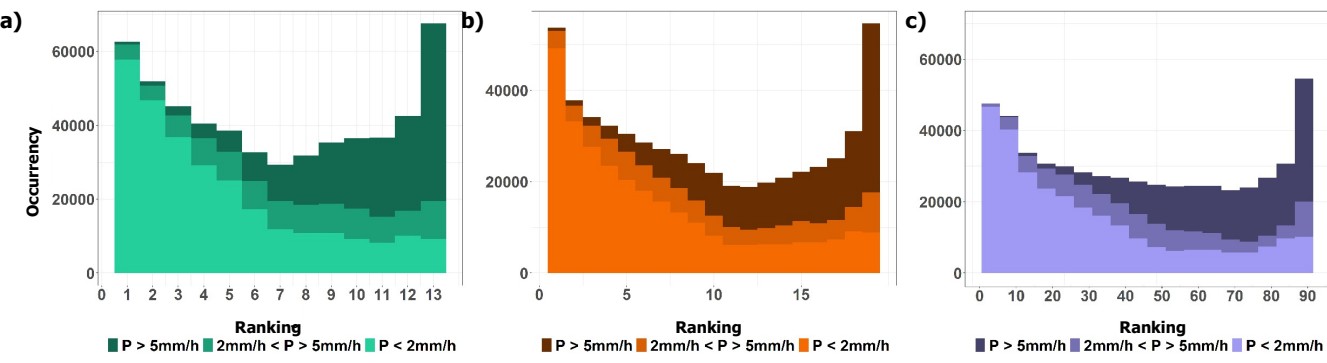

**Figure A5.** Rank diagrams of the three ensemble rainfall forecast products for rainfall rates under 2 mm/h, between 2 and 5 mm/h and above 5 mm/h and for a lead time of one hour : a) AROME-EPS, b) pepi, c) pertDpepi

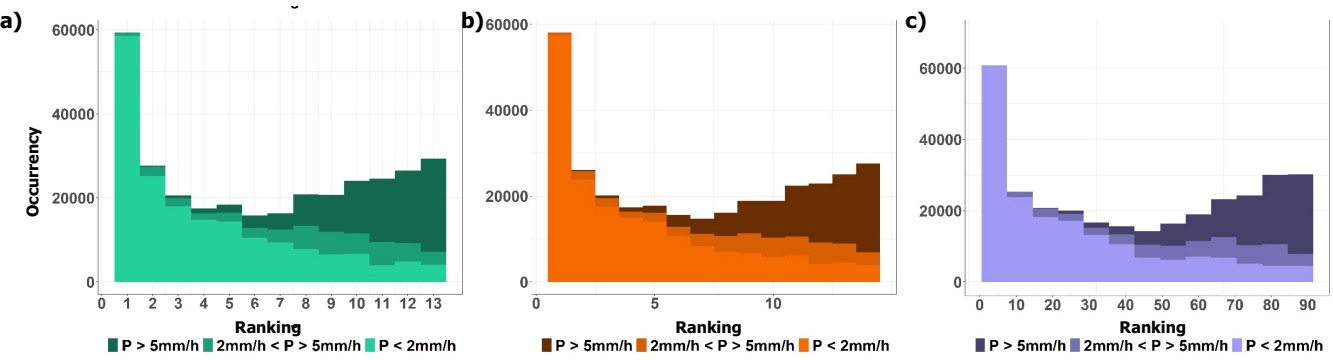

**Figure A6.** Rank diagrams of the three ensemble rainfall forecast products for rainfall rates under 2 mm/h, between 2 and 5 mm/h and above 5 mm/h and for a lead time of six hours : a) AROME-EPS, b) pepi, c) pertDpepi

**Appendix B:  Methodology used for the evaluation of discharge thresholds anticipation with hydrological ensemble forecasts**

The methodology used to evaluate the anticipation of the exceedances of selected discharge thresholds was directly inspired by the principle of the ROC curves method developed by Mason (1982); Wilks (2011). The ROC curve (Relative Operating Characteristics diagram) is a criterion evaluating the capacity of a forecasting system to detect the occurrence (reciprocally the non-occurrence) of an event. In hydrology, the event generally corresponds to the exceedance of a discharge threshold, and the ROC curve is generally built to summarize long time series of forecasts at one single basin outlet. Comparing at each time step the respective position of the forecast (for a given probability and a fixed lead time), the reference observation or simulation, and the considered discharge threshold, permits to calculate the probability of detection (POD) and the false alarm rate (FAR) (Wilks, 2011; Jolliffe and Stephenson, 2012). The ROC curve is then plotted by using the pairs of points (FAR; POD) corresponding to different forecast probabilities.

In this study, the conventional computation of a ROC curve was adapted following the idea to focus the analysis on the first exceedance of a selected alert discharge threshold, at the event temporal and geographical scales. This led to introduce the following evolutions:

- all the ungauged outlets located in the HFA (1174 in this study) are considered to build the contingency table. For this purpose, hydrographs simulated with observed rainfall are used as reference hydrographs (RS hydrographs hereafter), and the discharge thresholds are adapted at each outlet based on a given return period.

- only one unique time step for each outlet is considered, i.e. the first time at which the RS hydrograph exceeds the considered threshold, or the time of peak of the RS hydrograph, in case of no threshold exceedance. The use of many outlets (1174 in this study) compensates the small number of considered time steps for the elaboration of the ROC curves.

- all the forecasts delivered before and covering the considered time step are merged to evaluate the anticipation/detection of the threshold exceedance, without considering a unique lead time. In practice, we consider that a threshold exceedance is forecasted if at least one of the merged forecasts hydrographs exceeds the threshold. This neighborhood method is in the line of the methods discussed in Schwartz and Sobash (2017), the difference being that here it is a temporal neighborhood by max function that is considered.

According to these general principles, for a given forecast probability, the contingency table is obtained in the following way:

- If the threshold is exceeded by the RS hydrograph (fig B1), the date (day and hour) of the first exceedance is identified. All the runs of forecasts issued before and covering this date are then selected (i.e. 6 runs here according to the maximum forecast lead time of 6-hours and the 1-hour time step between successive runs). A hit is counted in the contingency table if at least one of the six runs exceeds the discharge threshold at any lead time (fig B1 a)), and a miss is counted if none of the six hydrological forecasts exceed the threshold at any lead time (fig B1 b)).

- In case the discharge threshold is not exceeded by the RS hydrograph (fig B2), the date of the peak discharge of the RS hydrograph is first identified. The runs of forecasts issued before and covering this date are then selected. Again, a false alarm is counted if at least one of the selected runs exceeds the discharge threshold at any lead time (fig B2 a)), and a correct rejection if none of the selected runs exceed the discharge threshold at any lead time (fig B2 b))

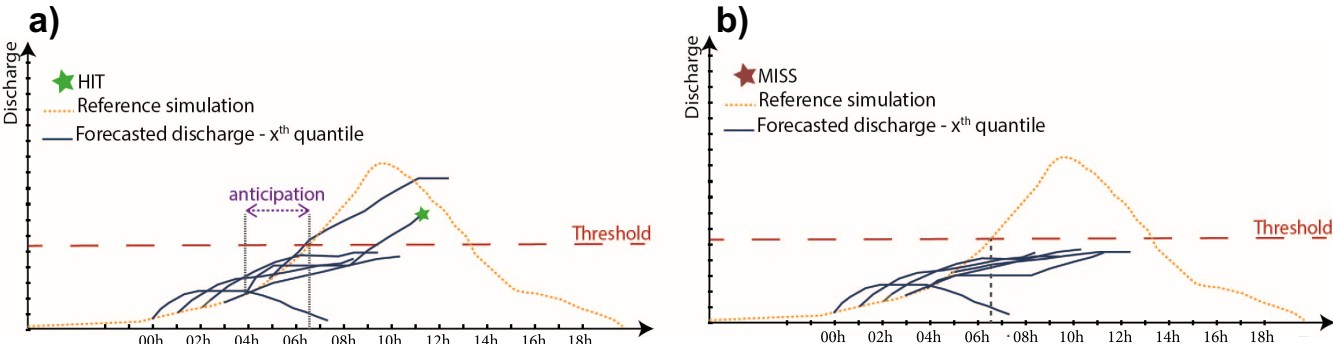

**Figure B1.** Illustration of the discharge threshold exceedance detection method: a) Hit case and b) miss. The way the anticipation is computed is also illustrated (time difference between the time the discharge threshold is exceeded for the RS hydrograph, and the time of the first run which detects the exceedance).

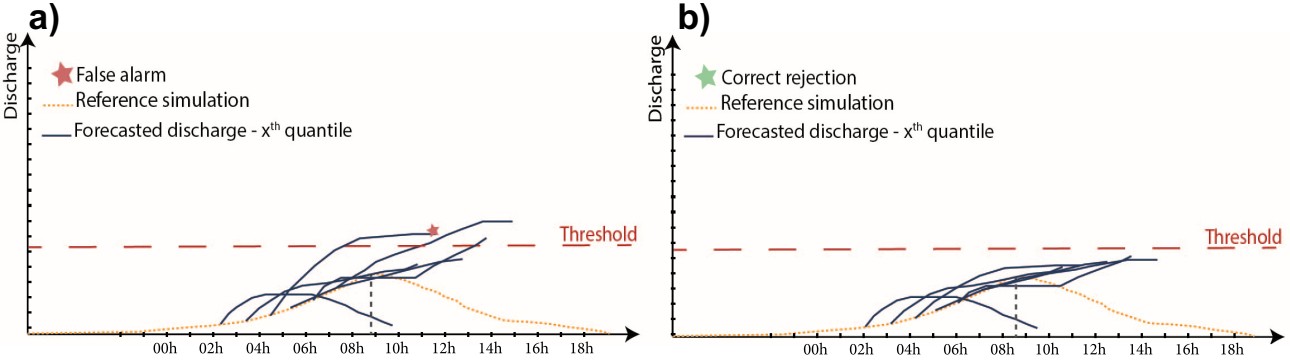

**Figure B2.** Illustration of the discharge threshold exceedance detection method: a) False alarm case and b) correct rejection.

Each contingency table finally contains as many values (hit, misses, false alarms, correct rejection) as the number of outlets in the HFA. The ROC curve can then be drawn by deriving the POD and FAR scores from the contingency tables, following the traditional way described above. An advantage of this way of building contingency tables, is that each table can be presented on a map, by drawing the evaluation obtained at each outlet or sub-basin (hit, miss, false alarm or correct rejection). This permits to easily analyze how the ensemble forecasts performed geographically according to these criteria.

Additionally, for each hit entry of the contingency tables, the anticipation time can also be evaluated (fig B1 a)). It corresponds to the difference between (i) the time of exceedance of the discharge threshold by the RS hydrograph and (ii) the time of the first forecast run that detects a threshold exceedance. We can thus draw the distribution of the anticipation times obtained over the considered outlets, for the different percentiles and ensemble forecast.

## Appendix C: Forecast hydrographs obtained for AROME-EPS for 3-hours lead times

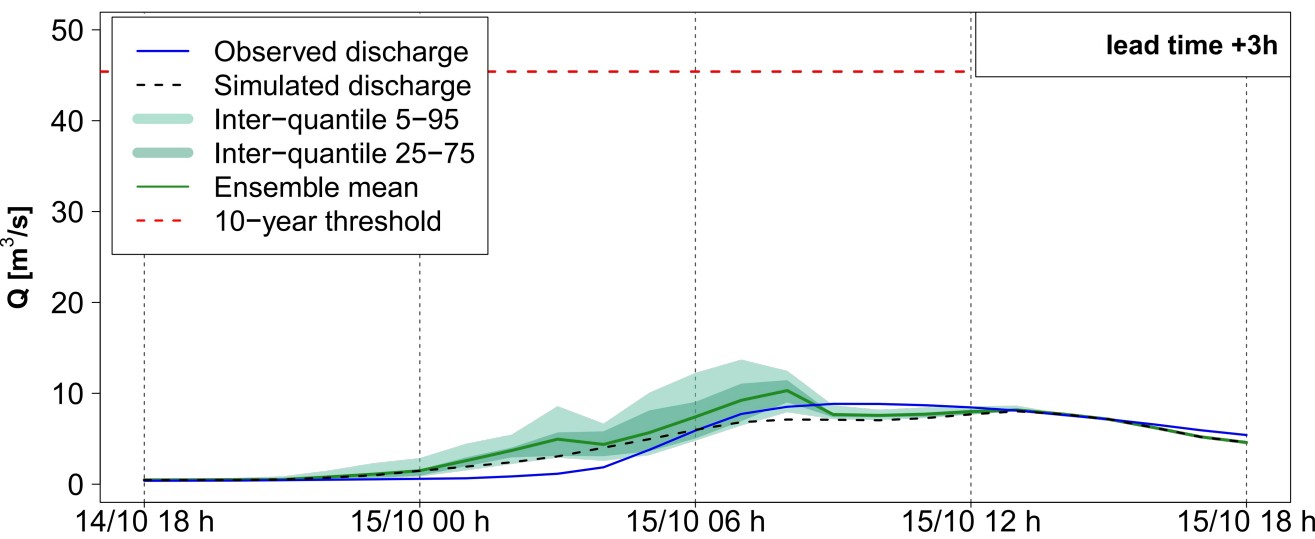

**Figure C1.** Detailed hydrological 3-hours lead time forecasts evaluation at outlet 1 for AROME-EPS. In blue, the observed discharge at the gauging station

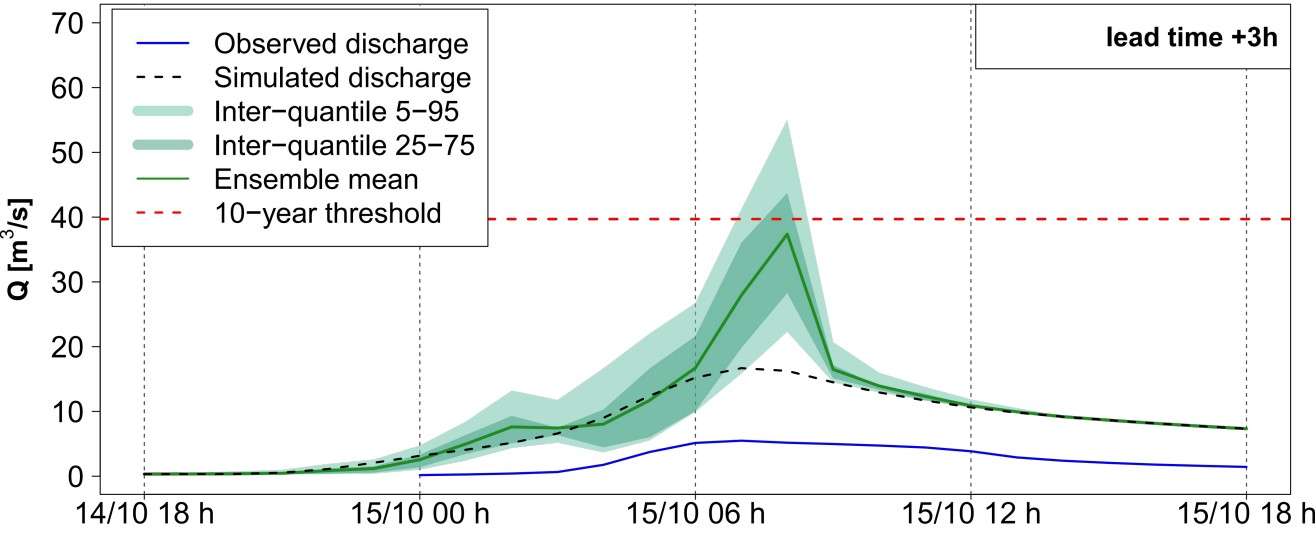

**Figure C2.** Detailed hydrograph for 3-hours lead time forecasts evaluation at outlet 2 for AROME-EPS. In blue, the observed discharge at the gauging station

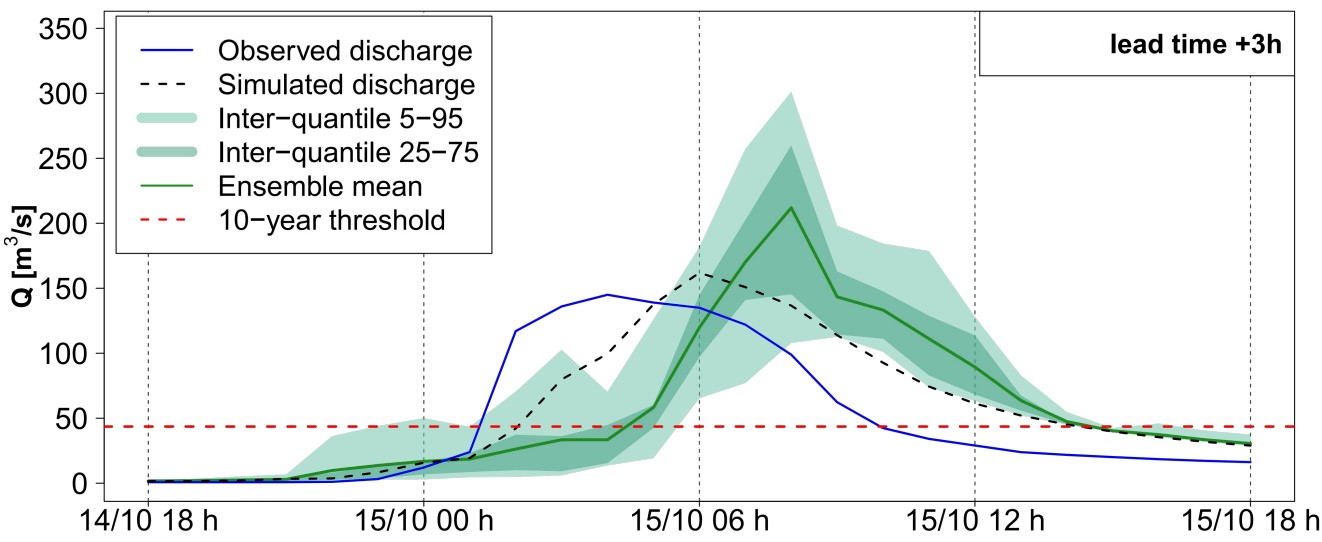

**Figure C3.** Detailed hydrograph for 3-hours lead time forecasts evaluation at outlet 3 for AROME-EPS. In blue, the observed discharge at the gauging station

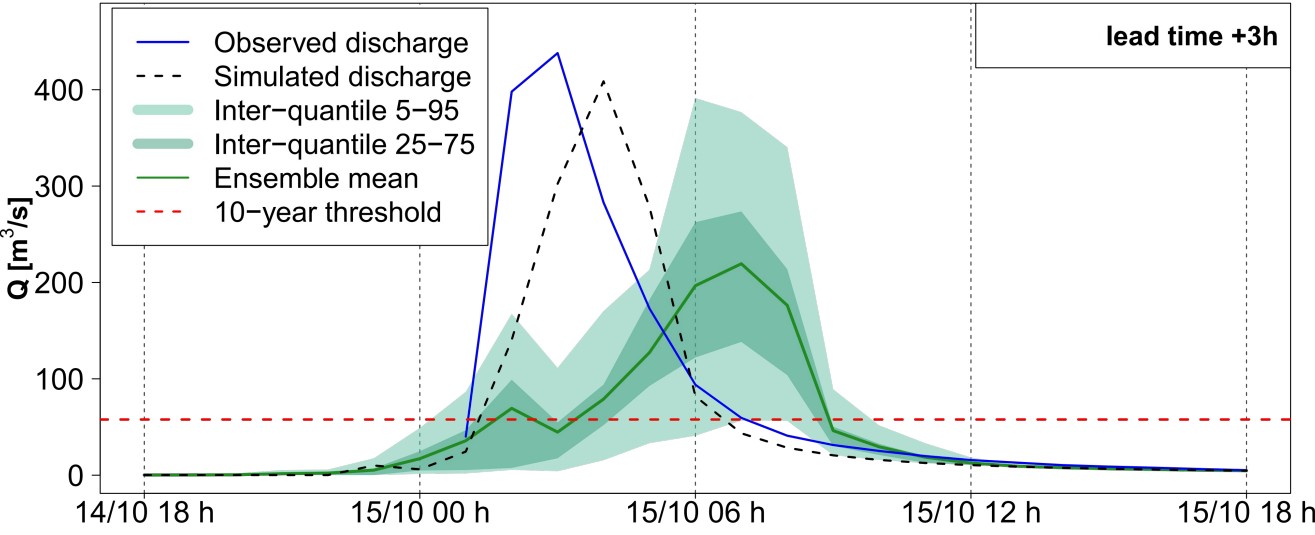

**Figure C4.** Detailed hydrograph for 3-hours lead time forecasts evaluation at outlet 4 for AROME-EPS. In blue, the observed discharge at the gauging station

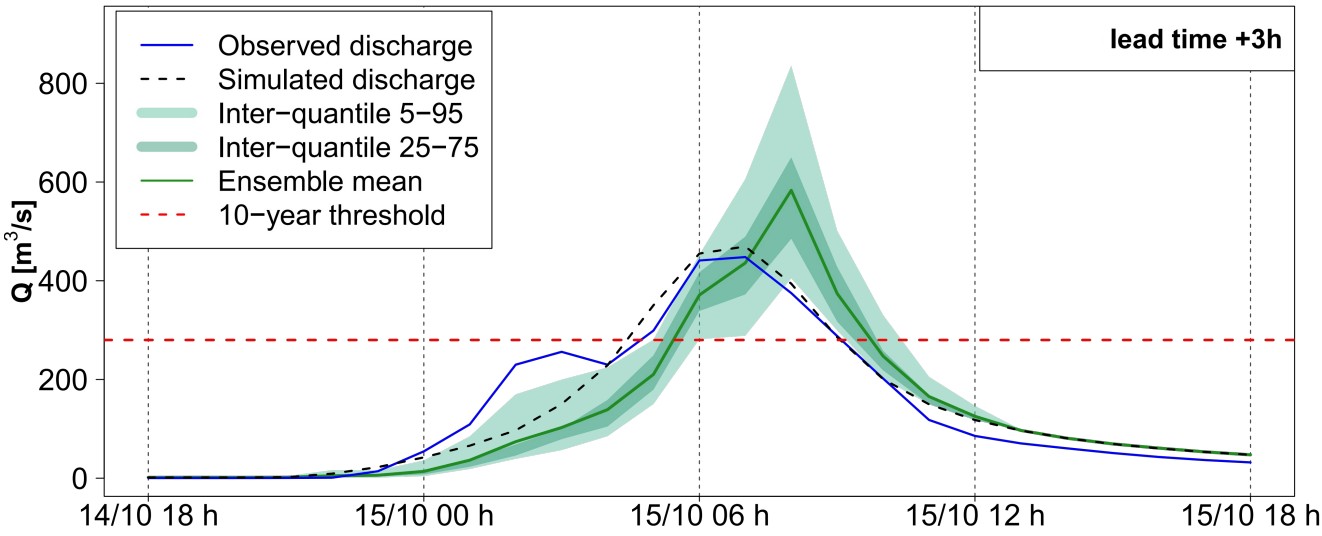

**Figure C5.** Detailed hydrograph for 3-hours lead time forecasts evaluation at outlet 5 for AROME-EPS. In blue, the observed discharge at the gauging station

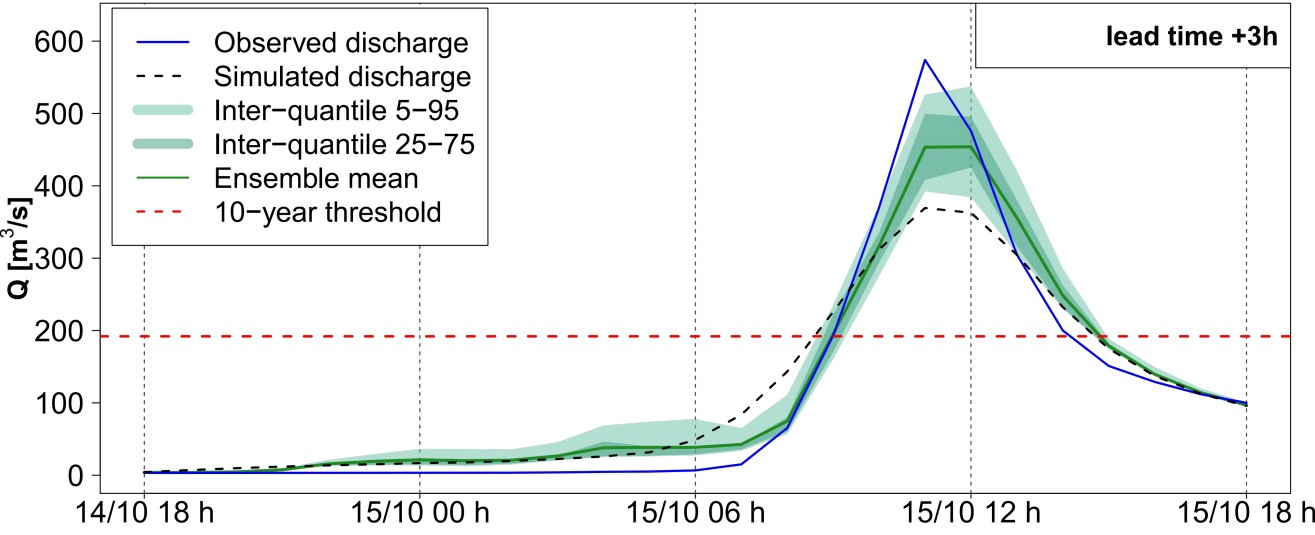

**Figure C6.** Detailed hydrograph for 3-hours lead time forecasts evaluation at outlet 6 for AROME-EPS. In blue, the observed discharge at the gauging station

## Appendix D: Spread/skill relationship and forecast hydrographs obtained for 6-hours lead times

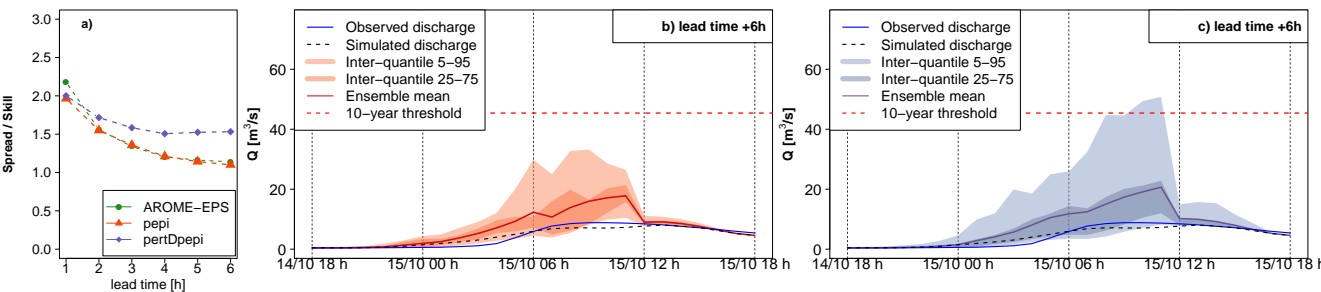

**Figure D1.** Detailed hydrological forecasts evaluation at outlet 1 (216 km$^2$): a) spread/skill relationship, b) hydrographs of 6-hours lead time forecasts with pepi, c) hydrographs of 6-hours lead time forecasts with pertDpepi. In blue, the observed hydrograph at the gauging station.

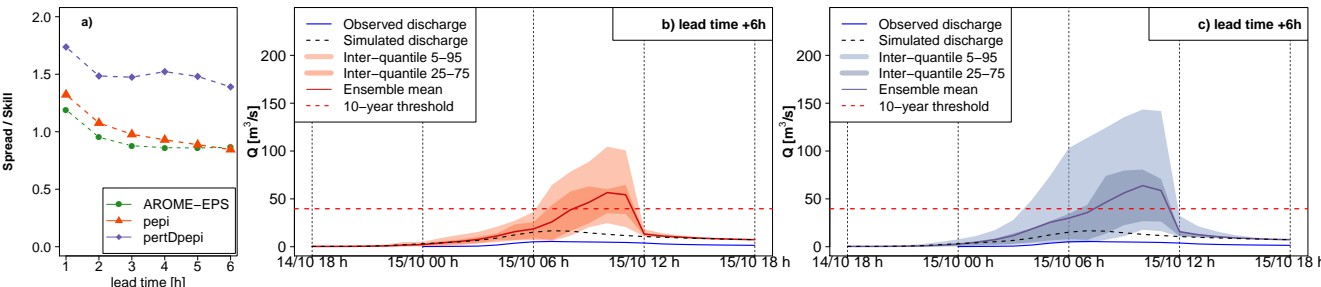

**Figure D2.** Detailed hydrological forecasts evaluation at outlet 2 (197 km$^2$): a) spread/skill relationship, b) hydrographs of 6-hours lead time forecasts with pepi, c) hydrographs of 6-hours lead time forecasts with pertDpepi. In blue, the observed hydrograph at the gauging station.

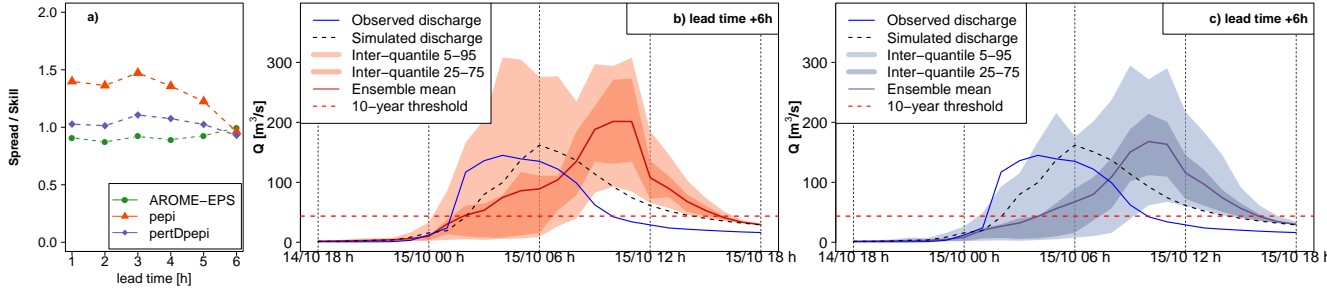

**Figure D3.** Detailed hydrological forecasts evaluation at outlet 3 (85 km$^2$): a) spread/skill relationship, b) hydrographs of 6-hours lead time forecasts with pepi, c) hydrographs of 6-hours lead time forecasts with pertDpepi. In blue, the observed hydrograph at the gauging station.

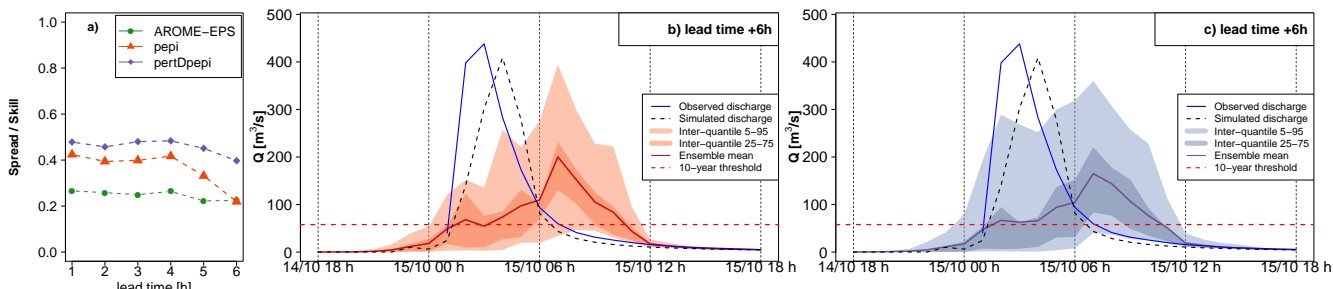

**Figure D4.** Detailed hydrological forecasts evaluation at outlet 4 (173 km$^2$): a) spread/skill relationship, b) hydrographs of 6-hours lead time forecasts with pepi, c) hydrographs of 6-hours lead time forecasts with pertDpepi. In blue, the observed hydrograph at the gauging station.

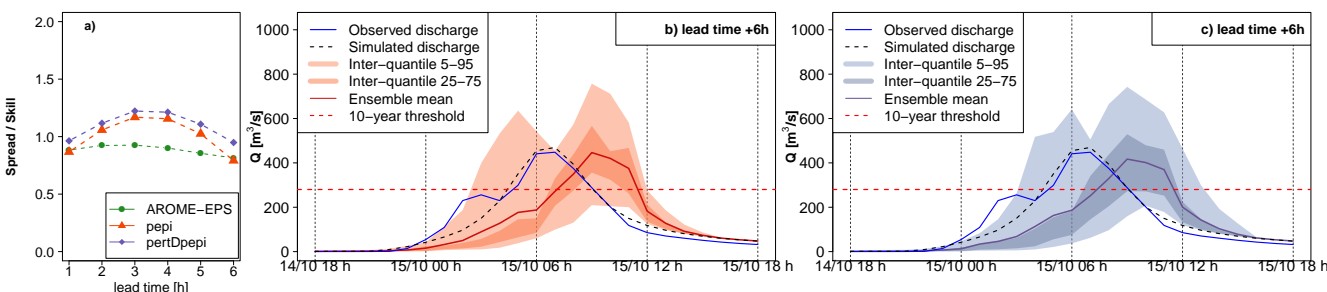

**Figure D5.** Detailed hydrological forecasts evaluation at outlet 5 (263 km$^2$): a) spread/skill relationship, b) hydrographs of 6-hours lead time forecasts with pepi, c) hydrographs of 6-hours lead time forecasts with pertDpepi. In blue, the observed hydrograph at the gauging station.

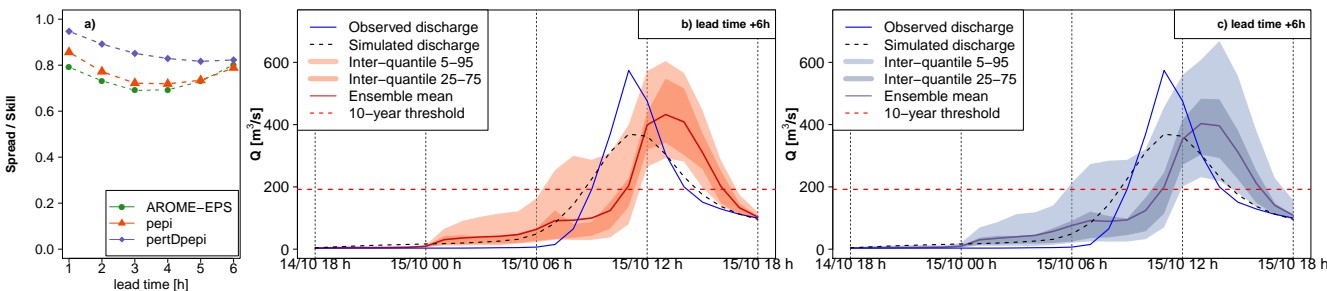

**Figure D6.** Detailed hydrological forecasts evaluation at outlet 6 (257 km$^2$): a) spread/skill relationship, b) hydrographs of 6-hours lead time forecasts with pepi, c) hydrographs of 6-hours lead time forecasts with pertDpepi. In blue, the observed hydrograph at the gauging station.

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
