# Peer review of "A methodological framework for the evaluation of short-range flash-flood hydrometeorological forecasts at the event scale"

_Natural Hazards and Earth System Sciences, 2022_

## Author Comment (AC2)

**GENERAL COMMENTS**

The manuscript proposes a framework to assess the quality of hydrometeorological forecasts for flash flood events and applies it to the event that affected the Aude basin in October 2019.

Conceptually, the proposed framework consists of determining the so-called hydrological focus time and hydrological focus area as the relevant temporal and spatial domains over which the hydrometeorological forecasts are evaluated in terms of the forecasted rainfall accumulations and hydrographs at different points of the river network using existing approaches.

The topic is relevant and the application of the methodology for the analysed event produces interesting results. However, the writing and organization of the manuscript need to be significantly improved to make it ready for publication. Also, some further discussion about the hypotheses made and the applicability of the methodology would make the manuscript more interesting.

Consequently, the manuscript requires major revisions before I can recommend its publication in Natural Hazards and Earth System Sciences.

We thank anonymous referee number 2 for the useful comments about this initial version of the manuscript. We provide hereafter our detailed answers and explanations about the modifications introduced in the revised version of the manuscript (which is already available). Thanks to this revision, we think the manuscript is now much easier to follow.

**MAJOR COMMENTS**

1. The text should be thoroughly revised to improve its clarity, provide a description of all the tools used, avoid repetitions (some aspects appear in several parts of the manuscript), reconsider figures with little discussion (e.g., Fig. 3, Fig. 7), make the text more synthetic (specially sections 4.3 and 5), describe and present all the elements in a sequential way (avoid jumping back and forth), and expand the captions to clearly describe all the figure elements.
   We achieved a general revision of the manuscript to improve its clarity, avoid repetitions, and provide details regarding all the unclear aspects.

2. Organization of the manuscript: Right now the manuscript does not read smoothly. In particular, I think that the readability would improve that Appendix A should be included as a subsection. This could be a rough organization of the manuscript:

   – Introduction
   – Methodology for an event-scale evaluation of hydro-meteorological ensemble forecasts: with the presentation of the 3 steps and the definition of HFA and HFT.
   – Case study, data and models
   – Application of the methodology to evaluate the Ens-QPF products during the event: describing how the methodology has been applied, including the contents of Appendix A.
   – Results
   – Discussion and conclusions: combining current sections 5 and 6.

   The content of Appendix A is important for a good understanding of the manuscript, and thus we agree it should be highlighted. However, the content of this appendix appears quite long to us to be incorporated in the text, and this would also complicate the structure of the manuscript. We preferred to add references to this appendix and to provide more details in the text about its content (section 4.2).

3. The proposed methodology adapts well to the spatio-temporal hydrometeorological features of the analysed event (which shows a quasi-triangular hyetograph in the catchment and mostly single-peak hydrographs). However, I miss some

discussion about how it could be applied to longer, more complex events; e.g., with multiple rainfall periods and multiple hydrograph peaks, or showing high variability of the magnitude of the floods within the affected area. In the latter case, I would like the authors to discuss the possibility of using more than one threshold to assess the quality of the hydrometeorological forecasts; in such a case, would the HFA and HFT be threshold dependent?

The reviewer raises an important question. The HFT and HFA need to encompass the flood event (or the various related flood peaks), and their definition depends on the spatio-temporal settings of the event. We agree that the presented event has relatively simple spatio-temporal features. For events with complex characteristics, particularly multiple flood peaks, it can be considered that each rising or peak phase of the flood event could be examined separately. However, in some cases the different phases of the floods may be difficult to separate: in such situations, each run of forecast may be examined separately along the event. These explanations have been added in the discussion section: "Particularly, in case of events with multiple flood peaks, the fact that the method focuses only on the first threshold exceedance can be seen as a limitation. In such a situation, each rising phase of the flood event could be examined separately, even if in some cases the different phases of the floods may be difficult to separate. An alternative could be to analyze the anticipation of threshold exceedances for each run of forecast during the event, independently of the times the thresholds are exceeded for the RS hydrographs.".

In case of variable flood magnitudes, we agree that several thresholds could be examined for the same event. Evaluating multiple flood events based on the same threshold would also be an interesting extension of the proposed approach. In both cases, it would require an adaptation of the HFA to ensure a balance between the sub-basins for which the considered threshold is or is not exceeded in the RS simulation. It can be noted that other evaluation criterions would less sensitive to the limits of the HFA (such as the Critical Success Index). A specific discussion of these aspects has been added in the text: "Providing an evaluation for multiple flood events, or for multiple thresholds for the same flood, may also be interesting complements to the proposed approach. This may nevertheless complicate the definition of appropriate HFT/HFA, which are event specific. The HFA will also have to be adapted to the considered threshold. To avoid changing the limits of the HFA, an option could be to use a score that does not account for Correct Rejections, and therefore would be less sensitive to the extent of the HFA, such as the Critical Success Index."

**MINOR COMMENTS**

1. Abstract: the final part of the abstract could be more informative about the results obtained in the study and the conclusions

   The following development has been included in the abstract to provide more details about the final results of the work: "The results show that, provided that the larger ensemble percentiles are considered (75% percentile for instance), these products correctly retrieve the area where the larger rainfall accumulations were observed, but have a tendency to overestimate its spatial extent. The hydrological evaluation indicates that the discharge threshold exceedances are better localized and anticipated if compared to a naive zero-future rainfall scenario, but at the price of a significant increase of false alarms. Some differences in the performances between the three ensemble rainfall forecast products are also identified".

2. Motivation of the study. The introduction provides an interesting description of the topic of flash flood forecasting systems and some of their limitations. However, I miss a better connection between the general context description and the presentation of the objective of the study that clearly states the motivation of the study and justifies the proposed analysis strategy.

   We have shortened and modified the text to better focus on the link between the general context of flash flood forecasting and the presentation of the objectives of the study.

3. Page 31, line 685: "to summary" could be "to summarize".

   This has been corrected.

4. Page 31, line 695: at this point the acronym "RS" has not yet been defined.

   This has been changed.

5. Page 31, lines 695-696: The following sentence is not fully clear: "The drastic reduction of the number of considered time steps is compensated by the common consideration of the large number of outlets hit by the event."

The sentence has been rephrased in the following way: "The use of many outlets (1174 in this study) compensates the small number of considered time steps for the elaboration of the ROC curves.".

6. Page 31, lines 704-706: Please, check the writing.

We checked the sentence and modified it: "For each outlet, the discharge threshold is then compared with the RS hydrograph for all time steps of the HFT".

7. Section 4 and Appendix A: Given that RS stands for "Reference Scenario" (page 16, line 360), the expressions "reference RS", "reference RS simulation" or similar need to be corrected.

The RS acronym is defined in section 2.2 as "reference simulation". We decided to keep this definition. The necessary corrections have been done in the text (page 16, section 4 and appendix), where only "RS hydrograph" or "RS scenario" are now used.

8. Page 31, lines 708-810: "All the discharge forecasts issued before and covering this date (according to the maximum forecast lead time, i.e 6 runs) are then selected. For a given forecast probability (ensemble percentile), a hit is counted in the contingency table if at least one of the six runs exceed the discharge threshold at any lead time (fig A1 - left) left), and a miss is counted if none of the six forecast hydrographs exceed the threshold at any lead time (fig A1 - right)". This sentence assumes that the reader is aware about the temporal resolution and lead times of the precipitation ensemble forecasts and how they have been applied to produce discharge forecasts. However, the first reference to Appendix A appears in page 6 (line 161), where none of this information has been provided.

The text has been adapted to mention that the six runs of forecasts correspond to the specific lead time (6h) and forecast refresh time (1h) of this study: "All the runs of forecasts issued before and covering this date are then selected (i.e. 6 runs here according to the maximum forecast lead time of 6 hours and the 1-hour time step between successive runs)".

Also, in this sentence, the way the probabilistic discharge forecasts are treated should be described better. If I understand well, the rainfall-runoff model is run with each member of the ensemble of precipitation forecasts to generate an ensemble of hydrographs (one per rainfall forecast member); and from these the ROC analysis is based on setting probability thresholds to obtain the associated time series of discharge forecasts. Because these are not necessarily obtained from a single run of the rainfall-runoff model, I would not use the term "hydrograph" when referring to them (page 31, line 711).

This is right, the hydrologic model is run for each member of the rainfall forecast to generate a hydrological forecast ensemble. From this ensemble, a probability is applied to obtain a time series of discrete discharge forecasts. Examining successively different probabilities finally leads to the ROC curve. This information has been added in the text (section 4.2) : "The hydrological model is first run for each member of the rainfall forecast to generate a hydrological forecast ensemble. From this ensemble, at each hydrological outlet in the HFA time series of forecast discharges are obtained for several probability thresholds.".

9. Fig A1. I would expect the oldest forecast to end at the evaluation time, and the newest forecast to be issued 1 hour before the reference time. In the figure, I cannot see this. Also, explain (at least in the figure caption) what the term "anticipation" used in the Figure shows.

Since the hydrological model is run at a 15-min time step, and the runs of forecasts are issued every hour, the end of the first considered run can slightly exceed the evaluation time (exceedance of the threshold by the RS hydrograph), and the last considered run can be issued less than 1 hour before the evaluation time. The definition of the anticipation has been provided in the caption.

10. Page 32, line 715: One could think that, if a correct negative occurs in the time range between t-6h and t, but the discharge forecasts exceed the threshold in a different time step, this situation should be classified as a false alarm. I would like to know the authors' opinion about this aspect and how it affects the presented results should be included in the manuscript.

It is right that all the forecasts runs issued during the event are not analyzed. We chose to focus systematically on the 6 runs covering the most critical phase of the event (i.e. the threshold exceedance, or in this case the maximum of the

RS hydrograph). Other choices could have been done, and the content of the contingency table may change if additional runs of forecasts were considered. We therefore added the following sentences in the discussion section: "It can be noted that only the runs of forecasts covering the most critical phase of the event (i.e. the time of the threshold exceedance, or the maximum of the RS hydrograph) were considered to build the contingency tables. The results obtained could differ if other runs of forecasts and/or other phases of the event were considered. Particularly, in case of events with multiple flood peaks, the fact that the method focuses only on the first threshold exceedance can be seen as a limitation. In such a situation, each rising phase of the flood event could be examined separately, even if in some cases the different phases of the floods may be difficult to separate. An alternative could be to analyse the anticipation of threshold exceedances for each run of forecast during the event, independently of the times the thresholds are exceeded for the RS hydrographs."

11. Page 32, lines 717-719: "as many values (. . . ) as the number of outlets in the HFA". By combining the results obtained in the different sub-catchments, one could be masking the quality of the forecasts in the most affected areas with those where the event did not event reach the threshold. This could be quite serious in moderate or very local events. Similarly, how would the method be applied in more complex events (e.g. with multiple flow peaks over a few days or affecting sub-catchments of different catchments)?

We agree that the analysis should avoid masking the performance of the forecasts in the most affected area. The risks of false alarms in the areas where the threshold was not exceeded should also not be masked. This is exactly why the choice of the HFA and of the considered threshold are very important: this helps including in the analysis both affected areas and areas where false alarms may be observed according to the issued rainfall forecasts. The representation of the results in the form of maps (Figure 9) and not only as a ROC curve also avoids an overly global view of results (and masking the exact location of errors). The use of a different score, such as the Critical Success Index, can also be an alternative to limit the sensitivity to the spatial extent of the HFA. Regarding the events with complex spatio-temporal features, see our response to general comment n°3.

12. Page 7, line 190: "The Aude River basin is located in southwestern France". It could be more appropriate to use "southern France".

We changed southwestern to southern.

13. Page 7, line 199: I do not fully understand what is meant by "to be compared to the local 100-year percentile of 200 mm in 6-hours (Ayphassorho et al., 2019)".

The sentence was rephrased as follows: "The maximum accumulated rainfall amounts over short durations were also extreme: up to 60 mm in one hour and 213 mm in six hours recorded at Villegailhenc (Figure 2), while the local 100-year rainfall accumulation is 200 mm in six hours (Ayphassorho et al., 2019)".

14. Figure 2, caption: Please, describe how the rainfall accumulation map was obtained. Could you please verify that this is a 47-h rainfall accumulation map as the caption suggests? Also, it could be interesting to include the location of the 31 stream gauges in the Aude catchment mentioned in section 3.2 (lines 219-220).

The rainfall accumulation map was computed for the October 14 00:00 to October 15 23:00 period. The rasters of hourly Antilope J+1 quantitative precipitation estimates (combining radar and ground measurements) were just added to obtain the map. This information has been included in the caption as follows: "The Aude River basin, its river network, and the observed rainfall accumulations observed from 14 October 2018 00:00 to 15 October 2018 23:00, according to the ANTILOPE J+1 quantitative precipitation estimates (see Section 3.2).".

15. Page 9, line 226, (title of Section 3.3). For consistency, use "AROME" everywhere within the text.

The modification was done.

16. Page 9, line 240: The sentence "The number of members in the "pepi" product is 18 (respectively 13) for a lead time of 1h (respectively 6h)." needs some rephrasing to guarantee its clarity. Is the 1-h leadtime pepi product used in this study?

We modified the sentence to provide more details:"The resulting "pepi" product provides forecasts for a maximum lead time of 6 hours. It combines 12 members from the last available AROME-EPS run, and 1 to 6 members from AROME-NWC, depending on the considered lead time. The resulting number of members varies between 13 (for 6-hour lead

time) and 18 (for the 1-hour lead time).". All the lead times of the three ensemble products are combined to compute ROC curves and threshold exceedance anticipation maps (see Appendix A).

17. Section 3.3: I suggest finding alternative notation for the terms "pepi" and "pertDpepi" that describes better these two sets of ensemble forecasts. What do these terms stand for? What are their spatial resolution and rainfall accumulation window?
Pepi stands for the contraction of AROME-EPS (AROME-PE in French) and AROME-NWP (AROME-PI in French), and pertDpepi for the PERT method (Vincendon, 2011) applied on pepi ensemble. We agree this notation is not fully explicit for English-speaking people, but nevertheless this does not affect the understanding of the manuscript in our opinion. The spatial resolution is 0.025° and the spatial window covers the metropolitan territory of France (see next comment).

18. Pages 9 and 10, lines 231-247: the description of rainfall ensemble forecasts needs to be rewritten to guarantee that it is clear how the forecasts from these 3 products have been applied in the study (not only what are the maximum lead times, but also if some spin-up time has been established, how the hourly frequency has been handled in the case of the AROME-EPS...). Also, information about the spatial resolution of the grids and about the rainfall accumulation windows needs to be provided.
For an improved clarity, the paragraph describing the spatial resolutions of the ensemble forecast products has been moved just below the description of the ensembles, and further details have been added, including the spatial window and resolution.

19. Page 10, lines 245-247: "The spatial shift applied to this product represents an ideal distance because i) it captures the main uncertainties due to the localization of the rainfall event, and ii) it is a shift that does not combine too incompatible areas." Is there any reference to support such a statement? How could this be verified?
This description has been rewritten (also in response to reviewer 3 about lines 619-621) as follows: "The shift scale of 20 km represents a typical forecast location error scale: according to Vincendon et al. (2011), 80% of location errors are less than 50 km. The value of 20 km has been empirically tuned to produce the largest possible ensemble spread on a set of similarly intense precipitation cases, without noticeably degrading the ensemble predictive value as measured by user-oriented scores such as the area under the ROC curves."

20. Page 10, lines 246-247: "it is a shift that does not combine too incompatible areas." Please, clarify.
We rephrased this sentence, see answer to comment n°19.

21. Figure 3: What is shown in a reliability diagram needs to be clearly described to facilitate the interpretation of this figure by the non-expert reader (for the ROC curve, at least, mention that this interpretation can be found in Appendix A). Also, the text in Fig 3a needs to be clearer (ensure the readability of all numbers).
We preferred here to remove this figure which is not completely in line with the objectives of the paper, as mentioned by referee n°3: evaluation of QPFs at large temporal and spatial scales and for a relatively low threshold of rainfall intensity (5 mm/h). Moreover, the figure is based on scores which differ to the ones developed in the paper (or at least which are computed differently).

22. Page 10, line 254. I suppose that "≈ 2 km2" should be "≈ 2 x 2 km2". Is this the original resolution of the EPS grids? How were the different resolutions between observations ( ≈ 1 x 1 km2)" and the forecasts treated to do the evaluation (e.g. Figure 3)? Were the observations upscaled to the forecasts grid? Or the forecasts interpolated to the observations grid?
The ensemble forecasts are actually provided on a 0.025° by 0.025° grid. For the comparison with observations (rank diagrams), the forecast values have been disaggregated on the 1 km x 1 km grid. These explanations have been added in the text.

23. Page 11, line 271. Please, specify that KGE stands for the Kling-Gupta efficiency, and provide a reference.
We detailed the acronym in the text and added a reference for the KGE criteria.

24. Page 11, lines 271-273: "The KGE calibration (validation) values obtained were of 0.80 (0.71), which indicates good model performance, except for one validation outlet, where a low KGE value of 0.1 was obtained (Figure 4a)." It is unclear where the reported KGE values (0.80 – 0.71) were calculated. At the downstream-most level-gauges? Are these the average KGE values at all the gauge stations? Besides the validation gauge with KGE 0.1, Figure 4a shows the KGE is, approximately, between 0.6-1 at the calibration gauges and between 0.3 and 0.8 at the validation gauges.

    The values provided in the text correspond to averaged values for the 16 calibration and 15 validation stream gauges. This explanation has been added in the text.

25. Figure 4b. The reference to the "HyMex estimates" is only provided in section 3.2. The reference to the section or to the work of Lebouc et al. (2019) could be added in the figure caption or in the description of CINECAR.

    We included a reference to Lebouc et al. (2019) in the caption of the Figure 4b.

26. Page 11, line 291: What is "ANTILOPE J+1"?

    ANTILOPE J+1 corresponds to the QPEs obtained with the ANTILOPE algorithm (Laurantin et al, 2008), by combining the radar data and rain gauge observations available the next day (J+1). This information has been added in the "Observed hydrometeorological data" section (Section 3.2).

27. Page 11, lines 296-297: "with some few exceptions that can be explained by the spatial averaging". What does it mean? Is not the same averaging applied to the 3 ensemble forecasts and over the same area?

    The spatial averaging is the same for the three ensemble products. Since there are several time steps (15-10 11h - Figure 5a) where the dispersion is surprisingly lower for pepi and pertDpepi than for AROME-EPS, we think this lower dispersion could be explained by the spatial averaging over the chosen spatial window: the members added in pepi an pertDpepi, even if different from the members included in AROME-EPS, may have a similar averaged value. We removed this confusing sentence, which is not essential.

28. Figure 5. The range of the y axis for the two panels should be the same. In the figure caption, it would be useful to state that the Aude catchment is 6074 km2.

    We modified the y-axis with the same range for the two panels, and modified the caption of Figure 5.

29. Page 14, lines 310 – 317. The selection of the HFT seems to be quite subjective. Why is it based on a threshold of the Aude average rainfall intensity of 2 mm/h? The discussion about the analysis of the results being dominated by periods of low rainfall intensities would also apply to the fact that several parts of the catchment registered low rainfall. Similarly, the decision of taking the Aude catchment as the HFA is arbitrary. How much these decisions could have an effect on the obtained results? Could the HFT and HFA be obtained based on more objective criteria? For instance, considering the spatio-temporal structure of the observed and forecasted 1-h rainfall accumulations as depicted by the space-time correlogram or variogram? Discussion about these questions would be very interesting.

    We agree the selection of the HFT and HFA can be relatively subjective. We therefore mentioned in the text that other thresholds could have been selected. We think the HFT and HFA should just be determined by answering the following question: when and where floods could have been observed according to observed AND forecast rainfall fields? Even if several choices are possible, answering this question should lead to set a HFT and a HFA accounting respectively for timing and spatial errors of rainfall forecasts. The thresholds we applied were determined in this way. For the HFT, the threshold of 2 mm/h can be seen as relatively low, but it corresponds to a spatial average and thus reflects significantly larger point rainfall intensities. For the HFA, almost all the Aude river basin is covered with rainfall forecasts accumulations exceeding 150 mm, at least for the 75% and 95% quantiles. We consider these thresholds (2 mm/h of spatial averaged intensity and 150 mm of point rainfall accumulation) both result in significant risks of flooding. Extending the HFA outside the limits of the Aude river catchment would even have been possible (but not essential, since the Aude catchment already largely exceeds the area of the actually observed intense rainfall cell). Examining the spatio-temporal structure of observed or forecast rainfall could be an alternative, but in our opinion this would also necessitate to set thresholds and would also result in some subjectivity.

30. Page 14, lines 329-331. The text gives the impression that some members clearly overestimate the rainfall in the catchment. Although Fig. 5 shows that this is the case by a few mm/h, there are no individual members showing average rainfall accumulations over the catchment similar to those of the 75%- and 95%-percentiles. Instead, the maps of Fig. 6 (second and third rows) are most likely the result of different members showing the largest accumulations in different locations in the catchment. Consequently, to a good extent what is referred in the text as "false alarms" are mostly location errors.

We agree that figure 6 is neither representative of individual members, nor of actual forecast rainfall accumulations, since it combines several runs of forecasts. It is rather a cumulated representation of the areas affected with high rainfall intensities, for the successive runs of forecasts and for a fixed lead time and quantile. We added a sentence in the beginning of this paragraph to remind this: "Note that the forecast panels do not correspond to rainfall accumulation for one unique run of forecast, but rather to a cumulated representation of the areas affected with high forecast rainfall intensities, for the successive forecasts issued during the event." We also completely agree that the spatial extent of the large rainfall rates in the second and third rows result from the location errors of the successive runs of forecast, and therefore added the following sentence in the paragraph: " .. the area of high intensities spreads and becomes larger than the area seen in the observed field of rainfall accumulations. This may be attributed to the location errors of some members in the successive runs of forecasts." We also modified the text of lines 329-331 to avoid any confusion : "Even if not entirely hit by the observed heavy precipitation event, the Aude River basin is almost entirely covered with repeated high forecast rainfall intensities during the event. This led to the choice of keeping the entire Aude River basin as HFA. Considering this whole area will help in evaluating the risks of false alarms attributed to rainfall forecast location errors when forecasting floods."

31. Page 15, line 339. "largest" could be replaced by "highest".
This has been modified.

32. Fig. 6. It would be very useful to provide the values of the event accumulation in the catchment for each panel. My impression is that the 75% percentiles show significantly larger catchment accumulations than those observed, and probably a lower percentile would be closer.

The values of event accumulation are represented in the observed panel (there is no quantile in the event observed accumulations as we one have one observation, not members). As mentioned in our answer to comment n°30, the right panels do not correspond to actual forecast rainfall accumulations, since they combine several runs of forecasts. Providing forecast rainfall accumulations for one single run of forecast would be possible, but this would not represent the event accumulation because of the limited lead time of 6 hours. .

33. Page 16, line 346-347: "As a consequence, to produce effective hydrological forecasts based on a good estimate of the rainfall rates…, users would need to work based on a high ensemble percentile value (the 75% percentile in the present case … )" I find this sentence misleading, as it could give the impression that this is the rainfall that has been used in the analysis (which would be contradictory with what is described in Appendix A, page 32, line 347, "for each considered forecast percentile"). Also, the discussion about how using a high percentile might generate false alarms could fit better in the discussion.

As mentioned in appendix A, all percentiles from 5% to 95% are used to calculate the ROC curves presented in Figure 8. This phrase "As a consequence, to produce effective hydrological forecasts based on a good estimate of the rainfall rates…, users would need to work based on a high ensemble percentile value (the 75% percentile in the present case … )" just explains why a specific interest is given to the 75% percentile in section 4.2 (Figures 9 and 10). The description of figure 6 has been grouped and modified, and we believe there is no risk of misunderstanding anymore. The sentence about the general decision principle has also been moved to the discussion section.

34. Caption of Fig. 7. Mention the hourly rainfall thresholds for the presented ranked histograms.
We completed the caption with the different thresholds used.

35. Discussion about Fig. 6 appears before and after the discussion about Fig. 7. Please, combine them (one option could be that Fig. 7 appears before Fig. 6).

We modified the text to group and simplify the description of Figure 6. Because Figure 7 is dependent on HFT and HFA, we prefer to show it after Figures 5 and 6.

36. Page 16, lines 360-361: "Hourly rainfall accumulations were uniformly disaggregated to run the model at a 15-min time resolution." Why is this necessary?
The Cinecar model runs at a 15-min time resolution, whereas the rainfall observations and forecasts are provided at an hourly time step. We modified the sentence: "Hourly rainfall accumulations were uniformly disaggregated to fit the 15-min time resolution of the model."

37. ) Page 16, lines 368-369 ("This means that one unique result (either a hit, a miss, a false alarm or a correct rejection) is obtained for each of the 1174 sub-basins"). Please, specify that this is for each probability value (see also comment 33).
The sentence has been modified : "This means that one unique result (either a hit, a miss, a false alarm or a correct rejection) is obtained for each of the 1174 sub-basins and for each ensemble percentile."

38. Page 16, line 371: By highlighting the 75% percentile in the ROC curve, it gives the impression that this result is obtained with the rainfall of Fig. 6 (see also comment 33), whereas this is the result obtained from setting a 75% on the forecasted discharges.
The ROC curves presented in figure 8 correspond to the hydrological forecasts (obtained with the Cinecar model). This is already mentioned in the caption of the figure, but we adapted the text to avoid any risk of misunderstanding. The 75% percentile has been highlighted here since it leads to balanced proportions of detections and false alarms.

39. Page 18, lines 385-386: "This is clearly the dominant effect for the 75% percentile of the pertDpepi ensemble product and the 2018 event." Please, refer to Fig. 9.
The reference has been added in the text.

40. Figure 9, caption: "Maps of anticipation (0-6h) of the 10-year return period discharge threshold". If I understand correctly, this is not what the figure shows.
The maps presented in this figure show the ability to anticipate a flow threshold exceedance for the 75% percentile for the RF0 scenario and the three hydrological ensemble forecasts. This is a spatial representation of the content of the contingency table for each outlet: hit (dark green), miss (dark red), correct rejection (light green) or false alarm (light red). We modified the caption to be more explicit: "Maps illustrating the detailed anticipation results (hits - misses - false alarms - correct rejections) of the 10- year return period discharge threshold, for the hydrological forecasts based on .." .

41. Page 19, lines 387-388: I would expect that the first point of the ROC for the 3 ensemble forecasts should be almost identical to that of RF0 scenario (which is almost the case). My interpretation is that the skill shown by the RF0 point (particularly the hits shown in Fig. 9) is due to the catchments' response to past rainfall. Do you agree?
Yes we agree. By definition of the RF0 scenario, the False Alarm Rate equals 0 and the Probability of Detection corresponds to the combined effect of past precipitation and propagation times. Since the first point of the ROC curves for the ensemble forecasts corresponds to the 5% percentile, one would expect a POD that is at least slightly better than for the RF0 forecast. This is exactly what is observed.

42. Page 19, line 389: "All ensemble forecasts lead to an increase of the number of hits (9)". Should "(9)" be "Fig. 9"?
The word "Figure" was missing. We corrected it.

43. Sections 4.2 and 4.3. The results of Section 4.2 were obtained with the CINECAR model, and those of Section 4.3 with GRSDi. If no comparison between models is provided, what is the advantage of using 2 different models? At least some discussion about the 4.2-Hydrological anticipation capacity of GRSDi should be provided.
The purpose of this work was not to compare the performance of the two hydrological models. Since the reference RS scenario corresponds to the simulated hydrographs, the evaluation of the hydrological ensemble forecasts do not highly depend on the model used. The advantage of using Cinecar for the evaluation of the 10-year discharge threshold (section 4.2) is that this model is highly distributed and enables to draw detailed anticipation maps including small ungauged basins (Figure 9). For hydrographs (section 4.3), we preferred here to use the GRSDi model since this model was not

exclusively calibrated on the 2018 event (even if the calibration period - October 2008 to October 2018 - includes the 2018 event), and thus the hydrological forecasts can be compared to both the RS scenario and the observed hydrographs to illustrate the importance of hydrological modeling errors in a real world forecasting situation (the GSRDi model is close to the models used for operational flood forecasting in France). A paragraph including these explanations has been integrated in section 3.4. More details about the application of the GRSDi model can be found in Peredo et al. (2022).

44. Page 20-21, lines 424 – 434. Please, add the reference to Figures 11 and 12.
    References have been added.

45. Page 21, line 441: it should be clarified how both the "spread" and the "skill score" have been calculated. Also, in the y axis of Figs. 11a-16a, it seems that the units spread / skill are mm. Is this correct?
    The spread/skill score has been calculated by following the methodology proposed by Fortin et al. (2014). While the methodology to obtain the skill score seems to be well known (root-mean-square error RMSE of the ensemble mean), the authors also explain that it is not the case for the spread and has previously led to some mistakes. They demonstrate that the ensemble skill should match the average of the standard deviation of ensemble forecasts, and they explain that the ensemble spread is in fact the square root of the average of the squared values of the standard deviations. They propose the following equation (equation 15 in Fortin et al. 2014).

$$spread = \sqrt{\left(\frac{R+1}{R}\right)\left(\overline{s_t^2}\right)^{\frac{1}{2}}} = \sqrt{\left(\frac{R+1}{R}\right)\frac{1}{T}\sum_{T=1}^{T}s_t^2} \qquad (1)$$

Where $R$ is the number of ensemble members, $s$ is the standard deviation of ensemble forecasts, $T$ the number of analyzed time steps, and $t$ is a time step.

We added a short explanation for the computation of the spread/skill score (section 2.3), and corrected the y-axis caption (adimensional), we thank the referee for helping us notice this mistake.

46. Figures 11-14: Some of the discharge forecasts show obvious biases with respect to the reference (simulated discharge). Some interpretation about this could be interesting. How do these biases affect the spread / skill results and their interpretation?
    Large bias between forecasted and simulated discharges should result in large skill values and therefore low spread/skill scores. Spread/skill scores significantly lower than the target value of 1 are observed only in the case of figure 14. But as already mentioned in the text, the reason in this case seems to be rather a timing error than a real systematic bias. In the case of figure 12, the large bias corresponds to a modelling error (difference between observed and simulated hydrograph) and not to a forecasting error (also mentioned in the text). Thus, in this case, the spread/skill score is logically not affected.

47. Figure 14 (panels b and c). The legend hides part of the results (observed and simulated discharges).
    We have modified the figure legends in order to avoid any covering of the discharge values.

48. Page 27: The title of section 5.2 could be more concise.
    We changed the title to "Performance of the three ensemble forecast products" instead of "What should be concluded about the comparative performance of the three forecast ensemble products evaluated?"

49. The study focuses on the evaluation of flash-flood hydrometeorological forecasts at the event scale. It could be interesting to add some discussion about how/if the method could be applied to evaluate the performance of the forecasting system on a multi-event framework. Also, it could be interesting to include some discussion about the applicability of the method to other regions and countries.
    Applying the method to other regions or countries prone to flash-floods should not cause any difficulty if the required data and models are available. A development on the question of applying the method in a multi-event and/or multi-site framework has been added in the discussion section: see our answer to general comment n°3.

50. The Introduction states that "We adopt the point of view of end-users, who aim at providing resources and assistance for evacuations and rescue operations at a regional scale." However, I have not found any analyses or results supporting this statement beyond a few statements in sections 5 and 6 that are quite general.

We reformulated this paragraph as follows: "In this approach, the evaluation is mainly focused on the capacity of the hydrometeorological forecasts to anticipate the exceedance of predefined discharge thresholds and to accurately localize the affected streams within the region of interest. These are two essential qualities of hydrometeorological forecasts that are needed to plan rescue operations in real time."

---

## Author Comment (AC3)

**Overview**
The manuscript shares its focus between the verification of accuracy of ensemble precipitation forecasts and different ways to convey (and analyse) the information provided in terms of discharge forecast by a meteo-hydrological forecasting chain. On the one hand, the theme of forecasting severe rainfall events is largely discussed in the introduction, but a in-depth analysis on the verification of output by NWP models (and related ensembles) is neglected. On the other hand, it is declared that a new framework for the evaluation of meteo-hydrological model coupling is proposed, but a proper review of past studies about this issue is not provided in the introduction and the proposed analysis recalls (and put together) different approaches commonly used in the operational practice of worldwide flood forecasting centers. In addition, many parts of the proposed evaluation framework appears as unsuitable for real-time applications. The overall feeling about the present manuscript is that it describes a very detailed post-event analysis, where the parts of novelty and originality do not clearly stand out. A clear choice about the main goal of the study should be taken and then properly developed. In my opinion, the strong point of the paper should be the availability of three meteorological ensemble products (even though it is not clear if a performance comparison for these ensembles is a novelty or past studies have already investigated the subject). The performance evaluations in terms of Quantitative Precipitation Forecasts (QPFs) should be based on a larger dataset and taking into consideration the concept of "fuzzy verification". The analysis of outcomes provided by a meteo-hydrological model chain driven by the available ensemble QPFs is an added value for the study.

We thank referee n°3 for the feedback, showing that the objectives of the paper were probably not presented sufficiently clearly. We managed to improve this in the revised version of the manuscript (which is already available, see our detailed answers hereafter describing the modifications introduced). The main misleading point was probably that the article does not aim at evaluating QPFs per se, but rather the performance of flood forecasts obtained by using these QPFs as input of rainfall-runoff models. The initial analysis of QPFs (section 4.1) is only meant to provide the required background information for the analysis of flood forecasts (sections 4.2 and 4.3): i.e. to check that the QPFs are of reasonable quality. The presented assessment method can only be implemented a posteriori and not in real time. It aims to provide a first detailed and informative diagnosis of the performance of flood forecasts, for single major flood events where such forecasts are needed. Moreover, an immediate post-event analysis is often needed to understand better what went right or wrong during the flood forecasting and response. This paper aims at serving such a purpose.

**General comments**

1. The main declared aim of the manuscript is to presents a methodological framework for the event-based evaluation of ensemble forecasts for floods, with respect to the needs of civil protection authorities. But, the proposed analysis is quite complex (several aspects and score to consider), maybe not suitable to the real-time operational practice of flood forecasting centers. The current contents of Section 4 sound more like a post-event analysis. In addition, the use of verification metrics like rank diagrams and ROC curves to analyze a single event has poor significance (Figs. 7 and 8). These metrics are commonly used over large datasets, in order to highlight statistical characteristics of the forecast product. The computation over a single event could be of some interest if compared to "historical" performances based on a long archive (for instance, for real time applications, the spread skill relationship in Figs. 11-16 does not add significant information with respect to the issue of warnings and outcomes shown in the remaining panels).
The statistical analysis in terms of discharge forecast should consider the whole period covered by QPFs (not just a flood event).

   The proposed evaluation is aimed at post-event analyses. Providing a real-time assessment is clearly not the objective of this paper. The aim is to provide an evaluation, after the event, including meaningful information for the users of hydrological forecasts. Providing assessments of the capacity to anticipate discharge thresholds, which may correspond locally to damage thresholds, and hence to decisions and actions of emergency services (not only flood forecasters) is essential to evaluate the usefulness of hydrometeorological forecasts for improved event management. Even if the proposed assessment is event specific and cannot be fully extrapolated to future events, each event being specific, drawing

lessons from each specific flood event appears essential to us. This objective was discussed with the French operational flood forecasting service (SCHAPI), which was directly involved in the research project hosting this work (PICS project).

Building an evaluation based on a large historical period of forecasts, or numerous flood events, would certainly be much more robust for an in-depth analysis of any systematic errors of a flood forecasting system. But unfortunately, this is often not possible, particularly when dealing with extreme flash floods. The reasons for this are detailed in the introduction (3rd paragraph): we are dealing here with experimental rainfall forecast products based on recent evolutions of NWP models, and which were not released for a long historical period. This is often the case when dealing with flash-flood forecasting. Additionally, our objective is to evaluate hydrological flood forecasts and not QPFs. From this point of view (flood forecasting), extending the analysis to a long period of low flows would not be relevant. Thus, the analysis has to be focused on the (rare) periods of floods. This is precisely the novelty of this paper, which adapts some conventional evaluation metrics to provide a first evaluation in the specific context where only some few relevant flood events have been observed and documented. For this purpose, we propose to exchange time for space by examining flood forecasts at numerous basins outlets (1174 outlets based on the highly distributed Cinecar rainfall-runoff model).

We modified the introduction to state more clearly that we propose here a post event analysis methodology, focused on flood forecasting evaluation, and aiming at providing a first useful analysis even if not fully comprehensive yet since only one event can be analyzed.

The coupling with a hydrological model represents a complementary tool for the verification of QPFs (since catchments can be seen as macro-raingauges with variable interception areas), given that the intermittence of the rainfall signal is dampened by the non-linearity in rainfall-runoff processes. In particular, the dynamics of the overall soil filling and depletion mechanisms and the flood routing play a fundamental role in determining results, as well as the role of the morphology of the basin that determines the time-space scale below which the variability of the rainfall field is dumped. The spatial integrating effect of a watershed filters out some of the spatial and temporal variability that complicate the point-by-point verifications that are more commonly used (Benoit et al., 2000).

Yes, indeed, the coupling with rainfall-runoff models is the key point of this paper. Our objective is to evaluate a hydrometeorological forecasting chain, and a standard evaluation of QPFs would not be satisfactory from this perspective, for the reasons mentioned by the reviewer.

2. The proposed evaluation of rainfall forecast is aimed to take into account spatial and temporal variability. The proposed analysis recalls in a some way the concept of the so-called "double-penalty effect" (i.e., the fuzzy verification introduced by Ebert, 2008 and Roberts and Lean, 2008, and discussed by Schwartz and Sobash, 2017). But the subject is treated neglecting specific past literature about this issue. Introduction and Section 4.1 should be revised accordingly.

Why has just one year of ensemble forecasts been used, given that products are available from 2018?

Thank you for having drawn our attention to these interesting references dealing with the evaluation of high resolution gridded rainfall forecasts. Since the objective of the paper is to evaluate flood forecasts, the literature review has not been focused on the evaluation of gridded rainfall products. The evaluation of flood forecasts can nevertheless be seen as a form of fuzzy verification of gridded rainfall products, taking account for the averaging effect and the non-linearity of the rainfall-runoff process, and also for the watersheds limits (which, unfortunately, are not fuzzy). We added a sentence in the introduction section to remind this possible link between flood forecasting verification and the concept of fuzzy verification of QPFs: "In one sense, flood forecasting verification can be seen as a form of fuzzy verification of rainfall forecasts (Ebert, 2008; Robert and Lean, 2008) accounting for the averaging effect and the non-linearity of the rainfall-runoff process, and also for the positions of watershed limits".

Regarding section 4.1, the objective of this section is mainly to prepare the evaluation of flood forecasts and not to provide a comprehensive evaluation of gridded QPFs. The concept of fuzzy verification could probably be applied to the rank diagrams (by building the diagrams on averaged values at different resolutions), but we consider this is out of scope of this paper.

Since some of the rainfall products are experimental, they were not released yet for the years following 2018. Extending the evaluation period would have been interesting if other significant FF events had hit the Aude catchment after 2018, with peak discharges exceeding the 10-year return period threshold. Unfortunately, this is not the case yet.

3. AROME-EPS and AROME-NWC with time lagging are merged to build an ensemble. Which are the reasons to merge the two products? Why is AROME-NWC with time lagging just used to build an ensemble?

AROME-EPS is an operational ensemble forecast product for short-range forecasting (0-45 h). AROME-NWP is an operational deterministic forecast product, specifically designed for very short lead times (0-6 h), which takes into account the latest radar observations. We are dealing here with short range nowcasting of flash-floods, and the AROME-NWP model appears well suited to this application. This is at the origin of the idea of merging both models to build an ensemble suited to very short lead times (0-6h).

4. The reasons for using two different hydrological models for different aims should be discussed.

We propose to add the following paragraph in the section "3.4 Rainfall-runoff models" : "The objective of this study is not to compare the rainfall-runoff models. Since the RS hydrographs (hydrographs simulated with Antilope rainfall observations) are systematically used as reference for the evaluation of the flood forecasts, the evaluation results should not be directly dependent on the rainfall-runoff model but rather on the nature of the rainfall forecasts used as input. The interest of using two models here is mainly to strengthen the evaluation, by involving two complementary models in terms of resolution and calibration approach: a) because of its high spatial resolution, the Cinecar models helps to extend the evaluation of discharge threshold anticipation to small ungauged catchments, b) because it was not specifically calibrated on the 2018 event (calibration on the whole 2008-2018 period), the GRSDi models offers an evaluation of the total forecast errors at gauged outlets, including both the rainfall forecasts errors and the rainfall-runoff modeling errors. This is achieved by the comparison of flood forecasts with both RS hydrographs and observed hydrographs. However, the proposed evaluation framework could also be applied by using one unique rainfall runoff model."

5. Contents of Section 4.3 should be reformulated taking into account the response times of the considered catchments. Outcomes depends concurrently by the accuracy of rainfall forecast for the event study as well as by the characteristic of the basin.

The response times are already taken into account based on the anticipation times obtained for the RF0 forecasts in section 4.2. These anticipation times are provided at the beginning of section 4.3 for outlet 3 to 6. According to these anticipation times, only the hydrographs presented for outlets 5 and 6 seem to be significantly influenced by the propagation times. This possible influence is mentioned in the text (see our answers to the specific comments).

**Specific comments**

– Line 7: "peak flood" in place of "flood rising limb", given that the statistical analysis is focused on the maximum value of the discharge forecast

We just mean here that the analysis considers forecasts issued before and during the flood rising limb. We reformulated the sentence in the following way: "The anticipation of the flood rising limb (discharge thresholds) is then analyzed, using .."

– Lines 15-17: this statement is questionable due to the limited dataset; results do not support "to draw robust conclusions". A reformulation needs.

We changed the formulation to: "to draw first conclusions".

– Lines 69-72: this content (i.e., point i) ) recalls what has mainly been done in this manuscript

We agree. The evaluation does not solve the question of statistical representativity, which is probably impossible based on a single event. This limitation is explicitly reminded in the conclusion of the paper (last paragraph). The added value of the approach is rather related to the two other points (ii and iii): detailed illustration of internal variability of forecast performance, including the case of ungauged basin outlets.

– Lines 72-73: this content (i.e., point ii) ) is questionable by the light of general comment 2)

As mentioned in the answer to comment 2, we are dealing here with flood forecasts which are already averaging the gridded rainfall forecasts, and can be seen as a form of fuzzy verification. However, this concept of fuzzy verification can

hardly be further extended to flood forecasting in our opinion, since the watershed limits are not fuzzy, and the floods should be forecasted on the right rivers (not on the neighboring rivers). It is therefore important for flood forecasting applications to illustrate the performances of forecasts along all the branches of the river network, particularly in the case of flash floods occurring on small ungauged rivers.

– Lines 89-91: this content should be revised taking into account the general comment 2).
See our answer to the general comment 2 and to the former remark. We are dealing here with flood forecasts, and the objective is not to evaluate directly the gridded QPFs. The initial analysis of rainfall forecasts is just here to prepare the evaluation of flood forecasts.

– Lines 119-121: this subject should be deeper investigated in the introduction.
The introduction has been modified to better highlight this objective.

– Lines 130-132: this subject should be deeper investigated taking into consideration the concept of fuzzy verification
See our answer to general comment 2. The objective of this first step is not to provide a comprehensive evaluation of QPFs, but just to prepare the evaluation of flood forecasts. The possible link between flood forecasts evaluation and the concept of fuzzy verification of QPFs has been presented in the introduction.

– Lines 133- 153: the proposed analysis and metrics fits well for a post-event analysis, but they are not suitable for real time operational practices, with respect to the point of view of end-users.
This is completely right, the evaluation framework corresponds to a post-event analysis. We think nevertheless that this kind of post-event evaluation could be very useful to end users to get aware about the possible limits of the flood forecasts and to learn how to efficiently use them (see our answer to general comment 1). Fortunately, end-users have also time to get prepared during the periods of low flows.

– Lines 145-146: this statement is questionable, given that an evaluation of performance based on the last hours is not indicative about the performance of hourly QPF in the following future time-steps
From our point of view, the text clearly indicates here that the rank diagrams are computed for the whole HFT (i.e. the whole intense period of the event), and not only on the last hours in a real time situation. We are not dealing here with a real time analysis.

– Lines 147-153: the use of rank diagrams to analyse a single event appears as no fully proper
The number of forecast-observation pairs used to compute the rank diagrams is 551.088, which is enough to visualize if the ensemble forecasts have significant biases and excessive or insufficient spread. One should keep in mind, however, that since the verification is performed on a rather short period and small area, the displayed rank diagrams are only used to qualify the ensemble behavior in our case study, and should not be interpreted as indicative of the long-term average performance of ensemble forecasts.

– Lines 157-162: these contents can be simply summarized by stating that the forecast is verified within a time window useful for the aims of end-users (warning issues)
We agree and thank the reviewer for the comment. We think it is however also important to mention that the different forecasts issued during this time window are aggregated and considered together to fill the contingency table. We propose the following modifications : "The evaluation is essentially based on a classical contingency table approach, with some adaptations aiming to focus the analysis on the most critical time window from a user perspective (time steps preceding the threshold exceedance), and to aggregate the forecasts issued during this time window, independently of the lead-times (see Appendix A for a detailed description of the implemented method)."

– Line 164: how is the 10-yr return period computed for the ungauged basins?
The 10-year return period discharge threshold is derived from the SHYREG database, which includes flood discharge quantiles at the outlets of ungauged catchments, for different durations and return periods ranging from 2 to 1000 years (Aubert, 2014). This origin of the 10-year thresholds is detailed in Appendix A and in the section on anticipation 4.2. A reference to Aubert et al. has been added here.

– Lines 215-222: the use of observations which were not available in real time to calibrate the hydrological models limits the operational use of the proposed forecasting chain. As well as the peak discharges estimated at ungauged locations during a post-flood field campaign, makes impossible to replicate the proposed framework for real time applications.
This is right, the evaluation procedure is not designed for a real time application, but for a post-event evaluation. This offers the opportunity to involve additional data (such as post-flood field data) and models.

– Line 233: if AROME-EPS is updated every 6 hours, it is not clear how figs 5, 6, 7, 11-16, B1-B6 show continuous hourly forecast with 1 to 6 hour lead times for each hourly time step.
A new hydrological ensemble forecast is built every hour by forcing the rainfall runoff model with the last available ANTILOPE QPEs, and the QPFs from the last AROME-EPS run for the next time steps. This means that the first time steps of the AROME-EPS runs are not systematically considered. A sentence has been added in section 4.2 to provide this explanation: "Forecasts are issued every hour, by using the ANTILOPE rainfall up to the time of forecast, and one of the 3 rainfall forecast ensembles, or a zero future rainfall scenario (RF0), for the 6 next hours."

– Line 245: "an ideal distance for the present case study" fits better than "an ideal distance".
This has been modified.

– Lines 248-253: is this comparison a novelty with respect to past literature? Why the 1-h lead time is not considered to build Fig.3? Additional rainfall accumulations larger than 5 mm/h should be considered to complete Fig.3.
This figure corresponds to a former evaluation of QPFs at larger temporal and spatial scales, and for a relatively low threshold of rainfall intensity (5 mm/h). It is therefore not completely in line with the objectives of the paper. We propose to remove this figure.

– Line 257: the description of the use of each model within the present study should be here introduced.
We added a specific development in section 3.4 to better justify the use of each model (see our answer to general comment 4).

– Line 270: specify the periods of the calibration and validation processes
The GRSDi model was calibrated for the period between October 2008 and October 2018, including the Aude's event. This has been performed at the calibration outlets shown in Figure 4. Validation was performed spatially, which means that the model was validated for the same time period but at the outlets not used for calibration. We slightly adapted the text to mention that the calibration and validation periods are the same.

– Line 271: define the acronym KGE
We modified it and added a reference describing the KGE score.

– Line 281: specify the time step at which this model runs
The model runs at 15 min time-step. We added this information in the text.

– Line 291: specify in the text the period of the temporal evolution; what does J+1 mean?
The temporal evolution shown in figure 5 is drawn from the 14[th] October 07:00 to the 15[th] October 19:00. We specified this in the text. ANTILOPE J+1 corresponds to the QPEs obtained with the ANTILOPE algorithm (Laurantin et al, 2008) readjusted with the radar and rain gauge observations available the following day (J+1). This explanation has been added in section 3.2.

– Line 292: the 1-h lead-time has poor significance for the aims of end-users (i.e., warning issues). The 3-h lead time is more significant.
Since we are dealing here with many upstream ungauged watersheds, a 1-h lead-time often exceeds the catchment response times. This is illustrated by figure 10, which shows that the anticipation times of the 10-year discharge threshold rarely exceed 1 hour based on the RF0 forecast (zero future rainfall): out of 467 sub-basins for which the threshold is exceeded by the RS scenario, the anticipation times are < 15 min for ≈230 sub-basins (misses) and in the [15 min, 1 hour] range for ≈185 other sub-basins. Considering this, we think that a 1 hour lead-time is already significant for such basins, even if we agree that 3 hours lead times would be preferable for end users but difficult to achieve.

– Caption Fig.4: define the acronym Hymex (or avoid to use it in the caption)

We added the meaning of the HyMeX acronym in the caption. It is also defined in the text, before the description of Figure 4 (l. 220-222).

– Line 304: it is not clear to what "rising limb" is referred

We modified the sentence to make it clearer: ".. a time-shift of 2 hours is observed during the rising phase of the hyetograph".

– Lines 305-309: for certain selected outlets, hyetographs for 6-h rainfall amount (for a fixed or moving average time window) should be also useful to evaluate the impact of rainfall forecast on the hydrological forecast, due to the integrating effect of the spatial-temporal variability of rainfall by the rainfall-runoff processes

A 6-hour rainfall accumulation would have the advantage to combine all the lead times. But since we are dealing with flash floods on upstream basins, a 6-h time step probably exceeds the typical response times of a large part of the considered basins. This is confirmed by the anticipation times obtained on Figure 10 for the RF0 scenario (no future rainfall). We think a 1-hour accumulation is more representative of the average response times of the considered basins.

– Lines 312-315: this statement is questionable, given that, in real-time, it is not known which areas will not contribute, even if a nowcasting forecast is available. The different scales involved between model predictions and raingauge measures, coupled with the high variability of the physical events and of the model errors, complicate the use of precipitation observations for atmospheric model validation, particularly in complex terrain endowed with a limited density of instruments. This areal variability enables to diagnose different problems associated with the atmospheric simulations, such as the quality of the larger scales simulated or the reliability of the description of small scale processes. The dependence between basins and sub-basins can be very useful to understand the possible problems of spatial shifting in the modelled atmosphere (Benoit et al., 2000; Jasper and Kaufmann, 2003).

We are not dealing here with a real time analysis, but post-event evaluation. We also benefit from a post event radar rainfall reanalysis. Therefore, the areas where significant rainfall was neither observed nor forecasted are known, which is a much more comfortable situation for the evaluation of forecasts.

– Line 319: the 1-h lead-time has poor significance for the aims of warning issues (observed rainfall plays the major role in the modelled basin response for this lead time). The 6-h lead time is more significant.

Since we are dealing with flash-floods and basins of limited areas, the observed rainfall does not necessarily play a dominant role even for a 1-hour lead time. This is confirmed by figure 10: the anticipation times rarely exceed 1 hour for the RF0 scenario (zero future rainfall).

– Line 332: the comment for line 319 is valid also here

See our answer to the former comment.

– Caption Fig.6: "amount" in place of "rates"

This has been corrected.

– Lines 345-350: these considerations should be done on the discharge ensemble (not on the ensemble QPFs), due to the non-linearity in rainfall-runoff processes

This section has been largely modified, and these statements have been moderated.

– Line 360: which is the need to run the model at 15-min time resolution?

The Cinecar models runs at a 15-min time step, which is well suited for the representation of the very fast dynamics of flash floods.

– Lines 364-365: how is the 10-yr return period computed for all the sub-basins (I guess that many of them are ungauged basins)?

The 10-year return period discharge threshold is derived from the SHYREG database, which provides a regionalization of flood discharge quantiles at the outlets of ungauged catchments. The peak discharge quantiles are estimated for return periods between 2 and 1000 years (Aubert, 2014). This explanation has been added in the text.

– Line 370: which hydrological runs were used to built Fig.8?

The hydrological runs and the method used to build the contingency table are described in l.356-369 (initial version of the manuscript) and in appendix A. The hydrological forecasts were obtained with the Cinecar model, and the contingency tables were built based on an adapted approach to combine the forecasts issued in the 6-hours preceding the "critical" times to be detected (threshold exceedances). A sentence has been added here to remind that the forecasts are issued every hour.

– Caption Fig.8: specify what represent the points on each line

The points represent the scores obtained for each percentile of the hydrological forecasts, from 5% to 95%. We added an explanation in the caption.

– Line 372: which is the starting time of RFO? How long is the RFO run driven by observed rainfall before the rainfall is set to zero?

As for the other forecasts, a RF0 forecast is issued every hour from the 14[th] October 07:00 to the 15[th] October 19:00. The observed rainfall Antilope J+1 is used from the initialization of the model (14[th] October 07:00) up to the forecast time, and 0 rainfall values are then used up to the 6-hour lead time. The following explanation has been added to explain how the different forecasts are issued: "The model is run from the 14[th] October 07:00 and hourly rainfall accumulations are uniformly disaggregated to fit the 15-min time resolution of the model. The forecasts are issued every hour, by using the ANTILOPE rainfall up to the time of forecast, and one of the 3 rainfall forecast ensembles, or a zero future rainfall scenario (RF0), for the 6 next hours."

– Lines 376-377: specify in the text the number of missed detections (as done for false alarms at line 379)

We added the number of misses in the text.

– Line 381: "contrasted effects" is not clear to what refers to.

We modified the sentence in the following way: ".. the effects of the spatial perturbation introduced by the pertDpepi ensemble differ depending on the area and ensemble percentile considered."

– Lines 383-386: these considerations could be misleading (the non-linearity in rainfall-runoff processes plays a major role; it is not an effect of what percentile to consider)

The explanation provided to the reduction of False Alarms with PertDpepi seems the most plausible to us, but we agree that other explanations could be advanced (such as a temporal shift with PertDpepi in this area). We moderated this sentence to be less affirmative.

– Caption Fig.8: are river gauge level available every 15 minutes? How can hits be computed everywhere with a 15-min time step?

We are dealing here with flash-floods in small and mostly ungauged river basins. Streamgauges are only rarely available. Thus, as explained in sections 2.2 and 4.2, and in Annex A, the simulated hydrographs obtained with the rainfall-runoff model and the observed rainfall (Antilope J+1) are considered as the reference to compute the scores. This leads to neglect the rainfall runoff modeling errors and to focus the analysis on the propagation of rainfall forecasts errors.

– Line 388: "rainfall forecast products" in place of "ensemble rainfall forecast products" (given the general validity of the sentence)

We modified this.

– Lines 390-392: the impact of RFO depends on the concentration time (i.e., the response time of the watershed to the rainfall) of the considered basins. Related false alarms decrease with the lead-time increasing (except for systematic errors in the hydrological simulation).

We agree that the anticipation with RF0 mainly depends on the response time of the considered basins, and the risks of false alarms for the hydrological forecast ensembles decrease with the increase of the response times. As illustrated by figure 10 (anticipation time with RF0), the response times are very limited for a large majority of river basins, which probably explains the high number of false alarms.

- Fig. 10: in the labels, the word "Ensemble" is not clear to what refer to
  We modified the label.

- Line 394: false alarms and misses should be also evaluated as function of different anticipation times
  As mentioned in appendix A, the anticipation times are computed by pooling together all the forecast lead times. This is the reason why all the forecasts can be summarized in a unique ROC curve (Fig.8) or anticipation map (Fig.9), or histogram of anticipation times. The advantage of this procedure is to provide a very synthetic view of the anticipation capacity. The results could alternatively be examined for fixed forecast lead-times, which would be much more conventional. In this case, we would obtain 6 different ROC curves and anticipation maps, and the comparison of anticipation times would not be possible in the same way as we did here (fig. 10).

- Lines 403-406: the question is doubtful, given that the concentration time strongly influences outcomes and the corresponding evaluation.
  We agree that the increase in anticipation times and the related risks of false alarms are probably lower for basins with large response times. However, according to figure 10, we think there is no doubt possible for basins considered here, which are mostly very small upstream basins with very limited response times. Figure 10 shows that anticipation times are significantly larger with the hydrological ensemble forecasts, but that an important number of false alarms appears in this case.

- Lines 409-414: which is the sense of the analysis in terms of PC? Is PC computed in the same way used for scores shown in Fig.9?
  Yes, the PC score is calculated from the same contingency tables as the ROC curves presented in Figure 9. We added an explanation for this in the text. The PC is a just way to summarize the contingency table obtained for each percentile of the ensemble forecasts. It allows having a unique score for each forecast product and each ensemble percentile.

- Lines 416-419: this sentence highlights the limit of the present manuscript, given that the proposed framework cannot be applied in real-time
  The objective is not here to provide a real-time evaluation, but a meaningful post event analysis (see our answer to general comment n°1). This sentence was written to highlight the importance of accounting for the relative benefits of increased anticipation and losses related with false alarms, before concluding about the actual usefulness of forecasts for an end-user. The sentence has been reformulated to better illustrate this idea.

- Lines 419-421: this statement has no sense (with respect the aims of flood warning). The accuracy of the rainfall forecast influences the quality of the hydrological forecast, but the use of RFO cannot be considered an alternative solution.
  We understand the remark. We don't mean here the RF0 scenario should be considered as an alternative solution, but just as an interesting reference for the evaluation of flood forecasts derived from rainfall ensemble forecasts. Depending on the response times of the considered basins, the RF0 scenario may sometimes offer a significant anticipation of discharge threshold exceedances. Therefore, using RF0 as reference helps to measure the actual gains in anticipation related to the use of rainfall forecasts. Moreover, by definition, the RF0 forecasts avoid false alarms, and thus helps in measuring the corresponding limits of the flood forecasts based on rainfall ensembles (generation of false alarms which can be very penalizing in real world situations).

- Line 425: quantify the size of the catchments related to outlets 1 and 2
  The drainage areas have been added in the caption of Figures 11 and 12.

- Line 426: have outlets 1 and 2 weak reaction to rainfall in general or just for this event?.
  The weak reaction is only due to the limited rainfall accumulation observed for this event.

- Line 429: quantify the size of the catchments related to outlets 3 and 4
  The drainage areas have been added in the caption of Figures 13 and 14.

- Line 432: quantify the size of the catchments related to outlets 5 and 6
  The drainage areas have been added in the caption of Figures 15 and 16.

– Line 441: briefly recall the definition of the spread/skill score and specify if it is referred to rainfall or discharge forecast
Spread-skill scores are computed for discharge forecasts. This has been mentioned in the text, and a short definition of the spread/skill score has also been added (section 2.3).

– Line 443: the choice of the lead-time should be appropriate to the concentration time of the investigated catchment to analyze outcomes. Otherwise, the outcomes seems to depend on the lead time of rainfall forecast
The lead time selected of 3 hours selected here exceeds the response times of a large majority of the selected sub basins (see the anticipation times obtained for RF0 on Figure 10). However, since we are dealing with short range nowcasting products limited to a maximum of 6 hours lead time, it is not always possible to extend the lead time beyond the response time of the considered basins. Thus, we preferred here to fix the lead time and illustrate the effect for basins of different sizes.

– Line 445: wrong label for the outlet number in Figs of appendix B (outlet 4 for all the graphs)
This has been modified.

– Lines 462-465: maybe, the outcome is affected by a spatial scale of the shift which is not optimal for the investigated catchment
Good point, we added a sentence to mention this: ".. highlighting an excess of spread in this ensemble product. It might be caused by an excessive spatial shift with respect to the geographical size of the investigated catchment."

– Line 471: which is the concentration time for these outlets?
The anticipation times obtained for outlets 3, 4, 5, 6 with the RF0 forecast (figure 10 of section 4.2) are indicated in the first paragraph of section 4.3. These anticipation times directly reflect the lead times up to which a good forecast can be obtained without rainfall forecasts. The concentration times are more difficult to determine and would be less informative about the possible forecast lead times.

– Line 481: typing error
Thank you, this has been corrected.

– Line 495: this outcome is likely influenced by the concentration time of investigated outlets
Outlet 3 and 4 are upstream outlets for which the anticipation times with the RF0 forecast are very limited: respectively 15 min and 0 min (miss). This traduces very short response times to rainfall and concentration times. Therefore, we don't think that the shape of the presented forecast hydrographs for a 3-hour lead time should be highly influenced by the concentration.

– Fig.14: in all the graphs, move the legend panel in order to do not cover lines of results
The legends have been moved in order to avoid any covering of the displayed information.

– Lines 525-527: the reasons for this outcome are the same cited at line 514 (influence of the concentration time of investigated outlets). Reformulate the sentence.

This paragraph compares the simulated and forecast hydrographs to the actually observed hydrograph, to reveal the relative importance of the rainfall-runoff modeling errors and the rainfall forecast errors. As you mention, the influence of the propagation times on the spread of the forecast has already been mentioned at line 514. The novelty here is just that the rainfall runoff modeling errors are also visible. The sentence has been reformulated to better highlight this point.

– Fig.16: in the graphs b) and c) move the legend panel in order to do not cover lines of results
As for figure 14, the legends have been moved in order to avoid any covering of the displayed information.

– Lines 548-554: a map displaying concentration times of investigated outlets satisfy the need. The threshold anticipation maps in Fig.9 describe just a case study related to the specific case study and forecast products. It cannot be used in general terms for flood warning purposes.

The anticipation times obtained for the RF0 forecast appear much easier to obtain to us, and directly traduce the specific features of the event. Concentration times would be more subjective and difficult to assess, and they cannot reflect a generic behavior of each basin, since they neglect the variations of basin dynamics due to the spatial variability of rainfall and to the intensity of the flood event.

We agree that the anticipation maps, as well as all the results presented in this paper, are event specific. This is precisely the aim of this paper, to provide an in deep analysis of the forecasts issued for one single event.

We don't see here any statement suggesting that these maps should directly be used for warning purposes. They just help to analyze some advantages/limitations of the forecasts issued during the studied event, and the links with the rainfall forecast products used as input of the chain.

– Lines 557-561: the meaning of "anticipation time" may be misunderstood. It derives by a combined effect of accuracy of currents forecast and response time of the investigated outlet.
The notion of Anticipation Time has a specific definition here, which is provided in the description of the methodology (Appendix A). We reminded this definition directly in the text of section 4.2 to avoid misunderstanding: "Note that the anticipation time is defined as the difference between the time of exceedance of the discharge threshold by the RS hydrograph and the time of the first run of forecast that detects the threshold exceedance (see Appendix A)".

– Lines 575-579: this analysis is significant when performed over a long dataset
We agree, but unfortunately the analysis on a long dataset cannot be performed for these products (see answer to general comment 1).

– Lines 584-595: the sense of these considerations is related to the role of QPFs in general, not specifically to ensembles.
We agree this corresponds to the general gains expected by using deterministic or ensemble QPFs. However, these effects are observed only if the QPFs are good enough, which is unfortunately not systematically the case yet for major flash floods events with huge spatio-temporal variability.

– Lines 596-601: the gain is due to NWC. Which is the added value to use NWC+EPS rather than just NWC? NWC alone was not considered in this work because there is no simple way to generate an ensemble forecast from it with a reasonable size (>5). The generation of ensembles from NWC forecasts is a complex topic which will be considered in a future study.

– Lines 619-621: maybe, the extension of 20 km is not the optimal dimension for the investigated case study. An investigation about this issue is worth to be performed.
This comment pertains to the description of the pertDpepi method (lines 241-247). The following development has been added in this section : "The shift scale of 20 km represents a typical forecast location error scale: according to Vincendon et al. (2011), 80% of location errors are less than 50 km. The value of 20 km has been empirically tuned to produce the largest possible ensemble spread on a set of similarly intense precipitation cases, without noticeably degrading the ensemble predictive value as measured by user-oriented scores such as the area under the ROC curve."

– Line 639: have authors considered to apply spatial perturbations just to NWC members?
See the response to comment about lines 596-601 above: the generation of an ensemble from NWC alone is an interesting option, but it is outside the scope of this study. It could conceivably lead to a better ensemble than using EPS, but more work would be needed to reach this goal.

– Lines 727-728: these contents may be misleading. The outcome depends specifically on the accuracy of the ensemble for the case study. It is not an information that can be estimated a priori by means of a statistical analysis and generally related to the lead time. It is strictly related to the investigated event and selected run of the ensemble. These contents can be referred just to a post-event analysis (and cannot be inferred for real-time operational practices).
We agree, the anticipation times computed here are event specific and just provide an in deep-post event analysis (see our response to general comment 1).

**References used in the review comment**

– Benoit R, Pellerin P, Kouwen N, Ritchie H, Donaldson N, Joe P, Soulis E (2000) Toward the use of coupled atmospheric and hydrologic models at regional scale. Mon Wea Rev 128: 1681–1706

– Ebert, 2008: Fuzzy verification of high resolution gridded forecasts: A review and proposed framework. Meteor. Appl., 15, 51–64, doi:10.1002/met.25

– Fortin, V., Abaza, M., Anctil, F., and Turcotte, R. (2014). Why should ensemble spread match the RMSE of the ensemble mean?. Journal of Hydrometeorology, 15(4), 1708-1713

– Jasper K, Kaufmann P (2003) Coupled runoff simulations as validation tools for atmospheric models at the regional scale. Q J R Meteorol Soc 129: 673–693

– Roberts, N. M., and H. W. Lean, 2008: Scale-selective verification of rainfall accumulations from high-resolution forecasts of convective events. Mon. Wea. Rev., 136, 78–97, doi:10.1175/ 2007MWR2123.1

– Schwartz, C. S. and Sobash, R. A.: Generating probabilistic forecasts from convection-allowing ensembles using neighborhood approaches: A review and recommendations, Mon. Weather Rev., 145, 3397–3418, https://doi.org/10.1175/MWR-D-16-0400.1, 2017.

---

## Referee Report (RR1)

The paper proposes a methodological framework for evaluating short-range flash-flood hydrometeorological ensemble forecasts at the event scale, and tests it on a major flash-flood event that hit the Aude River (France) in October 2018.

I read the manuscript for the first time after a first revision, which substantially addresses all the concerns raised by previous reviewers. The authors have made several changes in the paper structure, added explanations and the paper has improved.

Based on my personal reading, I believe that the manuscript is now well written and organized; the presented methodology is interesting for operational forecasting models. I believe that the manuscript deserves to be published in NHESS, after very minor revisions/clarifications:

- Line 245 (typos):  spatial is spatial
- Lines 300-305: referring to figure 4, is it a) referring to 1-hour lead time and b) to 6-hour, or vice-versa? When you write "except at the end of the rainfall event, on the 15$^{th}$ October between 7:00 and 11:00 utc, where all ensemble forecasts overestimate the rainfall rates, particularly for the 1-hour lead time forecast", it seems to me that this is more evident in figure 4 b), that is 6-hour lead time forecast. And when you write "for the 6-hour lead time forecast a time shift of 2 hours is observed during the rising phase ", it seems to me more evident in figure 4a), that is 1-hour lead time.
- Figure 7: I suggest to add in the caption the definition of POD and FAR (they are defined in appendix A)

---

## Referee Report (RR2)

I read the manuscript for the second time. The authors have improved the paper with clearer explanations and some changes in the paper structure, and convincingly replied to reviewers' comments.
Based on my personal reading, I believe the paper deserves to be published in NHESS.

---

## Author Response (AR2)

The revised version of the manuscript is characterized just by minor revisions with respect to the first submission. Several essential issues still remain unsolved. Actually, authors have not fulfilled the majority of main concerns highlighted in the first review process. The current contents of the paper represent a post-event analysis; there is no novelty and originality of contents (meteo-hydrological forecasting tools, statistical forecast verification, outcomes). Many themes are discussed throughout the manuscript, but none of them is deeply investigated by introducing innovative tools or results with respect to past studies. Challenging purposes are proposed, but the described outcomes do not allow supporting fully statements written throughout the manuscript. Even, the discussion of results brings to face with questions that remain unsolved. Many parts of the text describe reasonings and outcomes characterised by a weak significance or obvious conclusions (in particular, Sections 5 and 6; for instance, several times the outcome of the discussion of results is to state that ensembles are useful with respect to the scenario of using zero rainfall as forecast). In my opinion, the current manuscript resembles a technical or internal report that may be of interest for local forecasters and end-users (in particular, Sections 2, 3 and 4), but its soundness for researchers and readers of an international peer-reviewed journal is weak.

Main general concerns are recalled below.

We thank referee n°3 for this feedback about the revised version of the paper. Despite our point to point answers to the remarks formulated in the first turn of reviews, it seems that there is still some misunderstanding about the scope and the novelty brought by this paper. We provide further explanation on these aspects in our detailed answers below. The novelty and originality of the paper lies in our dealing with short-range flash-flood forecasts. This requires specific QPFs ensemble products (high temporal and spatial resolution, high refresh rate, seamless forecasts, …) that have, to our knowledge, never been used in any comparable study. These original forecasting tools are essential for addressing the large spatio-temporal variability and limited predictability of the heavy precipitation events generating these floods. The paper involves two new experimental ensemble products that aim to address these specific requirements, and have never been evaluated before. As of today, there is no consensus in the numerical weather prediction (NWP) community about the best approach to cater for the needs of flash-flood numerical prediction models at these scales. Such new experimental rain products are still not operationally available, and they are not available on enough past high precipitation events for hydrologists to evaluate the performance and make decisions on their added value (i.e., changing (or not) the configuration of operational systems). The usefulness of this paper lies in its comparison of the relative merits of several NWP approaches that could be implemented operationally in the near future, which is a question of strategic importance and societal significance in many meteorological and hydrological prediction institutes. To that end, we have presented a first evaluation focused on the capacity to provide efficient forecasts for one single and intense flash flood event.

The novelty brought by the proposed evaluation framework lies in the way the conventional metrics are combined and adapted, to obtain an as detailed and meaningful evaluation as possible of the hydrological forecasts for the considered event. We think that this kind of event-based evaluation can bring interesting complements to the conventional large-scale and statistically more representative evaluation of ensemble forecasts, which remains of course necessary. By definition, high impact flash flood cases are rare, and state-of-the-art numerical prediction tools are computationally expensive and continuously evolving. Thus, it is often impossible to properly assess the performance of these tools in a statistical way: event-based evaluations are unavoidable. Both approaches (statistical and event-based) should not be opposed but rather considered as complementary, and this opinion seems to be increasingly shared by the community (see for instance p.3-4 of this report of a ECMWF worshop on model uncertainty). The reviewer rightly points out that statistical evaluation of ensemble predictions of precipitation has already been performed in many studies, so there would be no point in bloating the article with yet another one. We believe that it is more productive in focusing the paper on the more original aspects of event-based evaluation challenges. Event based evaluations have been shown to be very useful communication tools to exchange with and get feedback from end users (Dasgupta et al., 2023). Our paper does not claim that event-based evaluations provide a complete picture of the performance of forecasting systems, we merely claim that such evaluations are important tools for designing of these systems and documenting their performance in rare, high-impact flood cases.

Our study aims at opening a discussion on this issue of event based evaluation. The introduction section of the paper has been adapted to make this objective appearing more clearly. We are aware that finding a solution is complex and might take some time (if ever) to be achieved. We believe however that the research community needs to address this issue. Our paper tries to contribute to this discussion: how can we better evaluate the quality of new hydrometeorological forecasting systems

that target to improve flash-flood forecasts? How can one decide which system is the best when we only have reforecasts for a single flood event to evaluate at a given river basin? We believe that these questions are important to the forecasting community, and they are clearly of interest to forecasting offices in many regions of the world that are affected by similar flash flood events. Kilometric-scale ensemble numerical weather prediction systems are only beginning to be used for nowcasting purposes, and their application to flash flood prediction is a timely question that will interest many readers, given the growing economical and human impacts of flash-floods worlwide.

The declared aim to focus on the needs of civil protection authorities appears incomplete: some verifications on QPFs (in particular, Figs 5 and 6) and discharge forecasts (Figs 10-15) were discussed just for very short lead times (the 1-h lead time in Figs 5-6, the 3-h lead time in Figs 10-15), neglecting longer lead times which are more proper and useful for the aim of warnings by authorities in charge of decisions in case of flood. Lead times shorter than 6-12 hours do not allow issuing timely warning and take effectively actions for safety and emergency services (bearing in mind also the time to collect observed data, run the hydrological models, analyze result and issue warnings).

The rainfall forecasts evaluated here correspond to short range forecasts limited to 6 hours lead-time. We fully agree that warnings issued with larger lead-times are very useful for preparedness actions. But lead times shorter than 6-12 hours are a reality for many operational forecasters and emergency managers dealing with flash floods (as is the case of the Mediterranean flash flood we are analysing in our paper), because of the fast evolution and limited predictability of the triggering heavy precipitation events. Many flood events have a too low predictability for warnings to be usefully issued more than a few hours in advance (Davolio et al., 2017; Carrio et al., 2022), as (again) demonstrated by several catastrophic events in the Mediterranean area in 2022. For this reason, several research contributions dealing with flash floods have focused in the last years on very short range (<6h) forecasts, based on radar advection and NWP blending approaches, and/or radar data assimilation in NWP models (Bowler et al., 2006; Berenguer et al., 2011; Silvestro and Rebora, 2012; Davolio et al., 2017; Poletti et al., 2019; Zanchetta and Coulibaly, 2020; Lovat et al., 2022).

Civil protection authorities which are facing flash-floods also express the need for short-range (<6h) forecasts, issued with high refreshment frequency, to help in localizing more accurately in space and time the areas at risk during the development of these events. Delivering flash flood forecasts with up to 6h lead time would represent a significant improvement in comparison with currently existing flash flood monitoring or now-casting systems, which often still rely on radar rainfall observations (Gourley et al., 2017). Particularly, the need for short range forecasts has been confirmed within the PICS project (https://pics.ifsttar.fr/en), through exchanges with an end-users group including the varied authorities involved in flash-flood crisis management in France (Javelle et al., 2021). The experimental short range-rainfall products studied in this paper have been specifically released to address this demand. This is the reason why some studied products are not yet operational, and also why this study specifically focuses on short lead-times.

Nevertheless, since the studied forecasts are released for up to 6-hour lead times, it is possible to illustrate the results obtained for a wider range of lead-times from 1 hour to 6 hours, for the rainfall forecasts (fig. 5-6) and the forecast hydrographs (fig. 10-15). We propose to include all these results in the revised version of the paper: the figures 5-6 and 10-15 have been all focused on the intermediate 3-hours lead-time, and the 1-hour and 6-hour lead times have been presented in a new appendix (Appendix A) for rainfall forecasts, and in appendix C for forecast hydrographs (corresponding to appendix B in the former version of the manuscript).

Another desired goal of the manuscript is to draw lessons from the analysis of the selected case study for the users of hydrological forecasts. But, two of the proposed ensemble systems are not routinely run (it seems that the experimental phase about these tools cover just the year 2018). It is not clear the sense of investigating performance of these products with respect to the aim of end-users if they have not at disposal such forecasting tools in the operational practice.

In the operational practice of flood forecasting, new tools and forecasting systems are rarely tested in real-time before being tested first, and evaluated, in research. This is because decisions involving flood forecasts and warnings/alerts impact human lives and hence have to be carefully taken. In many National flood forecasting services, forecasters are legally responsible for the warnings they launch (or not) and their consequences, in particular when human life losses are involved. This explains why the systems being evaluated in our paper are designed for operational use but are evaluated within a research project first, before implementation in real-time (PICS Project, https://pics.ifsttar.fr/en, which involved both research labs and operational flood forecasting services). The complexity of this evaluation is the question at the heart of this paper. The tested products have been proposed specifically to address the demand for short-range (6h) seamless ensemble forecasts, with a 1h refreshment rate.

Two of these products are experimental, and are generated in an original way, by merging different runs of two convection permitting NWP models, including one ensemble run and several (time lagged) deterministic runs. Since they are experimental and computationally expensive, these products have been released only for selected events of year 2018 and are not routinely run yet. Non-real time evaluation, as we have done here, is a prerequisite for establishing confidence that the tested systems are worth running in real time for preoperational evaluation, which will be the next step, but it is beyond the scope of our study. The first evaluation we propose in this paper aims to verify if such products could be useful for flash flood forecasting, and can be expected to have value for end users. Such preliminary analyses are necessary before the experimental products can be adapted and tested on larger periods to build statistically more robust conclusions, and finally be integrated in operational workflows. A new development has been added in the introduction section of the paper to clarify these aspects.

The way to build the ensembles based on time lagging (i.e., "pepi") and spatial shift (i.e., "pertDpepi") could be questionable about some characteristics that appear as unsolved (maybe, additional investigations would be useful to improve the characteristics of the two ensembles). On the one hand, it is not clear the physical meaning underlying the way to build the ensemble based on AROME-EPS and AROME-NWC (i.e., "pepi"). The main and only reason seems to obtain an ensemble with a certain number of members to run the hydrological model. But, the gain in performance of "pepi" are mainly due to NWC and the time lagging, the members associated to EPS do not give an added value. On the other hand, the spatial scale of the shift for "pertDpepi" appears as not optimal for the investigated study area (a sensitivity analysis of the extension of the spatial shift should be proposed).

We fully agree that the precipitation ensembles used could still be improved. Further improving the atmospheric ensembles would be a big endeavor that is beyond the scope of this paper. This paper just presents a first hydrological evaluation of three products, among which two are experimental and were specifically designed to address the need for seamless very short range rainfall forecasts to better anticipate flash floods. A sentence has been added in section 3.3 to better explain the origin of these new products. The point of using these precipitation ensembles in this study is to show that they clearly impact forecast performance for the considered event, as demonstrated by the ROC evaluations in fig 8. There, the ROC curves are well above the diagonal for all ensembles, which proves that the ensembles carry predictive value, regardless of the ensemble perturbation details, for this event. Regarding the way of building the ensembles:

- The AROME-EPS ensemble is a state-of-the-art weather forecasting ensemble and its perturbations are physically based as explained in the linked journal papers (Bouttier et al., 2012; Raynaud and Bouttier, 2016);

- The PEPI and pertDpepi ensembles are based on lagging and statistical field perturbation, which is the currently dominant approach to generate short-range precipitation ensembles (?Lu et al., 2007; Bowler et al., 2006).

Our study is the first that compares these both types of ensembles from a hydrological point of view, so it gives novel information about their respective merits, even if this first evaluation is focused here on one single event and can obviously not be extrapolated.

It should also be noted that these ensembles (pepi, pertDpepi) are not limited to run over the Aude river catchment only. They are designed to run over the whole French territory (as it is usually the case with NWP models in meteorology). Therefore, perturbations and other steps implemented when building the ensembles cannot be calibrated to satisfy one event in one river basin. This additional complexity justifies the event-based evaluation we are proposing in this paper.

The verification metrics are commonly used over large datasets, in order to highlight statistical characteristics of the forecast product. The use of verification metrics to analyze a single event has poor significance (rank histogram, ROC curves and spread-skill relationship in Figs. 6, 7 and panels a) in Figs 10-15, respectively).

We agree that the ensemble verification metrics are generally used over large datasets to reach statistical significance. An analysis focused on one single event cannot reach this objective. On the other hand, analyses focusing on single events are also needed (see for instance the conclusions of ECMWF workshop on model uncertainty). They enable to delve into the precise characteristics of forecasts, and to illustrate the information obtained for the most intense and critical events. This is the reason why they are interesting for end-users. We think both approaches are complementary and should not be opposed. What we propose here is only a single event analysis. We do not intend for the metrics we use to deliver statistically significant conclusions, they are just used here for characterizing the detailed behaviour of the ensembles for the analyzed event, and they are combined and adapted in an original way to this purpose. We added several sentences in the manuscript to avoid

any misleading on this point, and to remind that the conclusions are valid only for the considered event and should not be extrapolated to future events.

Even though the focus is to evaluate a rare flood event, a statistical analysis of discharges performed over a long dataset is significant to test false alarms due to potential overestimation of QPFs (especially, for events on very small areas like many of the investigated sub-basins). Otherwise, a deceptive reliability about the forecasting tool could be induced in the users of those forecasts. The statistical analysis in terms of discharge forecast should consider at least the whole period covered by QPFs (in the first submission, the performance of the meteo ensembles were shown for the whole year 2018), not just a flood event.

We agree that a statistical analysis over a long continuous period would be essential to characterize the risks of false alarms, and the actual statistical performance of the forecasts. The period of such an analysis should probably largely exceed the year 2018 and the Aude area, to include a significant number of flood events exceeding the 10-year discharge threshold (which is a relevant threshold in our opinion, since often corresponding to the observation of first significant inundations and damages). This is not yet possible, since the experimental rainfall ensembles we used here have been released only for a few intense rainfall events of 2018 (Aude river in October, Ardèche and Cèze rivers in August and Argens rivers in October), among which only the October flood in the Aude river exceeds a 10-year return period. The evaluation of the rainfall forecasts presented in the initial version of the manuscript was pooling all these three events, for the whole product geographic window covered by the rainfall forecast products, but its statistical significance also remains limited. The event analysis provided in the paper enables to examine in detail the hydrological forecasts obtained for one interesting intense flood event. The performance observed on such an event can help to decide if it is worth conducting a continuous analysis on a larger period of time, to draw robust statistical conclusions.

The selected case study has been already investigated by some past studies, at least by the meteorological point of view (with the same or similar QPF forecasting tools).

The event has been studied from a meteorological point of view by Caumont et al. (2021), including the performances of QPFs based on AROME and AROME EPS. Lovat et al. (2022) studied the performance of deterministic hydrological forecasts for this event, using AROME-NWC and PIAF short-range QPFs. Other studies focused on this event to improve a hydrological model (Peredo et al., 2022), or to evaluate automated flood mapping approaches (Hocini et al., 2021). But none of these studies have focused yet on short range ensemble hydrological forecasting, and the experimental pepi and pertDpepi QPF products have never been evaluated yet from a hydrological forecasting perspective.

The proposed framework for the evaluation of meteo-hydrological model coupling does not represent an innovation, a new approach. That analysis collects different approaches commonly used in the worldwide operational practice for the verification of accuracy of ensemble precipitation forecasts and to convey and analyze the information provided in terms of discharge forecast by a meteo-hydrological forecasting chain. Roughly summarizing, the forecast is verified over a study area of interest for local end-users (identified as "HFA") within a time window useful to issue warnings (identified as "HFT").

The novelty of the proposed approach lies in two aspects in our opinion: 1 - it proposes a new combination of well known evaluation metrics in order to provide a synthetic and as informative as possible analysis of the considered event, and 2 - the common contingency table and ROC curve approach has been adapted here (see appendix) to obtain one unique ROC curve summarizing the performance for all the lead-times, and enabling them to examine separately the anticipation times. To our knowledge, similar approaches have not been proposed yet in the literature: if we are wrong, we would be grateful to obtain the corresponding references.

The analysis related to anticipation times for the hydrological forecasts (i.e., Fig 9) is questionable and misleading. Outcomes depends on concurrently by the accuracy of rainfall forecast for the event study as well as by the characteristic of the basin (namely, the response times of the considered catchments to rainfall). Many statements describe reasoning of weak significance that lead to obvious outcomes or are strictly valid for the selected runs of QPFs. The representativeness and significant level of contents conveyed by Fig.9 are weak to gain insight about the proposed tools for flood forecasts (in particular when aimed to warning purposes).

We completely agree that the anticipation times presented on Fig.9 depend both on the accuracy of rainfall forecasts and the response times of the considered basins. This is the reason why the RF0 (zero future rainfall) forecast is presented on this figure: this reference run illustrates the part of anticipation that is only due to the basins' response times. The comparison of the anticipation times obtained with RF0 and with the QPF products shows the gains in anticipation associated with the QPF products. This important role of the reference RF0 forecast is explained in section 2.2 of the manuscript. The conclusion here is

that the gain in anticipation is significant with the QPFs for the studied event, which is in our opinion important information to characterize the added value of forecasts for this specific event. This conclusion should obviously not be extrapolated to future events: we have reminded this in the text to avoid misleading interpretation. We do not see in the comments of this figure any other questionable or misleading points.

**Anonymous Referee 4**

We thank referee n°4 for this very positive evaluation and his suggestions to improve the manuscript.

– Line 245 (typos): spatial is spatial We corrected this typing mistake.

– Lines 300-305: referring to figure 4, is it a) referring to 1-hour lead time and b) to 6-hour, or viceversa? When you write "except at the end of the rainfall event, on the 15th October between 7:00 and 11:00 utc, where all ensemble forecasts overestimate the rainfall rates, particularly for the 1-hour lead time forecast", it seems to me that this is more evident in figure 4 b), that is 6-hour lead time forecast. And when you write "for the 6-hour lead time forecast a time shift of 2 hours is observed during the rising phase ", it seems to me more evident in figure 4a), that is 1-hour lead time.

The paper now includes the results for a 3-hour lead time in figure 4, and for the 1-hour and 6-hour lead times in appendix A. We changed the analysis to mention that the overestimation at the end of the rainfall event is present rather for the 3-hour and 6-hour lead times, and that only the 6-hour lead-time shows a relatively systematic time shift of 1 to 2 hours during the whole event.

– Figure 7: I suggest to add in the caption the definition of POD and FAR (they are defined in appendix A)

We added the definitions in the caption of Fig.7

[revised manuscript text omitted]